**Subject Category:**
Biology (whole organism)

palaeontology/taxonomy and systematics/ evolution

feeding strategy, ecological niche, systematics, Ziphiidae

**Author for correspondence:**
Benjamin Ramassamy
e-mail: benjamin.ramassamy@laposte.net

# A new specimen of Ziphiidae (Cetacea, Odontoceti) from the late Miocene of Denmark with morphological evidence for suction feeding behaviour

## Benjamin Ramassamy[1] and Henrik Lauridsen[2]

[1]Department of Natural History and Palaeontology, The Museum of Southern Jutland, Lergravsvej 2, Gram 6510, Denmark
[2]Comparative Medicine Lab, Department of Clinical Medicine, Aarhus University, Palle Juul-Jensens Boulevard 99, Aarhus 8200, Denmark

(iD) BR, 0000-0002-2112-9167; HL, 0000-0002-8833-4456

A new fossil of Ziphiidae from the upper Miocene Gram Formation (*ca* 9.9–7.2 Ma) is described herein. Computed tomographic scanning of the specimen was performed to visualize the mandibles and to obtain a three-dimensional digital reconstruction. It possesses several characters of the derived ziphiids, such as the dorsoventral thickening of the anterior process of the periotic, the dorsoventral compression of the pars cochlearis and the short unfused symphysis. The specimen cannot be identified beyond the family level, because of the unusual nature of the preserved parts consisting of the mandibles, earbones and postcranial remains. It differs from other ziphiid species from the Gram Formation, *Dagonodum mojnum*, in its larger size and the more derived morphology of its mandibles and earbones. Its long and thickened stylohyal, combined with its reduced teeth, suggests that this new specimen relied primarily on suction feeding. By contrast, the other ziphiid species from the Gram Formation, *D. mojnum*, shows adaptations for a more raptorial feeding strategy. Assuming the two species were coeval, their co-occurrence at the same locality with two different feeding strategies, may represent a case of niche separation. They may have hunted different types of prey, thus avoiding direct competition for the same food resource.

## 1. Introduction

Beaked whales (Ziphiidae) represent a diversified family of echolocating toothed whales (Odontoceti), currently represented

by at least 22 species in six genera [1] with a potential new species of *Berardius* suspected in the North Pacific [2]. Their best-known modern representatives are capable of regular deep dives beyond 1000 m to reach their foraging grounds, where they prey mostly on cephalopods and more occasionally on bathypelagic fish and crustaceans [3–10]. Most extant ziphiids are typified by a strong reduction of their tooth count to one or two mandibular pairs, often only erupted in adult males [11]. Beaked whales do not use them to capture or manipulate their prey; instead, they use suction as their main feeding strategy, except perhaps for the toothed ziphiid *Tasmacetus shepherdi* which retains a set of functional teeth in both the upper and lower jaws [4]. Suction feeding forces ziphiids to be more selective with respect to the size of their prey, thus allowing different species of beaked whales to be sympatric without competing for the same food resource [12,13].

Recently, Hocking *et al.* [14] proposed a new framework to understand the evolution of feeding in predatory aquatic mammals. Instead of thinking of the different feeding styles as rigid categories, they argue that feeding strategies of aquatic mammals follow a particular evolutionary sequence that can be used to predict the origin of particular feeding styles. Under this framework, the specialization for suction feeding of extant beaked whales should arise from ancestors that used a more raptorial feeding strategy. The fossil record of Ziphiidae confirms this prediction: some of the most basal beaked whales possessed elongated jaws and numerous functional interlocking teeth potentially used to capture their prey [15–18]. However, morphological evidence suggest that some of them were also capable of using suction at least in the most posterior part of the mandibles [17,18]. For example, *Dagonodum mojnum*, a late Miocene ziphiid from the Gram Formation of Denmark was interpreted as a more raptorial feeder than extant beaked whales based on its numerous interlocking teeth and elongated jaws, despite moderate adaptations to suction feeding [18].

A new fossil Ziphiidae from the same locality is described here. The preserved parts of the specimen consist of the lower mandibles, earbones, part of the hyoid apparatus and forelimb elements. This paper aims at describing the specimen and proposing a reconstruction of its autecology based on morphological features. Aspects of feeding strategies and ecological niches occupied by the ziphiids from the Gram Formation are also discussed.

# 2. Material and methods

## 2.1. Specimen preparation and computed tomography

The specimen was discovered in 2007 and prepared by means of mechanical tools at the curatorial department of the Museum of Southern Jutland (Denmark). A co-polymer of acrylates (MA/EMA Paraloid B72) was used as an adhesive to keep the fragments of the lower jaw together. Photos of the specimen were taken using a Fujifilm FinePix HS10 with a focal length of 4.2–126.0 mm.

Specimens coming from the Gram Formation are fragile, difficult to handle and prepare. Furthermore, the preparation sometimes results in the loss of information about the original placement of the bone structures. Similar use of computed tomography (CT) analysis has already been applied to fossil Ziphiidae with great success [19].

To alleviate the preparation work and avoid extensive manipulation of the specimen, the lower jaws were scanned using a clinical CT system (Siemens Somatom; Siemens Medical Solutions, Forchheim, Germany) with the following parameters: $0.98 \times 0.98 \times 0.60 \, \text{mm}^3$ voxel size; 140 kVp tube voltage; 185 µA s tube charge, resulting in an acquisition time of approximately 60 s. Data were reconstructed using a B45s convolution kernel. The three-dimensional reconstruction unveiled the dorsal and lateral side of the lower jaws as preserved that otherwise would have not been accessible without extensive preparation. Visualizations of the scanned fossil were done using the DICOM-viewer OsiriX (Pixmeo SARL) and image segmentation and construction of an interactive model of the fossil were done in Amira 5.6 (FEI, Visualization Sciences Group). The digital reconstruction is available in electronic supplementary material, figure S1.

## 2.2. Geological and palaeoenvironmental setting

Originally, three members were recognized in the Gram Formation: the lowermost glaucony-rich clay, the Gram clay and the Gram Sand member [20]. The Glauconite clay member is now recognized as part of the Ørnhøj Formation and the Gram Sand member as the Marbæk Formation [21]. The type section is found at the Gram Formation where a 13.1 m thick section of Gram clay is exposed [21]. Neither the base nor the top is visible, the reference section being 16 m thick [22]. The Gram Formation consists of

dark brown clay with siderite concretions in the lower part and a few fine-grained wave rippled sand beds in the upper part [21]. The Gram Formation was deposited in a fully marine environment with water depth up to 100 m [23]. The occurrence of storm beds in the upper part suggests a progradation of the shoreline [21].

Estimation of the age of the formation is based on several lines of evidence. Rasmussen [20] identified five biozones based on the abundance of mollusc fauna found in the Gram Formation. A more recent biostratigraphy based on dinoflagellate cysts suggests that the Gram Formation was deposited between the late Serravalian and Tortonian age with the consistent occurrences of the dinoflagellate cysts *Hystrichosphaeris obscura*, *Spiniferites solidago* and *Labyrinthodinium truncatum* [24]. The most precise age estimation of the Gram Formation is indicated by a magnetic analysis of a 16 m vertical profile [25]. Beyer [25] identified a reverse polarity zone of less than 70 000 years at 14.8 m deep, approximately the basis of the mollusc biozones identified by Rasmussen [22]. This leaves three possible datations for the Tortonian stage: 7.1, 7.4 and 9.9 Ma [25]. Furthermore, the analysis of the accumulation rates indicates that the Gram Formation was deposited during 120 000 years with a much faster deposition rate in the uppermost 8 m of the formation (approx. 20 000 years) [25].

Based on these multiple lines of evidence, the Gram Formation can be dated from the Tortonian age with a maximum age of 9.9 Ma (based on the reverse polarity zone) and a minimum age of 7.2 Ma corresponding to the Tortonian–Messinian boundary (based on the dinoflagellate cysts biostratigraphy).

## 2.3. Size estimation and evaluation of trophic level

Cetaceans, particularly obligate suction feeders, are known to select their prey relative to their own size [12]. Therefore, assessing the size of a ziphiid individual and comparing it with other species may help estimating the trophic level at which it used to feed.

To do so, two cranial measurements were collected from different ziphiid specimens: the bizygomatic width and the condylobasal length (data available in electronic supplementary material, dataset S2). Many fossil specimens had to be discarded, because their partial skull did not allow a good estimation of the condylobasal length and/or the bizygomatic width. The fossil species *Ninoziphius platyrostris*, *Nazcacetus urbinai* and *Messapicetus gregarius* were included based on the measurements provided in their respective descriptions [16,17,26]. In the absence of a preserved skull for the specimen NHMD 189993 described herein, such measurements were not available. The anteroposterior length and posterior transverse width of the mandibles were used instead of the bizygomatic width and the condylobasal length, respectively. The posterior transverse width of the mandibles is a good estimator of the bizygomatic width but in this case, results in a slight underestimation of the latter dimension due to the lack of the most posterior parts of the mandibles. Anteroposterior length of the mandibles is significantly shorter than bizygomatic width in odontocetes; as such the latter dimension should only be taken as an indicator of minimum size rather than a precise proxy. Cranial measurements were nonetheless selected for other ziphiids, because the mandibles of beaked whales are often disarticulated and the posterior width of the mandibles is, therefore, not always measurable.

Ziphiid species were regarded as representatives of four size categories: very large-sized ziphiids (8–10 m), large-sized ziphiids (5.5–7.5 m), medium-sized ziphiids (4–4.5 m) and small-sized ziphiids (3–4 m). Those categories were defined in Bianucci *et al*. [27] based on a regression of the postorbital width relative to the body length of different ziphiid species.

A natural logarithmic transformation was applied to the cranial measurements to attenuate the effect of allometry and correct for heteroscedasticity [28,29]. A MANOVA (multivariate analysis of variance) was performed to evaluate whether the cranial measurements were sufficient to assess each size category. It was followed by a Tukey's honest significant difference test on each variable to compare differences between the size categories. Linear regression was also performed on the dataset to assess the relationship between the two cranial measurements. All analyses were performed with the software R v. 3.6.0 [30].

## 2.4. Nomenclature

*Institutional Abbreviations*—IRSNB, Institut Royal des Sciences Naturelles de Belgique, Brussels, Belgium; MNHN, Muséum National d'Histoire Naturelle, Paris, France; MSM, Museum Sønderjylland Naturhistorie og Palæontologi, Gram Lergrav, Gram, Denmark; MSNUP, Museo di Storia Naturale dell'Università di Pisa, Italy; MUSM, Museo de Historia Natural, Lima, Peru; NMNZ, National Museum of New Zealand Te Papa Tongarewa, Wellington, New Zealand; NHMD, Statens

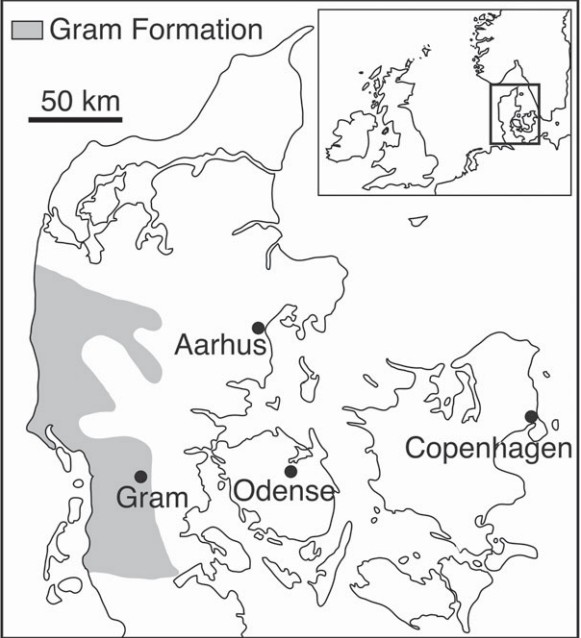

**Figure 1.** Current extension of the outcrops of the Gram Formation in Denmark (shaded area). The finding site is situated in the Gram claypit, 1.5 km north of Gram. Modified from Rasmussen [20].

Naturhistoriske Museum, Copenhagen, Denmark; USNM, United States National Museum of Natural History, Smithsonian Institute, Washington, DC, USA.

*Anatomical terminology*—The anatomical terminology of the skull follows Mead and Fordyce [31]. The terminology used by Fitzgerald [32] and Marx *et al.* [33] was followed for the postcranial remains. The nomenclature of Reidenberg and Laitman [34] was used for describing elements of the hyoid apparatus.

# 3. Results

## 3.1. Systematic palaeontology

Order CETACEA Brisson, 1762
 Suborder ODONTOCETI Flower, 1867
 Family ZIPHIIDAE Gray, 1850
 Genus and species indet.

*Referred Material*—NHMD 189993, subcomplete mandibles, the right stylohyal, 14 isolated teeth including one tusk, two periotics and the right tympanic, the right humerus and associated radius, parts of the nasal (unambiguous identification of a side is impossible).

*Horizon and Locality*—The finding locality is situated 1.5 km north of the town of Gram, Southern Jutland, Denmark (55°18025.67″ N, 9°3032.51″ E; figure 1). The specimen NHMD 189993 was dated on the basis of the associated mollusc fauna assemblage. The high percentage of *Carinastarte vetula reimersi* (accounting for 55% of the specimens identified) and the co-occurrence of the species *Gemmula badensis* and *Turritella tricarinata* suggest that the specimen was originally found in the assemblage Zone V [20].

The assemblage Zone V belongs to the upper part of the Gram Formation dated from the Tortonian age, based on the co-occurrence of the dinoflagellate cysts *Hystrichosphaeris obscura*, *L. truncatum* and *Spiniferis solidago* [24]. The maximum age of the assemblage zones from the Gram Formation is estimated to 9.9 Ma based on the presence of a polarity zone shorter than 70 000 years [25]. NHMD 189993 can, thus, be dated from the mid- to late Tortonian, *ca* 9.9–7.2 Ma.

*Systematic Attribution of the Specimen*—The specimen is assigned to the family Ziphiidae based on the following combination of characters: the enlargement of the apical or subapical mandibular tooth; the reduction of the dorsal keel on the posterior process of the periotic; the mediolateral thickening of the anterior process of the periotic; in dorsal view, the anterior shift of the pars cochlearis of the periotic.

NHMD 189993 clearly differs from the other species found in the Gram Formation, *D. mojnum*, based on the following characters: the reduction of the mandibular teeth; the shorter unfused symphysis; the

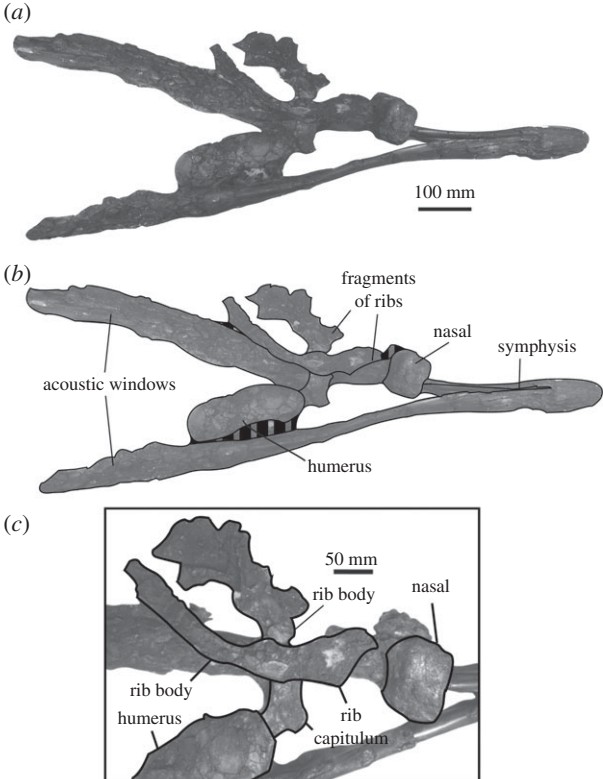

**Figure 2.** Mandibles, cranial and postcranial remains of NHMD 189993. (*a*) ventral view; (*b*) corresponding drawing; (*c*) detail of the preserved nasal and other postcranial remains.

dorsoventral thickening of the anterior process of the periotic; the dorsoventral compression of the pars cochlearis; the presence of a cochlear spine.

Identification beyond the genus was not possible because of the unusual nature of the preserved material. Most fossils of ziphiids are represented by cranial remains, mostly the rostral, prenarial and vertex region [15,25,35–37]. Mandibles, earbones and postcranial remains are more rarely preserved. Despite the unusual features present on the periotics (presence of a cochlear spine, depression along the medial surface of the posterior process) and the relatively large size of the specimen, the lack of cranial remains makes it nearly impossible to compare with many similar-sized ziphiids whose mandibles are not preserved (e.g. *Africanacetus*, *Globicetus*, *Tusciziphius*). Genus and sp. indet. NHMD 189993 possesses several derived crown Ziphiidae features (e.g. [16,17,26,36]) the dorsoventral thickening of the anterior process of the periotic bone, the dorsoventral compression of the pars cochlearis of the periotic, the short and unfused mandibular symphysis. However, a recent phylogenetic analysis proposed that some members of the more basal *Messapicetus* clade displayed derived characters indicative of a convergent evolution between stem and crown Ziphiidae [38]. Mandibles, earbones and postcranial material of the most derived members of this clade, *Globicetus*, *Tusciziphius* and *Imocetus* are not known [36]. It is, therefore, impossible to assess whether NHMD 189993 was a crown ziphiid or a member of the *Messapicetus* clade.

By a measure of caution, the advice of Barnes [39] and Fordyce and Muizon [40], which suggest that the identification of a new cetacean species should at least include skull and rostrum, is followed until more cranial material is available.

## 3.2. Description and comparisons

### 3.2.1. Cranium and mandible

*Overview and ontogeny*—NHMD 189993 is interpreted as an adult based on the complete fusion of the humeral head to the humeral shaft and the epiphyseal ankylosis of both epiphyses of the radius. In the porpoise *Phocoena phocoena*, extensive ankylosis of the postcranial skeleton characterizes adult specimens [41].

The most robust parts of the mandibles and postcranial elements of NHMD 189993 are well preserved compared to the more fragmentary cranial remains and ribs (figures 2 and 3). The earbones were found

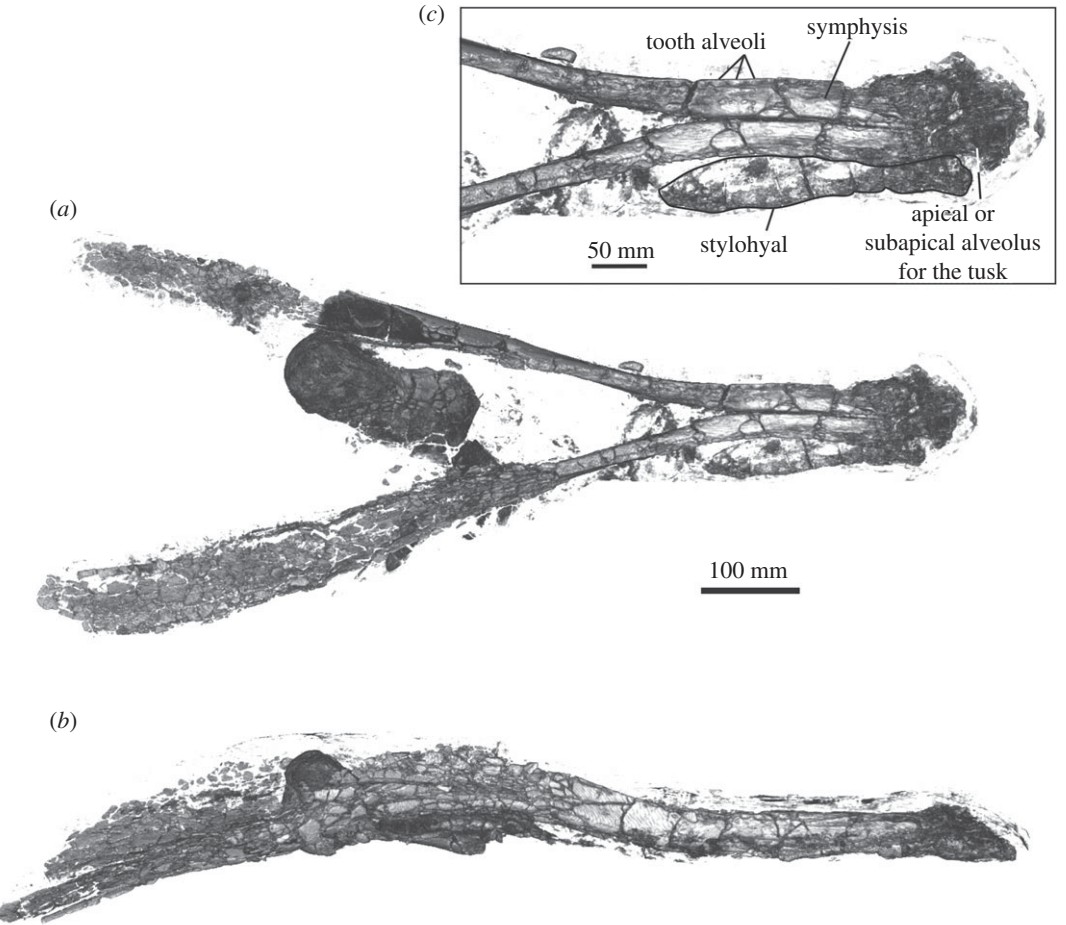

**Figure 3.** Digital reconstruction of the mandible and postcranial remains of NHMD 189993. (*a*) dorsal view; (*b*) lateral view; (*c*) detail of the anterodorsal part of the mandible.

close to the lower jaw, the right periotic still having the stapes firmly attached to it (figures 4 and 5; only right periotic illustrated). The humerus and radius were originally still articulated (figure 6*a*–*f*), whereas the stylohyal lay along the right lateral side of the symphysis (figure 6*o*–*q*). The teeth were collected out of their mandibular sockets around the bones (figure 6*g*–*n*). The preserved parts of the mandibles are 1032 mm long and 412 mm wide. More measurements of the specimen are available in tables 1 and 2.

*Nasal*—Because the nasal is the only piece identifiable from the shattered cranium of the specimen, its orientation is difficult to reveal. Unambiguous identification of a side is impossible. The dorsal exposure is flat and rectangular (figure 2*c*). The ratio between the width and length of the visible dorsal surface is 0.70 (60 mm long and 86 mm wide). No excavation is visible on the surface of the nasal bone.

*Periotic*—Measurements of the right periotic are available in table 2. The anterior process of the periotic is transversely thickened, with a large rounded protuberance along the dorsomedial surface of the periotic (figure 4*a*–*c*). Such strong lateromedial and dorsoventral thickening is observed in all crown ziphiids, but not in the stem ziphiids *D. mojnum*, *M. gregarius* and *N. platyrostris* [16–18]. In the latter, the thickening occurs only lateromedially. The tip of the anterior process is pointed. In ventral view, the anterior bullar facet is anteroposteriorly elongated and elliptical. Posteromedially to this facet, the accessory ossicle is still articulated in the fovea epitubaria (figure 4*d*). It extends along the dorsomedial margin of the anterior process. It is less rounded and developed than in *Berardius*, *Hyperoodon*, some species of *Mesoplodon* (*M. carlhubbsi*, *M. europaeus*, *M. grayi*, *M. mirus*), *Nazcacetus* and *Tasmacetus*. In ventral view, a sulcus extends anteroposteriorly along the accessory ossicle, and separates it in two portions (figure 4*d*). A similar sulcus is also observed in *Hyperoodon ampullatus*, *Mesoplodon densirostris* and *T. shepherdi*, although less developed in these species. The sulcus observed in NHMD 189993 could indicate the origin of the tendon of m. tensor tympani [31]. Posteriorly to the fovea epitubaria and the accessory ossicle, the mallear fossa develops along the medial margin of the

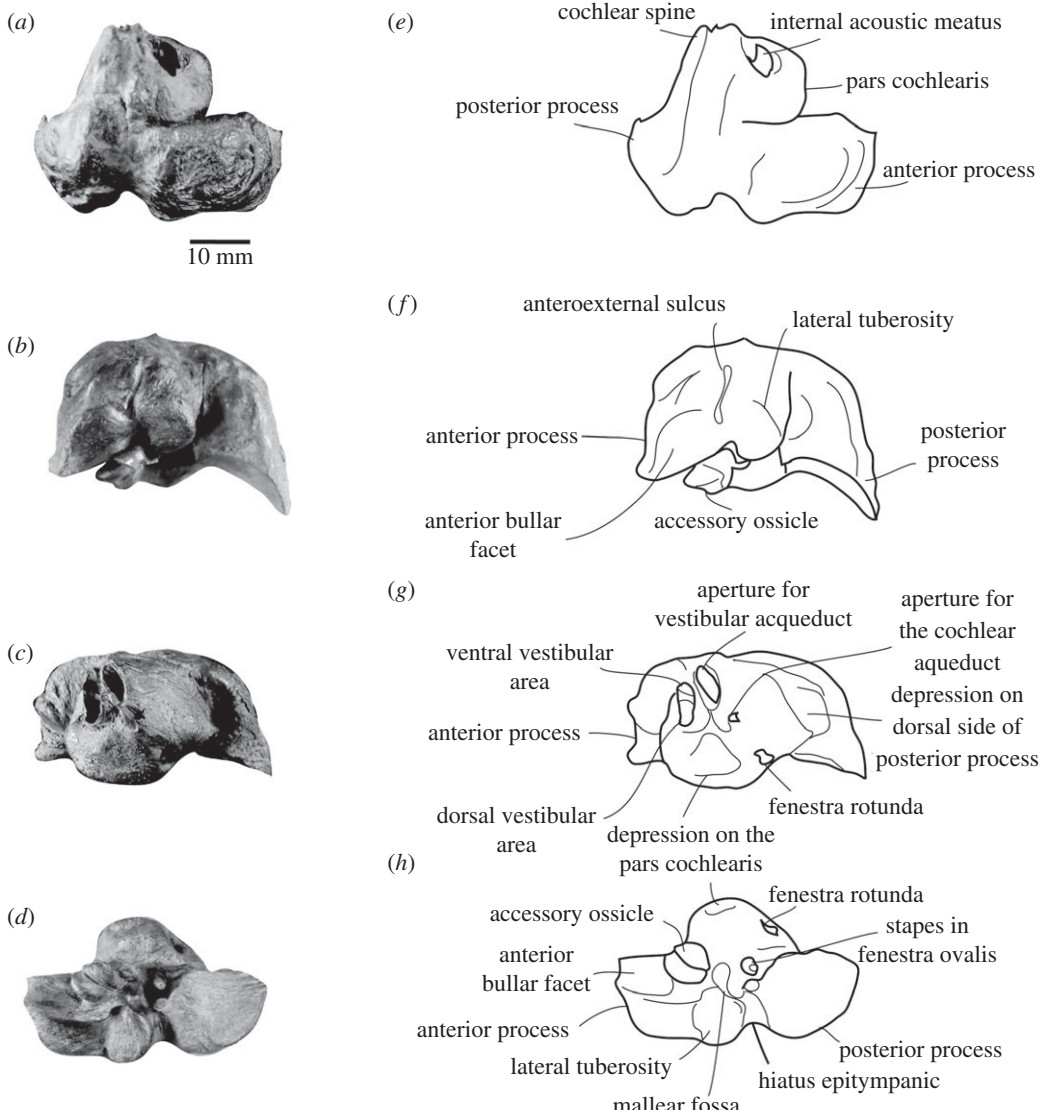

**Figure 4.** Right periotic of NHMD 189993. (*a*) dorsal view; (*b*) lateral view; (*c*) medial view; (*d*) ventral view; (*e*–*h*) corresponding drawings.

lateral tuberosity. In ventral view, the anterior process is separated from the lateral tuberosity by the anteroexternal sulcus, which can also be seen in lateral view. The anteroexternal sulcus is also present in the periotic of *D. mojnum*. In ventral view, the lateral tuberosity is lateromedially elongated, a character observed in all ziphiids, except *D. mojnum*, *M. gregarius* and *N. platyrostris* [16–18]. The fenestra ovalis is rounded. Posteroventrally to the fenestra ovalis, a deep hiatus epitympanicus separates the posterior process from the lateral tuberosity. In ventral view, the posterior process of the periotic is fan-shaped: it is rounded and widens abruptly posteriorly (figure 4*d*). A fan-shaped posterior bullar facet is a characteristic of all ziphiids, except *D. mojnum*, *N. platyrostris* and *M. gregarius* [16–18]. In medial view, the posterior process is oriented posteroventrally. NHMD 189993 lacks a distinct keel along the whole posterior process, a feature present in all ziphiids [17]. A deep depression excavates the anteromedial side of the posterior process, just posterior to the pars cochlearis (figure 4*c*). This depression seems unique to NHMD 189993 and was not observed among the ziphiids for which the periotic portion is known.

In dorsolateral view, the pars cochlearis is anteriorly shifted, a feature that distinguishes a ziphiid from an eurhinodelphid periotic (figure 4*a*) [16]. In ventromedial view, the pars cochlearis is rectangular, because of its straight anteromedial corner. It is also dorsoventrally compressed, a feature observed in crown ziphiids, but absent in *N. platyrostris* and members of the *Messapicetus* clade [17]. In ventral view, the pars cochlearis bears a triangular depression similar in shape to *D. mojnum* [18]. This depression is also visible in the species *Mesoplodon mirus* (USNM 504612, USNM 550351, USNM

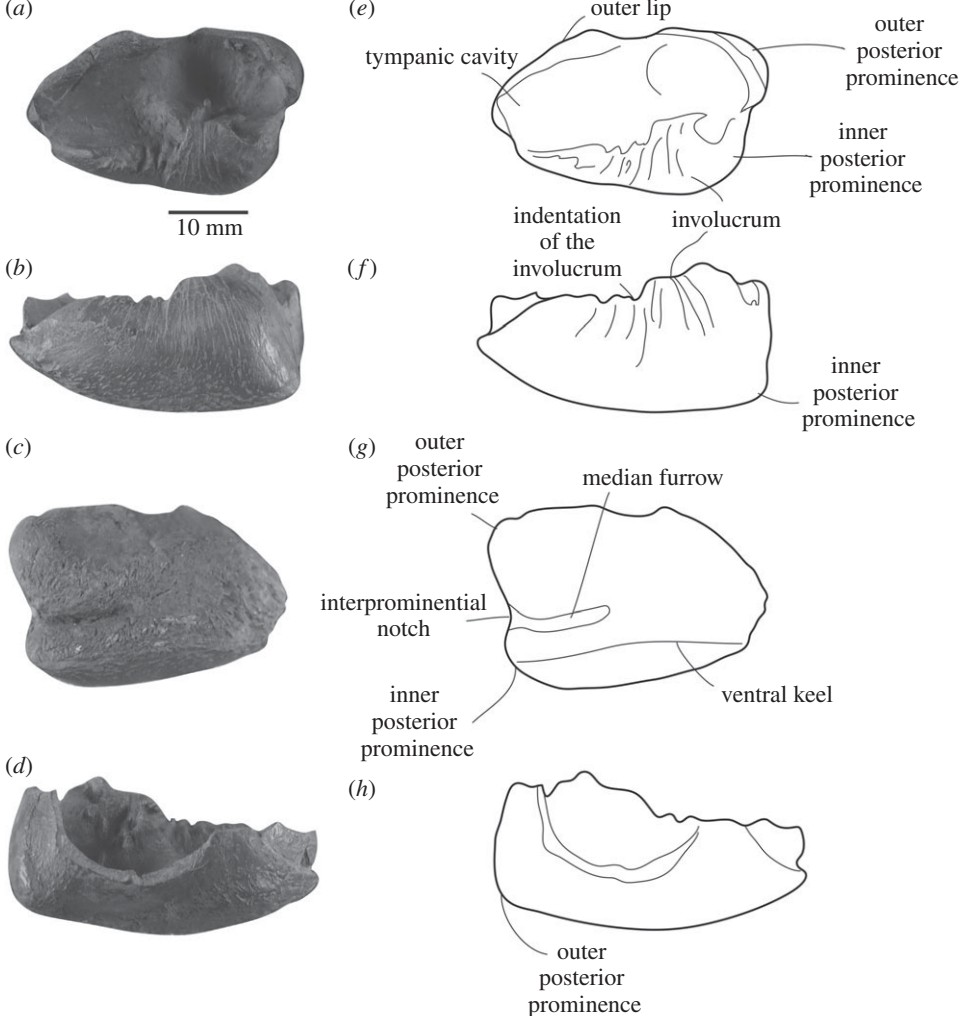

**Figure 5.** Right tympanic of NHMD 189993. (*a,e*) dorsal view; (*b,f*) medial view; (*c,g*) ventral view; (*d,h*) lateral view.

572961, USNM KLC112) and *M. bidens* (MNHN 1975.112, NHMD CN5x), but is elliptical and more elongated anteroposteriorly than in NHMD 189993. The tear-shaped fenestra rotunda is oriented posteroventrally. Posterordorsally to the internal acoustic meatus, the periotic bears a large cochlear spine. This unusual feature in Ziphiidae is present in *N. platyrostris* and *Berardius arnuxii* [17]. Its presence was also observed in *Berardius bairdii* (USNM 571524). The cochlear spine in NHMD 189993 is moderately developed dorsally, a condition similar to that of the genus *Berardius* and differing from the well-marked cochlear spine of *N. platyrostris*. In dorsomedial view, the internal acoustic meatus is elliptical; this feature is also observed in *N. platyrostris* and is connected to the presence of the cochlear spine. A thick crest separates the internal acoustic meatus from the aperture for the vestibular aqueduct (figure 4*c,g*). Inside the internal acoustic meatus, the dorsal vestibular meatus is separated from its ventral counterpart by a transverse crest. The ventral vestibular area occupies almost two-thirds of the surface of the internal acoustic meatus. Posterior to the vestibular area of the internal acoustic meatus, the aperture for the vestibular aqueduct is anteroposteriorly compressed. Ventrally to the vestibular aqueduct, the aperture for the cochlear aqueduct is reduced to a small opening (figure 4*c*).

*Tympanic bulla*—The right tympanic bulla is partially preserved (figure 5). Measurements are available in table 2. It lacks the base of the pedicle, the sigmoid process and the dorsal part of the outer lip. In ventral view, the bulla is heart-shaped, because of the interprominential notch well marked posteriorly that separates the inner and outer posterior prominences. In ventral view, the inner posterior prominence is compressed transversely. The outer posterior prominence is twice larger than the inner posterior prominence (figure 5*c*). The degree of compression of the inner posterior prominence recalls some species of *Mesoplodon* (e.g. *M. bidens* NHMD CN5x, *M. bowdoini* NMNZ MM2653, *M. europaeus* USNM 504349), *M. gregarius* and *N. platyrostris*. In *Hyperoodon* spp. and

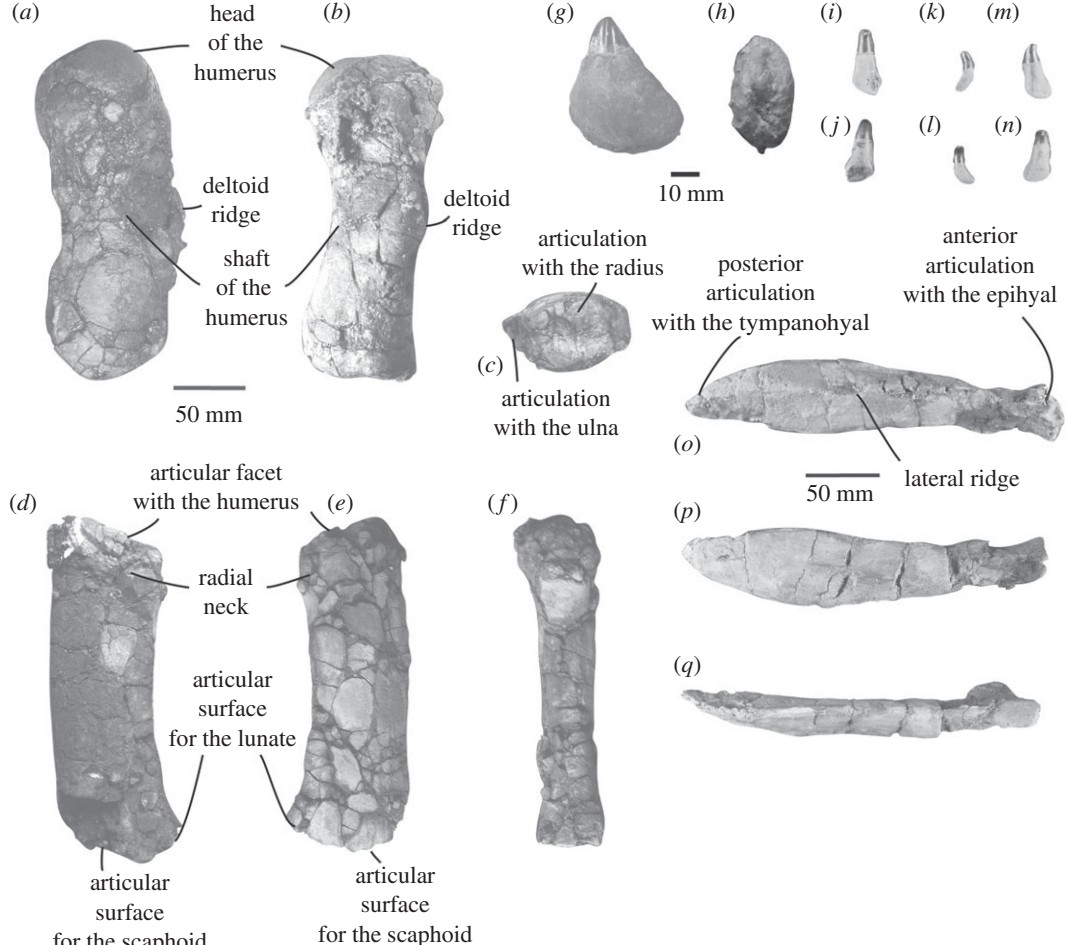

**Figure 6.** Postcranial elements of NHMD 189993. Right humerus in (*a*) lateral view; (*b*) dorsal view; (*c*) posterior view. Associated right radius in (*d*) lateral view; (*e*) medial view; (*f*) dorsal view; isolated tusk in (*g*) lateral view; (*h*) ventral view; three isolated teeth in (*i,k,m*) medial view; (*j,l,n*) lateral view; right stylohyal in (*o*) lateral view; (*p*) medial view; (*q*) dorsal view.

**Table 1.** Measurements of the mandible, cranial remains and forelimb bones of the specimen NHMD 189993. All measurements are in mm.

| feature | NHMD 189993 |
| --- | --- |
| mandibles | |
| anteroposterior length as preserved | 1032 |
| maximum posterior width as preserved | 412 |
| symphyseal portion anteroposterior length | 289 |
| symphyseal portion maximal transverse width | 61 |
| humerus | |
| anteroposterior humeral head diameter | 73 |
| maximal humeral length | 204 |
| maximal distal width | 80 |
| width at the level of the deltoid ridge | 92 |
| radius | |
| maximal radius length | 186 |
| width at mid-length | 63 |
| proximal width | 67 |
| distal width | 73 |

**Table 2.** Measurements of the periotic and the tympanic bone of NHMD 189993. All measurements are in mm and taken in ventral view unless noted otherwise.

| feature | NHMD 189993 |
| --- | --- |
| right periotic | |
| maximal anteroposterior length | 47 |
| maximal transverse width | 33 |
| pars cochlearis maximum anteroposterior length | 25 |
| pars cochlearis maximum transverse width | 27 |
| anterior process maximum anteroposterior length | 21 |
| anterior process maximum transverse width | 20 |
| posterior process maximum anteroposterior length | 18 |
| posterior process maximum transverse width | 21 |
| lateral tuberosity in lateral view transverse width | 10 |
| right tympanic bulla | |
| tympanic anteroposterior length | 43 |
| tympanic maximum transverse width | 26 |
| inner posterior prominence maximum transverse width | 11 |
| outer posterior prominence maximum width | 15 |
| dorsoventral height in lateral view as preserved | 21 |
| involucrum indentation on the tympanic | |
| dorsoventral height in medial view | 12 |

*Z. cavirostris*, the inner posterior prominence is more reduced and is even shorter posteriorly. In ventral view, the interprominential notch connects to the deep median furrow. The median furrow extends roughly until the first third of the bulla (figure 5*c*). The median furrow is more developed than in *Hyperoodon* spp. and *Z. cavirostris*, but less extended than in the stem ziphiids *D. mojnum*, *M. gregarius* and *N. platyrostris*. In ventral view, a keel extends along the whole anteroposterior length of the bulla.

The involucrum is indented, a feature visible in both dorsal and medial views (figure 5*a,b*). In medial view, the ventral part of the bulla is incurved, but does not reach the dorsalmost margin of the posterior portion of the involucrum, as in *D. mojnum* (figure 5*b*). The anterior margin of the tympanic bulla is too damaged to assess the degree of development of the tympanic spine, if present. However, the broken anterolateral margin of the bulla develops anteriorly into a thin bone plate (figure 5*a*), a condition similar to *N. platyrostris*, where the tympanic spine is absent [17].

*Stapes*—The right stapes is still firmly attached to the periotic in the fenestra ovalis and could not be removed (figure 4*c*). The stapes is conical, widening at its oval base, as observed in several ziphiids species [26]. The head of the stapes has a circular outline. The small and circular vestigial stapedial foramen opening is situated approximately at mid-length of the stapes. The muscular process is well developed and situated at the level of the head of the stapes.

*Mandible*—Both mandibles of NHMD 189993 lack the posterior part of the acoustic window and the mandibular condyle (figures 2 and 3). The symphyseal portion of the mandible is unfused (figure 3*c*). It is not ankylosed as in the long-snouted stem ziphiids *D. mojnum*, *Messapicetus* spp., *N. platyrostris* and genus and sp. indet. MUSM 3237 [16–18,38,42]. The symphysis is 289 mm long and represents at most 28% of the total length of the mandible (the total length of the preserved parts is 1032 mm). This value is much lower than in the long-snouted ziphiids *D. mojnum*, *Messapicetus* spp., *N. platyrostris* and in the extant species *T. shepherdi*, where the symphysis extends at least along 36% of the mandible total length [17]. The transverse section of the symphyseal portion of the mandibles NHMD 189993 is triangular, differing from the half-circled section of *Berardius* spp., *D. mojnum*, *Messapicetus* spp., *N. platyrostris*, *T. shepherdi* and MUSM 3237 [16–18,42]. The symphyseal portion of the mandible is turned upwards. This feature is also present in *H. ampullatus*, *M. bidens*, *M. grayi*, *M. mirus*, *N. urbinai* and *Z. cavirostris* [26]. The short unfused triangular symphysis of NHMD 189993 is close to *Chavinziphius maxillocristatus* whose mandible exhibits similar features [38].

The apex of the mandibles is heavily fractured, but the fragments preserved their original position, thus allowing an estimation of the original outline. In ventral view, the apex is rounded and probably possessed an enlarged alveolus for the tusk. This interpretation fits with the shape and size of the preserved tusk that is similar to those of several long-snouted stem beaked whales (*D. mojnum*, *Mess. gregarius*) possessing a pair of tusks in apical position (figure 6*g,h*). Furthermore, no other alveolus along the alveolar groove is sufficiently developed to support the tusk. The apex of the mandible is too fractured to identify precisely whether the tusk was positioned apically or subapically, as observed by Dalebout *et al.* [43] in *Mesoplodon perrini*. It is also possible that NHMD 189993 possessed two pairs of tusks, even though only one tusk is preserved with the specimen (figure 6*g,h*). This character is observed in *Berardius* spp., *Anoplonossa forcipata* and *D. mojnum* [18,44]. One mental foramen is visible along the lateral side of the mandible. It is elongated, well individualized and situated slightly posterior to the symphysis.

The outline of individualized alveoli can be distinguished in the alveolar border (figure 3*c*). The number of detected alveoli along the left dentary is 17, which is probably a slight underestimation, because of the eroded and fractured surface of the alveolar border of the most apical parts of the symphysis. It is not possible to assess the presence of a diastema between the tusk and the rest of the alveolar groove. The alveoli are oval, transversely compressed like in the *Messapicetus* spp. [15,16]. However, they are much more reduced than in the latter and much shallower compared to long-snouted stem ziphiids, *C. cristatus* and the species *T. shepherdii* [17,18,38]. In the three-dimensional reconstruction of the lower jaw, in lateral view, the position of the bone fragments posterior to the alveolar groove suggests the presence of a precoronoid crest. However, the dorsal surface of the acoustic window is too fractured to draw definitive conclusions of this issue. Further measurements are available in table 1.

*Teeth*—Along the mandibles of NHMD 189993, 14 isolated teeth were recovered (figure 6*g–n*). Their crown is approximately as developed as the root dorsoventrally and curves lingually. The crown progressively widens ventrally and projects posteroventrally. The section at the base of the crown is circular, whereas the transverse section of the root is more oval. An oval root is present in *Messapicetus* spp., unlike the circular section observed in *T. shepherdi* and the squared root observed in the species *D. mojnum* and *N. platyrostris*. A faint mesial keel is present in some of the smallest teeth of NHMD 189993. Despite the presence of individualized alveoli, the reduced size of the teeth and the particularly shallow alveoli suggest that the teeth of NHMD 189993 were not as robust as in other known toothed beaked whales, perhaps still embedded in the gum, as observed in some specimens of extant ziphiids (e.g. *Hyperoodon ampullatus*, *Mesoplodon grayi*; [45,46]).

An enlarged tooth interpreted as a tusk was also found. This tooth is more massive than the other reduced teeth (figure 6*g,h*). The tusk is triangular, with a root more developed dorsoventrally than the crown. The root of the tooth is transversely compressed with an oval outline (figure 6*h*). As suggested by the outline of the apex of the mandible, the tusk most likely fitted in apical or subapical position on the mandible. The slightly rounded tip of the crown also suggests that the tusk is slightly worn and as such was originally erupted. The tusk resembles those of long-snouted ziphiids, *Berardius* spp. and *T. shepherdii* due to their transverse compression. It differs from the apical tusk present in males *H. ampullatus* and *Z. cavirostris,* which is more conical. It also differs from the genus *Mesoplodon* where the tusk is heavily compressed transversely, even in species in which the tusk is in apical or subapical position (*M. mirus* USNM 504612; *M. hectori* NMNZ MM0002901; *M. perrini* USNM 504260).

### 3.2.2. Postcranial elements

*Hyoid apparatus*—The right stylohyal is 238 mm long, 48 mm wide and 26 mm thick (figure 6*o–q*). The length of this bone is almost twice longer than in *Mesoplodon layardi* (NMNZ 1899: 109 mm; NMNZ 2917: 166 mm). The stylohyal length of NHMD 189993 resembles more the one observed in *Hyperoodon planifrons* (NMNZ 1806: 272 mm; NMNZ DM 1878: 246 mm) and *Ziphius cavirostris* (NHMD CN1: 248 mm). The ratio between length and width of the stylohyoid of NHMD 189993 is closer to *M. gregarius* than *N. urbinai* (4.96 in NHMD 189993; 4.63 in *M. gregarius;* 4.10 in *N. urbinai*). This suggests that the stylohyal of *N. urbinai* is wider than long when compared with NHMD 189993 and *M. gregarius*. However, the stylohyal of NHMD 189993 is significantly thicker than that of *M. gregarius*: the ratio between width and thickness is 1.84 in NHMD 189993, whereas it is 1.39 in *M. gregarius*.

A constriction is present on the most anterior part of the stylohyal, at the level of the articulation with the epihyal (figure 6*o*). This constriction is observed in the species *Berardius arnuxii* (MNHN A3244) and *T. shepherdii* (MM 2908). In lateral view, the stylohyal progressively widens from anterior to posterior and reaches its maximum transverse width in the posterior part of the bone. The posterior margin of the bone

that articulates with the tympanohyal is pointed (figure 6*o*). The shape of the stylohyal of NHMD 189993 resembles those of *Z. cavirostris* and *H. planifrons*, even though in those species, the transverse widening is more pronounced (between 29% and 68% wider), with a flatter dorsal surface. This shape is also observed in *Mesoplodon europaeus* [34]. It differs from the stylohyal observed in several other species of *Mesoplodon* examined (*M. bidens* MNHN 1963-259, MNHN 1963-111; *M. europaeus*, NMNZ 550390; *M. layardii* NMNZ 2917), where the lateral and medial margins of the bone are straight, without transverse widening. A ridge runs along the lateral side of the stylohyal of NHMD 189993 (figure 6*o*), as observed in the ziphiids *M. europaeus* and *M. mirus* [34]. This ridge gives a triangular transverse section to the bone.

*Humerus*—The right humerus is fully preserved (figure 6*a–c*). Further measurements are available in table 1. It is 204 mm long and 92 mm wide at the level of the deltoid ridge. The ratio between the humeral length and the estimated bizygomatic width (or posterior width of the mandible as used in the specimen) is similar to *M. gregarius* (in *M. gregarius*: 0.48; in NHMD 189993: 0.50). Both species display a proportionally longer humerus than most extant ziphiids [47]. The head of the humerus is hemispherical. In lateral view (figure 6*a*), the humeral head represents a quarter of the total length of the humerus. In *M. gregarius*, the head is more prominent and anterolaterally oriented: it represents almost a third of the total length of the humerus. In lateral view, the deltoid ridge is well developed along the anterior margin of the humerus (figure 6*a*). It develops approximately at mid-length of the humerus, and over a third of its length. The presence of a developed deltoid ridge is a characteristic of extant Ziphiidae, even though not as much developed as in Physeteridae [48]. The posterior part of the humerus of NHMD 189993 does not widen, thus differing from the condition observed in many odontocetes [48]. This feature is a characteristic of the ziphiid humeri [48]. In posterior view, the articular facets for the radius and the ulna are well separated by a crest (figure 6*c*). Each facet occupies approximately half of the posterior surface of the humerus.

*Radius*—The associated right radius was originally found articulated with the humerus (figure 6*d–f*). The radius curves anteroposteriorly. It measures 186 mm long, 63 mm wide at mid-length. The facet for articulation of the humerus is oriented anterodorsally (figure 6*d–e*). Posteriorly, the articulations for the scaphoid and the lunate are well defined; they occupy approximately half of the posterior width of the radius. The articulation for the scaphoid is straight in lateral view, whereas the articulation for the lunate is more oblique and faces posterodorsally. In all Ziphiidae, and differing from other odontocetes, the posterior part of the radius is not widened [48]. The overall shape of the radius does not significantly differ from extant ziphiids examined (e.g. *Berardius arnuxii* NMNZ 415, MNHN A3244; *Mesoplodon layardii* NMNZ 2917; *T. shepherdii* NMNZ MM 2908; *Z. cavirostris* NHMD CN1). However, its radius is wider than in *M. gregarius*, in which the ratio between the length and the width of radius is 0.25 (versus 0.34 in NHMD 189993). Further measurements are available in table 2.

*Ribs*—two partial ribs of NHMD 189993 are preserved (figure 2*c*). Their body is heavily fractured and fragmented. Judging from the similar outline of each rib, they were probably from the same pair. They are tentatively inferred to be the pair 2, because of their thick, yet flattened body. Both are double-headed with a marked neck separating the capitulum from the tuberculum.

### 3.2.3. Size estimates of the specimen

Condylobasal length and bizygomatic width were strongly correlated across the dataset ($R^2 = 0.78$; figure 7). The combination of the two linear measurements was sufficient to separate the four size categories ($p$-value < 0.0001), and each size category was well distinguishable.

The four size categories were better separated using the bizygomatic width, particularly in the case of the medium-sized and large-sized ziphiids. Species from these two categories displayed a similar range of variation in condylobasal length due to the strong variability of the anteroposterior length of the rostrum in those species. For example, the large-sized ziphiid *Z. cavirostris* displayed a condylobasal length similar to other medium-sized ziphiids, such as *M. mirus* and *M. europaeus* (figure 7).

In contrast, the long-snouted medium-sized ziphiids *M. gregarius*, *N. platyrostris*, *D. mojnum* and *M. grayi* displayed condylobasal length matching some large-sized ziphiids (e.g. *M. layardii*). The long-snouted stem ziphiids were well separated from other species of their size category, including *Mesoplodon grayi*. The latter also display a strong elongation of the rostrum, but are also characterized by a smaller bizygomatic width. Small-sized ziphiids were easily distinguished from other size categories. They consist of the living *Mesoplodon peruvianus* and the fossil *N. urbinai*.

Based on the estimate of the bizygomatic width and condylobasal length, NHMD 189993 would be a large ziphiid, between 5.5 and 7.5 m. Its condylobasal length and bizygomatic width is within the range

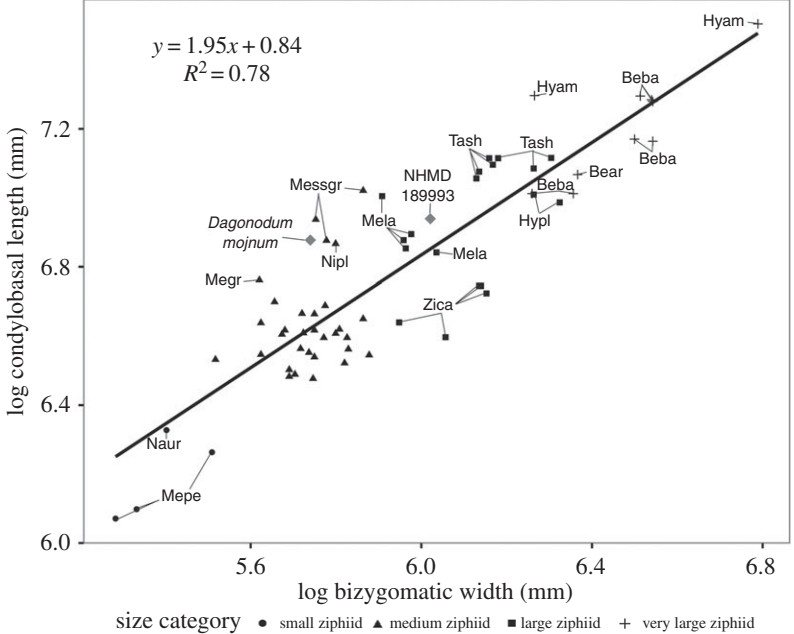

$$y = 1.95x + 0.84$$
$$R^2 = 0.78$$

**Figure 7.** Log-transformed bizygomatic width plotted against condylobasal length in extinct and extant Ziphiidae. Bear, *Berardius arnuxii*; Beba, *Berardius bairdii*; Hyam, *Hyperoodon ampullatus*; Hypl, *Hyperoodon planifrons*; Megr, *Mesoplodon grayi*; Mela, *M. layardii*; Mepe, *M. peruvianus*; Messgr, *Messapicetus gregarius*; Naur, *Nazcacetus urbinai*; Nipl, *Ninoziphius platyrostris*; Tash, *Tasmacetus shepherdi*; Zica, *Ziphius cavirostris*. The cluster of medium ziphiids corresponds to the species *M. bidens*, *M. europaeus*, *M. ginkgodens* and *M. mirus*.

of the extant Ziphiidae *M. layardii* whose size generally ranges between 5.5 and 6 m [49]. Based on the similar condylobasal length, the degree of elongation of the rostrum in NHMD 189993 is probably more similar to *M. layardii* than to the shorter rostrum of *Z. cavirostris* or the extremely elongated rostrum of long-snouted stem ziphiids. The size of NHMD 189993 clearly differs from that of the other fossil ziphiid found in the Gram Formation, *D. mojnum*, a medium-sized ziphiid whose size probably ranged between 4 and 4.5 m.

# 4. Discussion

## 4.1. Suction feeding

All extant beaked whales are specialized suction feeders: they generate powerful suction pressures with their tongue acting like a piston, to capture and engulf their prey [4]. Many odontocetes can use suction feeding for capturing and/or transporting the prey [50], but extant Ziphiidae are obligate suction feeders, due to the absence of functional teeth to capture their prey [4]. Furthermore, they exhibit a lateral closure of the intraoral cavity combined with a wider and thicker hyoid apparatus compared to odontocetes relying on a more raptorial feeding strategy [4]. The only extant ziphiid species that is perhaps not an obligate suction feeder is *T. shepherdi*. Unlike other ziphiids, this species retains a set of erupted teeth likely functional [51]. Based on one stomach content mostly consisting of the fish species *Merluccius hubbsi*, MacLeod *et al.* [13] speculated that this species may be specialized in feeding on deep-water fish rather than cephalopods, thus limiting the competition with other species of beaked whales in the southern oceans where it occurs.

Several lines of evidence suggest that NHMD 189993 was capable of using suction feeding to a larger extent than other toothed ziphiids. First, the stylohyal is strongly thickened and elongated. Its anteroposterior length is similar to that of the large ziphiids *H. planifrons* and *Z. cavirostris*, whereas it is almost twice longer than in *M. layardii*, a species close in body size to NHMD 189993. A thickened stylohyal is necessary to support strong tongue muscles. The styloglossus and the hyoglossus are the two main muscles responsible for the retraction of the tongue in a piston-like manner during suction [34]. The styloglossus originates on the lateral surface of the stylohyal, which is particularly thickened in the ziphiids *M. mirus* and *M. europaeus*. Both species also possess a strong styloglossus: Reidenberg

and Laitman [34] noted that these species possessed the largest styloglossus relative to total body length from their sample. The thickening of the stylohyal is associated with the development of a ridge along the lateral surface of the bone giving the stylohyal a triangular shape in transverse view [34]. This ridge was observed in several other ziphiid species (e.g. *Hyperoodon* spp., *Berardius* spp., *Mesoplodon bidens*, *M. layardii*, *Z. cavirostris*) and seems a characteristic of the ziphiids species specialized to suction feeding. The same ridge is present in NHMD 189993.

The thickening of the stylohyal is accompanied by a reduction of the teeth in NHMD 189993. On each dentary, the specimen possessed at least 17 alveoli, a number similar to *T. shepherdi* (18–28) [51], but largely inferior to long-snouted stem ziphiids (*D. mojnum*, 29; *M. gregarius*, 25–26; *N. platyrostris*, 40–42) [16–18] or the fossil ziphiid *Chavinziphius maxillocristatus* (at least 50) [38]. Additionally, several features of the teeth and the alveoli differ between the aforementioned species and NHMD 189993. The alveoli of the latter, although individualized, are particularly shallow, greatly differing from the condition observed in other toothed beaked whales where the alveolar groove is deep, but the septa are not necessarily well differentiated (e.g. *M. gregarius*) [16]. The teeth themselves are also reduced in size compared to other toothed ziphiids including *T. shepherdi*. With the exclusion of the tusk, the longest tooth of NHMD 189993 measures 20 mm with a maximum diameter of 11 mm. The tooth measurements of the specimen are even smaller than in the medium-sized ziphiids *D. mojnum*, *M. gregarius* and *N. platyrostris*. Furthermore, in ziphiids with functional teeth, the robust crown shows apical wear or interlocking facets, suggesting that their dentition was functional. This is not the case in NHMD 189993 where the small crown does not show the sign of interlocking. In many teeth, the apex is broken off and does not allow for an estimation of the degree of apical wear, but the few teeth with a preserved apex do not show signs of wear. Therefore, we hypothesize that the small teeth of NHMD 189993 were either embedded within the gum or too small to be used ordinarily for capturing the prey. Nevertheless, we do not discard the possibility that NHMD 189993 could have occasionally used its reduced teeth (if erupted) to manipulate or capture some of its prey.

## 4.2. A potential case of niche separation

Morphological evidence and the different size estimates of *D. mojnum* and NHMD 189993 suggest that these two species occupied two different ecological niches. Despite the relatively inaccurate age estimation of the Gram Formation (7.2–9.9 Ma), the study of the variation of accumulation rates suggests that sediments were deposited during approximately 120 000 years [25]. Therefore, *D. mojnum* and NHMD 189993, both found in the Gram claypit, were probably coevals. The mollusc faunae found in association with NHMD 189993 indicates that it was found in the biozone V (the uppermost part of the Gram Formation), whereas *D. mojnum* was found in the biozone III, IV or V [18]. Since Beyer [25] observed a significant increase in the uppermost 8 m of the formation, there is a possibility that the two specimens were not separated from more than 20 000 years. Assuming that the two species were contemporary, the co-occurrence of two different sized species of Ziphiidae at the same location suggests a case of niche separation.

Cases of niche separation are known in extant ziphiids: *Mesoplodon* species consistently feed on smaller prey type (generally, cephalopods under 500 g) compared to *Hyperoodon* and *Ziphius* species (cephalopods over 1 kg) [13]. The difference of prey size targeted may explain why species of *Mesoplodon* are often sympatric with the latter [52–55]. Size is not the only component, even though an important one [12], allowing niche separation between ziphiid species. In the case of *D. mojnum* and NHMD 189993, the difference in specialization to suction feeding reinforces this hypothesis. The species *D. mojnum* possesses some adaptations to suction feeding (transverse thickening of the basyhyal and thyrohyal; presence of a precoronoid crest) [18], but not to the extent of NHMD 189993 that probably relied more prominently on this feeding strategy. Obligate suction feeders with a reduced tooth count are often more teuthophagous [4,56], even though some ziphiids can still feed on fish [13]. Perhaps, the more specialized oral apparatus of NHMD 189993 is more indicative of a more predominantly teuthophagous diet than *D. mojnum*. Interestingly, the fossil of a cuttlefish (Sepiida) was found in the Gram Formation (MSM DK718; unpublished data), a possible prey type for NHMD 189993.

Other cases of niche separation between fossil ziphiids probably occurred at other locations where they show a diversity of sizes or feeding strategies. Bianucci *et al.* [57] already proposed this interpretation to explain the high diversity of fossil ziphiids trawled from the sea floor off South Africa. Fossil ziphiids from the Neogene of Antwerp and fished from the Atlantic Ocean floor off the Iberian Peninsula also exhibit a great range of skull sizes, which could be indicative of ecological

niche segregation [35,37]. In the absence of precise datation for these three localities, it is unclear whether the different species were living during the same time span.

# 5. Conclusion

Despite the rich fossil record of beaked whales, the discovery of postcranial material still represents a rare finding [18,42,58]. A new fossil of Ziphiidae, NHMD 189993, consisting of the mandible, earbones, the stylohyal, isolated teeth including the tusk, the right humerus and associated radius is described here. This fossil is dated to the mid- to late Tortonian (ca 9.9–7.2 Ma). Despite the lack of cranial material for comparing with other similarly sized fossil ziphiids, NHMD 189993 (here referred to Ziphiidae gen. and sp. indet.) clearly differs from the other species known from the Gram Formation, *D. mojnum*.

Unlike *D. mojnum* and other long-snouted stem ziphiids, the morphology of the oral apparatus of NHMD 189993 suggests that it was well adapted for suction feeding. The reduced teeth were possibly still embedded in the gum, and morphological features of the thickened stylohyal support this interpretation.

The two fossil species *D. mojnum* and NHMD 189993 probably occupied different ecological niches with NHMD 189993 relying on evasive prey such as cephalopods. Assuming that *D. mojnum* and NHMD 189993 were chronologically concomitant, the spatial co-occurrence of these two species can be illustrative of a case of niche separation. Together with sexual dimorphism [59], the specialization toward specific ecological niches in Ziphiidae may partly explain the rich specific diversity of this family.

Data accessibility. The datasets supporting this article are available as part of the electronic supplementary material from Figshare: https://figshare.com/s/c75c55bdca6ef3ae3a6f.

Authors' contributions. B.R. identified and interpreted the ziphiid remains. H.L. created the digital reconstruction and the associated figures. B.R. made calculations for the size estimates and collected the measurements. Both authors participated in the preparation of the illustrations and the writing of the manuscript. Both discussed the results and commented on the manuscript at all stages.

Competing interests. We have no competing interests.

Funding. This project was funded by the Dansk Slots- og Kulturstyrelsen (grant no. FORM.2016-0021).

Acknowledgements. The authors thank Mette Steeman for her suggestions and support for the completion of the project. They are also indebted to Frank Osbæk and Trine Sørensen who were in charge of the preparation of the specimen. They thank the following colleagues for kindly allowing them to access some of the comparative material used in this study: Morten Tange Olsen and Daniel Klingberg Johansson to access the NHMD collection, Anne-Lise Folie for the IRSNB collection, Christian de Muizon and Christine Lefèvre for the MNHN collection, Chiara Sorbini for the MSNUP collection, Thomas Schultz for the NMNZ and Charles Potter for the USNM collection. The authors are also grateful for the useful feedback of the associate editor, Giovanni Bianucci and of an anonymous reviewer.

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
