## [Reviewer comments · Royal Society Open Science]

Review History

RSOS-191347.R0 (Original submission)

Review form: Reviewer 1

Is the manuscript scientifically sound in its present form?

Yes

Are the interpretations and conclusions justified by the results?

Yes

Is the language acceptable?

Yes

Do you have any ethical concerns with this paper?

No

Have you any concerns about statistical analyses in this paper?

No

Recommendation?

Accept with minor revision (please list in comments)

Comments to the Author(s)

Referee's comments to

"A new specimen of Ziphiidae (Cetacea, Odontoceti) from the late Miocene of Denmark with morphological evidences for suction feeding behaviour"

by Benjamin Ramassamy and Henrik Lauridsen.

Dear Editor,

I read over the above article, which was sent to me for review. I think that this tight paper represents an interesting addition to the current knowledge about the fossil history and palaeobiology of beaked whales ziphiids that merits publication on Royal Society Open Science. Notably, it shows that even fossil specimens whose preservation state is not extraordinary can be unexpectedly informative when modern techniques of imaging and "traditional" systematic palaeontology and functional anatomy join in an integrate research effort. At the same time, I find that some aspects of the article should be reappraised and reconsidered before publication:

1) I would ask the authors to provide as many details as possible about the geological, chronostratigraphic and palaeoecological framework of the finding site, possibly via the addition of a devoted "Geological and palaeoenvironmental setting" chapter. This would adequately support the palaeoecological themes addressed in the Discussion. Otherwise, I have no prime suggestions for improving the scientific quality of the study, which is good from what I can judge.

2) Overall, the manuscript is sufficiently well organized and presented. Nevertheless, the English text could be improved here and there - I attach an annotated version of the main text which features many proposals of rephrasing and minor modifications (take present that I am not a native English speaker anyway). Moreover, several additional comments and suggestions are provided therein - please take a look at them also when reconsidering your manuscript. For summarizing, I support publication of the above manuscript after proper minor revision (see Appendix A).

Review form: Reviewer 2 (Giovanni Bianucci)

Is the manuscript scientifically sound in its present form?

Yes

Are the interpretations and conclusions justified by the results?

Yes

Is the language acceptable?

Yes

Do you have any ethical concerns with this paper?

No

Have you any concerns about statistical analyses in this paper?

No

Recommendation?

Accept with minor revision (please list in comments)

Comments to the Author(s)

The manuscript of Ramassamy and Lauridsen describes new fragmentary ziphiid remains from the upper Miocene of Denmark. The fossil is not very exciting, but with the help of the CT scan and including an intriguing paleoecological analysis, the paper appears overall good and, in my opinion, deserves to be published.

It is well organized and figures and tables are exhaustive.

I just recommend to revise the style: sometimes the sentences could be simplified (e.g. 'the species *M. bidens*' could be abbreviate in '*M. bidens*'; moreover it is not necessary to report 'genus and sp. indet.' all the time that NHMD 189993 is cited). Several similar small corrections are reported in an annotated copy of the manuscript (Appendix B).

Here three significant suggestions:

- The stylohyal of NHMD 189993 should be compared also with the stylohyals of *Nazcaetus urbanai*, *Messapicetus gregarius*, and *Tusciziphius crispus*. It is important considering that this bone is a relevant element of the feeding apparatus.
- The mandible and the related dentition of *Chavinziphius* could be also considered, having small and numerous teeth as NHMD 189993.
- From Peru there is another fragmentary ziphiids (MUSM 3237) having partial mandibles and ear bones preserved: perhaps it could be compared with NHMD 189993, also considering that the Peruvian specimen was dated 9-8,5 Ma.

Finally, perhaps worth mentioning somewhere the late Miocene fossil of *Messapicetus* from Menorca, imaged by means of CT, reported in Bianucci et al., APP, 2019, <https://doi.org/10.4202/app.00593.2019>. Just a suggestion (as I am one of the authors...).

Giovanni Bianucci

Decision letter (RSOS-191347.R0)

16-Sep-2019

Dear Dr Ramassamy

On behalf of the Editors, I am pleased to inform you that your Manuscript RSOS-191347 entitled "A new specimen of Ziphiidae (Cetacea, Odontoceti) from the late Miocene of Denmark with morphological evidences for suction feeding behaviour" has been accepted for publication in Royal Society Open Science subject to minor revision in accordance with the referee suggestions. Please find the referees' comments at the end of this email.

The reviewers and handling editors have recommended publication, but also suggest some minor revisions to your manuscript. Therefore, I invite you to respond to the comments and revise your manuscript.

- Ethics statement

- Data accessibility

<http://datadryad.org/submit?journalID=RSOS&manu=RSOS-191347>

- Competing interests

- Authors' contributions

- Acknowledgements

- Funding statement

Please ensure you have prepared your revision in accordance with the guidance at <https://royalsociety.org/journals/authors/author-guidelines/> -- please note that we cannot publish your manuscript without the end statements. We have included a screenshot example of

the end statements for reference. If you feel that a given heading is not relevant to your paper, please nevertheless include the heading and explicitly state that it is not relevant to your work.

Because the schedule for publication is very tight, it is a condition of publication that you submit the revised version of your manuscript before 25-Sep-2019. Please note that the revision deadline will expire at 00.00am on this date. If you do not think you will be able to meet this date please let me know immediately.

Please note that Royal Society Open Science charge article processing charges for all new submissions that are accepted for publication. Charges will also apply to papers transferred to Royal Society Open Science from other Royal Society Publishing journals, as well as papers

submitted as part of our collaboration with the Royal Society of Chemistry (<http://rsos.royalsocietypublishing.org/chemistry>).

on behalf of Kevin Padian (Subject Editor)
openscience@royalsociety.org

Associate Editor Comments to Author:

Associate Editor: 1

Comments to the Author:

The two reviewers consider your paper largely ready for acceptance; however, a number of minor matters require attention - including revisiting the written English. You might consider utilising a resource such as <https://royalsociety.org/journals/authors/language-polishing/> or a colleague who is a native speaker of English to assist.

Reviewer comments to Author:

Reviewer: 1

Comments to the Author(s)

Referee's comments to

"A new specimen of Ziphiidae (Cetacea, Odontoceti) from the late Miocene of Denmark with morphological evidences for suction feeding behaviour"
by Benjamin Ramassamy and Henrik Lauridsen.

Dear Editor,

I read over the above article, which was sent to me for review. I think that this tight paper represents an interesting addition to the current knowledge about the fossil history and palaeobiology of beaked whales ziphiids that merits publication on Royal Society Open Science. Notably, it shows that even fossil specimens whose preservation state is not extraordinary can be unexpectedly informative when modern techniques of imaging and "traditional" systematic palaeontology and functional anatomy join in an integrate research effort. At the same time, I find that some aspects of the article should be reappraised and reconsidered before publication:

- 1) I would ask the authors to provide as many details as possible about the geological, chronostratigraphic and palaeoecological framework of the finding site, possibly via the addition of a devoted "Geological and palaeoenvironmental setting" chapter. This would adequately

support the palaeoecological themes addressed in the Discussion. Otherwise, I have no prime suggestions for improving the scientific quality of the study, which is good from what I can judge.

2) Overall, the manuscript is sufficiently well organized and presented. Nevertheless, the English text could be improved here and there - I attach an annotated version of the main text which features many proposals of rephrasing and minor modifications (take present that I am not a native English speaker anyway). Moreover, several additional comments and suggestions are provided therein - please take a look at them also when reconsidering your manuscript. For summarizing, I support publication of the above manuscript after proper minor revision.

Reviewer: 2

Comments to the Author(s)

The manuscript of Ramassamy and Lauridsen describes new fragmentary ziphiid remains from the upper Miocene of Denmark. The fossil is not very exciting, but with the help of the CT scan and including an intriguing paleoecological analysis, the paper appears overall good and, in my opinion, deserves to be published.

It is well organized and figures and tables are exhaustive.

I just recommend to revise the style: sometimes the sentences could be simplified (e.g. 'the species *M. bidens*' could be abbreviate in '*M. bidens*'; moreover it is not necessary to report 'genus and sp. indet.' all the time that NHMD 189993 is cited). Several similar small corrections are reported in an annotated copy of the manuscript.

Here three significant suggestions:

- The stylohyal of NHMD 189993 should be compared also with the stylohyals of *Nazcaetus urbinai*, *Messapicetus gregarius*, and *Tusciziphius crispus*. It is important considering that this bone is a relevant element of the feeding apparatus.
- The mandible and the related dentition of *Chavinziphius* could be also considered, having small and numerous teeth as NHMD 189993.
- From Peru there is another fragmentary ziphiids (MUSM 3237) having partial mandibles and ear bones preserved: perhaps it could be compared with NHMD 189993, also considering that the Peruvian specimen was dated 9-8,5 Ma.

Finally, perhaps worth mentioning somewhere the late Miocene fossil of *Messapicetus* from Menorca, imaged by means of CT, reported in Bianucci et al., APP, 2019, <https://doi.org/10.4202/app.00593.2019>. Just a suggestion (as I am one of the authors...).

Giovanni Bianucci

Author's Response to Decision Letter for (RSOS-191347.R0)

See Appendix C.

Decision letter (RSOS-191347.R1)

04-Oct-2019

Dear Dr Ramassamy,

I am pleased to inform you that your manuscript entitled "A new specimen of Ziphiidae (Cetacea, Odontoceti) from the late Miocene of Denmark with morphological evidences for suction feeding behaviour" is now accepted for publication in Royal Society Open Science.

on behalf of Prof Kevin Padian (Subject Editor)
openscience@royalsociety.org

Appendix A**ROYAL SOCIETY
OPEN SCIENCE****A new specimen of Ziphiidae (Cetacea, Odontoceti) from the
late Miocene of Denmark with morphological evidences for
suction feeding behaviour**

Journal:	Royal Society Open Science
Manuscript ID	RSOS-191347
Article Type:	Research
Date Submitted by the Author:	03-Aug-2019
Complete List of Authors:	Ramassamy, Benjamin; Museum of Southern Jutland, Department of Natural History and Palaeontology; Lauridsen, Henrik; Aarhus University Hospital Skejby, The Department of Clinical Medicine, Comparative Medicine Lab
Subject:	palaeontology < BIOLOGY, taxonomy and systematics < BIOLOGY, evolution < BIOLOGY
Keywords:	Feeding Strategy, Ecological niche, Systematics, Ziphiidae
Subject Category:	Biology (whole organism)

Author-supplied statements

Relevant information will appear here if provided.

Ethics

Does your article include research that required ethical approval or permits?:

This article does not present research with ethical considerations

Statement (if applicable):

CUST_IF_YES_ETHICS :No data available.

Data

It is a condition of publication that data, code and materials supporting your paper are made publicly available. Does your paper present new data?:

Yes

Statement (if applicable):

CUST_IF_YES_DATA :No data available.

Conflict of interest

I/We declare we have no competing interests

Statement (if applicable):

CUST_STATE_CONFLICT :No data available.

Authors' contributions

This paper has multiple authors and our individual contributions were as below

Statement (if applicable):

B.R. identified and interpreted the ziphiid remains. H.L. created the digital reconstruction and the associated figures. B.R. made calculations for the size estimates and collected the measurements. Both authors participated in the preparation of the illustrations and the writing of the manuscript. Both discussed the results and commented on the manuscript at all stages.

A new specimen of Ziphiidae (Cetacea, Odontoceti) from the late Miocene of Denmark with morphological evidences for suction feeding behaviour

BENJAMIN RAMASSAMY*,¹ and HENRIK LAURIDSEN²

¹Department of Natural History and Palaeontology, the Museum of Southern Jutland, Lergravsvej 2, Gram, 6510, Denmark;

²Comparative Medicine Lab, Department of Clinical Medicine, Aarhus University, Palle Juul-Jensens Boulevard 99, Aarhus, 8200, Denmark;

Keywords: Feeding Strategy; Ecological niche; Systematics; Ziphiidae

1. Summary

A new fossil of Ziphiidae from the upper Miocene Gram Formation (ca. 9.9-7.2 Ma) is described. Computed Tomographic scanning of the specimen was performed to visualize the mandibles and to obtain a digital reconstruction. It possesses several derived characters in the Ziphiidae, such as the dorsoventral thickening of the anterior process of the periotic, the dorsoventral compression of the pars cochlearis, and the short unfused symphysis. The specimen cannot be identified beyond family level, because of the unusual nature of the preserved parts consisting in mandibles, earbones and postcranial remains. It differs from the other species of ziphiid from the Gram Formation, *Dagonodum mojnium*, based on its larger size and the more derived morphology of its mandibles and earbones. Its long and thickened stylohyal combined with its teeth reduction suggests that this new specimen relied primarily on suction feeding. By contrast, the other ziphiid species from the Gram Formation, *D. mojnium*, was adapted to a more raptorial feeding strategy. Assuming the two species were chronologically concomitant, their co-occurrence at the same locality with two different feeding strategies, may represent a case of niche separation. They may have hunted different types of prey, thus avoiding competition for a same food resource.

2. Introduction

Beaked whales (Ziphiidae) represent a diversified family of echolocating toothed whales (Odontoceti), currently represented by at least 22 species in 6 genera [1] with a potential new species of *Berardius* suspected in the North Pacific [2]. Their best-known modern representatives are capable of regular deep dives beyond 1000 meters to reach their foraging grounds, where they prey mostly on cephalopods and more occasionally on bathypelagic fish and crustaceans [3–10]. Most extant ziphiids are typified by a strong reduction of their teeth count reduced to one or two pairs, often only erupted in adult males [11]. Beaked whales do not use them to capture or manipulate their prey; instead, they use suction feeding as their main feeding strategy, except perhaps for the toothed ziphiid *Tasmacetus shepherdi* which retains a set of functional teeth [4]. Suction feeding forces them to be more selective relative to the size of their prey, thus allowing different species of beaked whales to be sympatric without competing for the same food resource [12,13].

Recently, Hocking et al. [14] proposed a new framework to understand the evolution of feeding in predatory aquatic mammals. Instead of thinking the different feeding styles as rigid categories, they argue that feeding strategies of aquatic mammals follow a particular evolutionary sequence that can be used to predict the origin of particular feeding styles. Under this framework, the specialization to suction feeding of extant beaked whales should arise from ancestors using a more raptorial feeding strategy. The fossil record of Ziphiidae confirm this prediction: some of the most basal beaked whales possessed elongated jaws and numerous functional interlocking teeth potentially used to capture their prey [15–18].

*Author for correspondence (benjamin.ramassamy@laposte.net).

†Present address: Department of Natural History and Palaeontology, the Museum of Southern Jutland, Lergravsvej 2, Gram, 6510, Denmark

However, morphological evidences suggest that some of them were also capable to use suction feeding at least in the most posterior part of the mandibles [17,18]. *D. mojnun*, a new fossil genus and species with close affinities to *M. gregarius*, was described from the Gram Formation in Denmark [18]. This species was interpreted as a more raptorial feeder than extant ziphiids based on its numerous interlocking teeth and elongated jaws, despite moderate adaptations to suction feeding.

A new fossil Ziphiidae from the same locality is described here. The preserved parts of the specimen consist in the lower jaw, earbones, part of the hyoid apparatus, and forelimb elements. This project aims at describing the specimen and at proposing a palaeoecological reconstruction based on morphological features. Elements relative to feeding strategies and the ecological niches occupied by the ziphiids from the Gram Formation are discussed.

3. Materials and Methods

3.1 Specimen Preparation and Computed Tomography

The specimen was discovered in 2007 and prepared by means of mechanical tools at the curatorial department of the Museum of Southern Jutland. A co-polymer of acrylates (MA/EMA Paraloid B72) was used as an adhesive to keep the fragments of the lower jaw together. Photos of the specimen were taken using a Fujifilm FinePix HS10 with a focal length of 4.2-126.0 mm.

Specimens coming from the Gram Formation are fragile, difficult to handle, and prepare. Furthermore, the preparation sometimes results in the loss of information in relation with the original placement of the bone structures. To alleviate the preparation work and avoid extensive manipulation of the specimen, the lower jaw was scanned using a clinical computed tomography (CT) system (Siemens Somatom; Siemens Medical Solutions, Forchheim, Germany) using the following parameters: 0.98×0.98×0.60 mm³ voxel size; 140 kVp tube voltage; 185 μAs tube charge, resulting in an acquisition time of ~60 s. Data was reconstructed using a B45s convolution kernel. The CT-scans unveiled the dorsal and lateral side of the lower jaw as preserved that otherwise, would have not been accessible without extensive preparation. Visualizations of the scanned fossil were made in the DICOM-viewer OsiriX (Pixmeo SARL) and image segmentation and construction of an interactive model of the fossil was made in Amira 5.6 (FEI, Visualization Sciences Group). The digital reconstruction is available in the Supplementary Data (Supplementary Figure S1).

3.2 Size Estimation and Evaluation of Trophic level

Cetaceans, particularly obligate suction feeders, are known to select their prey relative to their own size [12]. Therefore, assessing the size of a ziphiid individual and comparing it with other species may help estimating the trophic level at which the specimen fed.

To do so, two cranial measurements were collected from different ziphiids specimens: the bizygomatic width and the condylobasal length (available in Supplementary Dataset S2). Many fossil forms had to be discarded, because their partial skull did not allow a good estimation of the condylobasal length and/or the bizygomatic width. The fossil species *Ninoziphius platyrostris*, *Nazcacetus urbinai*, and *Messapicetus gregarius* were included based on the measurements provided in their respective descriptions [16,17,19]. In absence of a preserved skull for the specimen NHMD 189993 described herein, such measurements were not available. The anteroposterior length and posterior transverse width of the mandibles were used instead of, respectively, the bizygomatic width and the condylobasal length. The posterior transverse width of the specimen is a good estimator of the bizygomatic width but in this case, is slightly underestimated due to the lack of the most posterior parts of the mandibles. Anteroposterior length of the mandibles is significantly shorter than bizygomatic width in odontocetes and is therefore only an indicator of minimum size rather than a precise estimator. Cranial measurements were selected for other ziphiids, because the mandibles of beaked whales are often disarticulated and the posterior width of the mandibles is therefore not always measurable.

Ziphiid species were placed in four size categories: very large-sized ziphiids (8-10 m), large-sized ziphiids (5.5-7.5 m), medium-sized ziphiids (4-4.5 m), and small-sized ziphiids (3-4 m). These categories were defined in Bianucci et al. [20] based on a regression of the postorbital width relative to the body length of different ziphiid species.

A natural logarithmic transformation was applied to the cranial measurements to attenuate the effect of allometry and correct for heteroscedasticity [21,22]. A MANOVA (Multivariate analysis of variance) was performed to evaluate whether the cranial measurements were sufficient to assess each size category. It was followed by a Tukey's HSD (honest significant difference) test on each variable to compare differences between the size categories. Linear regression was also performed on the dataset to assess the relationship between the two cranial measurements. All Analyses were performed with the software R 3.6.0 [23].

3.3 Nomenclature

Institutional Abbreviations—**IRSNB**, Institut Royal des Sciences Naturelles de Belgique, Brussels, Belgium; **MNHN**, Muséum National d'Histoire Naturelle, Paris, France; **MSM**, Museum Sønderjylland Naturhistorie og Palæontologi, Gram

Lergrav, Gram, Denmark; **MSNUP**, Museo di Storia Naturale dell'Università di Pisa, Italy; **MUSM**, Museo de Historia Natural, Lima, Peru; **NMNZ**, National Museum of New Zealand Te Papa Tongarewa, Wellington, New Zealand; **NHMD**, Statens Naturhistoriske Museum, Copenhagen, Denmark; **USNM**, United States National Museum of Natural History, Smithsonian Institute, Washington D.C., U.S.A.

Terminology—The anatomical terminology of the skull and earbones follows Mead and Fordyce [24]. The terminology used by Fitzgerald [25] and Marx et al. [26] was followed for the postcranial remains. The nomenclature of Reidenberg and Laitman [27] was used for describing elements of the hyoid apparatus.

4. Results

4.1 Systematic Palaeontology

Order CETACEA Brisson, 1762
Suborder ODONTOCETI Flower, 1867
Family ZIPHIIDAE Gray, 1850
Genus and species indet.

Referred Material—NHMD 189993, subcomplete mandibles, the right stylohyal, 14 isolated teeth including a tusk, two periotics and the right tympanic, the right humerus and associated radius, parts of the nasal

Horizon and Locality—The locality is situated 1.5 km north of the town of Gram, Southern Jutland, Denmark (55°18'02.67"N, 9°30'32.51"E; Fig. 1). The specimen was dated based on the mollusc fauna identified in association with genus and sp. indet. NHMD 189993. The high percentage of *Carinastarte vetula reimersi* (55% of the specimens identified) and the co-occurrence of the species *Gemmula badensis* and *Turritella tricarinata* suggest that the specimen was originally found in the assemblage Zone V [28].

The assemblage Zone V belongs to the upper part of the Gram Formation dated from the Tortonian age, based on the co-occurrence of the dinoflagellate cysts *Hystichosphaeris obscura*, *Spiniferis solidago*, and *Labyrinthinium truncatum* [29]. The minimum age of the Assemblage Zones from the Gram Formation is estimated to 9.9 Ma based on the presence of a polarity zone of less than 70 000 years [30]. The specimen genus and sp. indet. NHMD 189993 can be dated from the mid- to late Tortonian, ca. 9.9–7.2 Ma.

Systematic Attribution of the Specimen—The specimen is identified as a member of the family Ziphiidae based on the following combination of characters: the enlargement of the apical or subapical mandibular tooth; the reduction of the dorsal keel on the posterior process of the periotic; the mediolateral thickening of the anterior process of the periotic; in dorsal view, the anterior shift of the pars cochlearis of the periotic.

Genus and sp. indet. NHMD 189993 clearly differs from the other species found in the Gram Formation, *Dagonodum mojnium*, based on the following combination of characters: the reduction of the mandibular tooth, the shorter unfused symphysis, the dorsoventral thickening of the anterior process of the periotic, the dorsoventral compression of the pars cochlearis, the presence of a cochlear spine.

Identification beyond the genus was not possible, because of the unusual nature of the preserved material. Most fossil ziphiids are represented by cranial remains, mostly the rostral, preaural, and vertex region [15,20,31–33]. Mandibles, earbones, and postcranial remains are more rarely preserved. Despite the unusual features present on the periotics (presence of a cochlear spine, depression along the medial surface of the posterior process) and the peculiar size of the specimen, the lack of cranial remains makes it nearly impossible to compare it with many similarly sized ziphiids only known from the preaural region (e.g. *Africanacetes*, *Globicetus*, *Tusciziphius*). Genus and sp. indet. NHMD 189993 possesses several derived features thought to be characteristic of a crown Ziphiidae [e.g. 16,33,33]: the dorsoventral thickening of the anterior process of the periotic bone, the dorsoventral compression of the pars cochlearis, the short and unfused symphysis. However, a recent phylogenetic analysis proposed that some members of the more basal *Messapicetus* clade displayed derived characters indicative of a convergent evolution between stem and crown Ziphiidae (Bianucci et al., 2016a). Mandibles, earbones and postcranial material of the most derived members of this clade, *Globicetus*, *Tusciziphius*, and *Imocetus* are not known [33]. It is therefore impossible to assess whether genus and sp. indet. NHMD 189993 was a crown ziphiid or a member of the *Messapicetus* clade.

By measure of caution, the advice of Barnes [34] and Fordyce and Muizon [35], which suggest that the identification of a new species should at least include skull and rostrum, is followed until more cranial material is available.

4.2 Description and Comparisons

4.2.1 Cranium and Mandible

Overview and ontogeny—The specimen is interpreted as an adult based on the complete fusion of the humeral head to the humeral shaft and the epiphyseal ankylosis of each epiphysis of the radius. In the porpoise *Phocoena phocoena*, extensive ankylosis of the postcranial skeleton characterizes adult specimens [36].

The most robust parts of the mandibles and postcranial elements of ~~genus and sp. indet.~~ NHMD 189993 are well preserved compared to the more fragmentary cranial remains and ribs (Fig. 2-3). The earbones were found with the lower jaw, the right periotic still having the stapes firmly attached to it (Fig. 4-5; only right periotic illustrated). The humerus and radius were originally still articulated (Fig. 6A-F), whereas the stylohyal lied along the right lateral side of the symphysis (Fig. 6O-Q). Teeth were collected around the ~~specimen out of their mandibular sockets~~ (Fig. 6G-N). The preserved parts of the mandibles are 1032 mm long and 412 mm wide. More measurements of the specimen are available in Table 1 and 2.

Nasal—Because the nasal is the only piece identifiable from the shattered cranium of the specimen, its orientation reveals different. The dorsal exposure is flat and rectangular (Fig. 2C). The ratio between the width and length of the visible surface is 0.70 (60 mm long and 86 mm wide). No excavation is visible on the surface of the nasal bone.

Periotic—Measurements of the right periotic are available in Table 2. The anterior process of the periotic is transversely thickened, with a large rounded protuberance along the dorsomedial surface of the periotic (Fig. 4A-C). ~~This strong thickening both~~ lateromedial and dorsoventral, is observed in all crown ziphiids, but not in the stem-ziphiids *Dagonodum mojnum*, *Messapicetus gregarius*, and *Ninoziphius platyrostris* [16–18]. ~~In the latter,~~ the thickening occurs only lateromedially. The tip of the anterior process is pointed. In ventral view, the anterior bullar facet is anteroposteriorly elongated and elliptical. Posteromedially to this facet, the accessory ossicle is still articulated in the fovea epitubaria (Fig. 4D). It extends along the dorsomedial margin of the anterior process. It is less rounded and developed than in *Berardius*, *Hyperoodon*, some species of *Mesoplodon* (*M. carlhubbsi*, *M. europaeus*, *M. grayi*, *M. mirus*), *Nazcacetus*, and *Tasmacetus*. In ventral view, a sulcus extends anteroposteriorly along the accessory ossicle, and separates it in two (Figure 4D). A similar sulcus is also observed in *H. ampullatus*, *M. densirostris*, and *T. shepherdi*, although ~~less developed in those species.~~ The sulcus observed in ~~genus and sp. indet.~~ NHMD 189993 could refer to the origin of the tendon of m. tensor tympani [24]. Posteriorly to the fovea epitubaria and the accessory ossicle, the malleolar fossa develop along the medial margin of the lateral tuberosity. In ventral view, the anterior process is separated from the lateral tuberosity by the anteroexternal sulcus, which can also be seen in lateral view. The anteroexternal sulcus is also present in the periotic of *D. mojnum*. In ventral view, the lateral tuberosity is lateromedially elongated, a character observed in all ziphiids, except *D. mojnum*, *M. gregarius*, and *N. platyrostris* [16–18]. The fenestra ovalis is rounded. Posteroventrally to the fenestra ovalis, a deep hiatus epitympanic separates the posterior process from the lateral tuberosity. In ventral view, the posterior process of the periotic is fan-shaped: it is rounded and widens abruptly ~~from anterior to posterior~~ (Fig. 4D). A fan-shaped posterior bullar facet is characteristic of all ziphiids, except *D. mojnum*, *N. platyrostris*, and *M. gregarius* [16–18]. In medial view, the posterior process is oriented posteroventrally. ~~Genus and sp. indet.~~ NHMD 189993 lacks a distinct keel along the whole posterior process, a feature present in all ziphiids [17]. A deep depression excavates the anteromedial side of the posterior process, just posterior to the pars cochlearis (Fig. 4C). This depression seems unique to ~~genus and sp. indet.~~ NHMD 189993 ~~and was not observed in the periotic bone of other ziphiids.~~

In dorsolateral view, the pars cochlearis is anteriorly shifted, a feature that distinguishes ~~an eurhinodelphid from a~~ ziphiid periotic (Fig. 4A) [16]. In ventromedial view, the pars cochlearis is rectangular, because of its straight anteromedial corner. It is also dorsoventrally compressed, a feature observed in crown ziphiids, but absent in *N. platyrostris* and members of the *Messapicetus* clade [17]. In ventral view, the pars cochlearis bears a triangular depression similar in shape to *D. mojnum* [18]. This depression is also visible in the species *Mesoplodon mirus* (USNM 504612, USNM 550351, USNM 572961, USNM KLC112) and *M. bidens* (MNHN 1975.112, SNM CN5x), but is elliptical, more elongated anteroposteriorly ~~in both species.~~ The tear-shaped fenestra rotunda is oriented posteroventrally. Posterodorsally to the internal acoustic meatus, the periotic bears a large cochlear spine. This unusual feature in Ziphiidae is present in ~~the species~~ *N. platyrostris* and *Berardius armuxii* [17]. Its presence was also observed in the species *B. bairdii* (USNM 571524). The cochlear spine in ~~genus and sp. indet.~~ NHMD 189993 is moderately developed dorsally, a condition similar to the genus *Berardius* and differing from the well-marked cochlear spine of *N. platyrostris*. In dorsomedial view, the internal acoustic meatus is elliptical; this feature is also observed in the periotic of *N. platyrostris* and is connected to the presence of the cochlear spine. A thick crest separates the internal acoustic meatus from the aperture for the vestibular aqueduct (Fig. 4C, G). Inside the internal acoustic meatus, the dorsal vestibular meatus is separated from its ventral counterpart by a transverse crest. The ventral vestibular area occupies almost two thirds of the surface of the internal acoustic meatus. Posteriorly to the vestibular area of the internal acoustic meatus, the aperture for the vestibular aqueduct is anteroposteriorly compressed. Ventrally to the vestibular aqueduct, the aperture for the cochlear aqueduct is reduced to a small opening (Fig. 4C).

Tympanic bulla—The right tympanic bulla is partially preserved (Fig. 5). Further measurements are available in Table 2. It lacks the base of the pedicle, the sigmoid process, and the dorsal part of the outer lip. In ventral view, the bulla is heart-shaped, because of the interprominent ~~notch~~ well marked posteriorly that separates the inner and outer posterior prominences. In ventral view, the inner ~~posterior~~ prominence is compressed transversely and is less developed posteriorly than the twice-larger outer posterior prominence (Fig. 5C). The compression of the inner posterior prominence is similar to some species of *Mesoplodon* (e.g. *M. bidens* SNM CN5x, *M. bowdoini* NMNZ MM2653, *M. europaeus* USNM 504349), *Messapicetus gregarius*, and *N. platyrostris*. In *Hyperoodon* spp. and *Z. cavirostris*, the inner posterior prominence is more reduced and is even shorter posteriorly. In ventral view, the interprominential notch connects to the deep median furrow. The median furrow extends roughly until the first third of the bulla (Fig. 5C). The median furrow is less developed in *Hyperoodon* spp. and *Z. cavirostris*, but more extended in the stem ziphiids *D. mojnum*, *M. gregarius*, and *N. platyrostris*. In ventral view, a keel extends along the whole anteroposterior length of the bulla.

The involucrum is indented, a feature visible both in dorsal and medial view (Fig. 5A-B). In medial view, the ventral part of the bulla is incurved, ~~but does not reach~~ the dorsalmost margin of the posterior portion of the involucrum, as in *D. mojnum* (Fig. 5B). The anterior margin of the tympanic bulla is too damaged to assess the degree of development of the

tympanic spine, if present. However, the broken anterolateral margin of the bulla develops anteriorly into a thin bone plate (Fig. 5A), a condition similar to *N. platyrostris*, where the tympanic spine is absent [17].

Stapes—The right stapes is still firmly attached to the periotic in the fenestra ovalis and could not be removed (Fig. 4C). The stapes is conical, widening at its oval base, as observed in several ziphiids species (Lambert et al., 2009). The head of the stapes has a circular outline. The small and circular vestigial stapelial foramen opening is situated approximately at mid-length of the stapes. The muscular process is well developed and situated at the level of the head of the stapes.

Mandible—The mandible of ~~genus and sp. indet.~~ NHMD 189993 lacks the posterior part of the acoustic window and the mandibular condyle (Fig. 2-3). ~~Further measurements are available in Table 1.~~ The symphyseal portion of the mandible is unfused (Fig. 3C). It is not ankylosed as in the long snouted stem ziphiids *Ninoziphius platyrostris*, *Messapicetus* spp., and *Dagonodum mojnum* [16–18,37]. The symphysis is 289 mm long and represents at most 28 percent of the total length of the mandible (the total length of the preserved parts is 1032 mm). This value is much lower than in long snouted ziphiids and *Tasmacetus shepherdi*, where the symphysis extends at least along 36 percent of the mandible total length [17]. The symphyseal portion of ~~genus and sp. indet.~~ NHMD 189993 is triangular, differing from the half-circled section of *Berardius* spp., *D. mojnum*, *Messapicetus* spp., *N. platyrostris*, and *Tasmacetus shepherdi* [16–18]. The symphyseal portion of the mandible is turned upwards. This feature is also present in ~~the species~~ *H. ampullatus*, *M. bidens*, *M. grayi*, *M. mirus*, *N. urbinai*, and *Z. cavirostris* [19].

The apex of the mandibles is heavily fractured, but the fragments conserved their original position, allowing an estimation of the original outline. In ventral view, the apex is rounded and likely possessed an enlarged alveolus for the tusk. This interpretation fits with the shape and size of the preserved tusk that is similar to those of several long snouted stem beaked whales (*D. mojnum*, *Mess. gregarius*) possessing a pair of tusks in apical position (Fig. 6G-H). Furthermore, no other alveolus along the alveolar groove is sufficiently developed to support the tusk. The apex of the mandible is too fractured to identify precisely whether the tusk was positioned apically or subapically, as observed by Dalebout et al. [38] in *Mesoplodon perrini*. It is also possible that ~~genus and sp. indet.~~ NHMD 189993 possessed two pairs of tusks, even though only one tusk is preserved with the specimen (Fig. 6G-H). This character is observed in *Berardius* spp., *Anoploussa forcipata*, and *D. mojnum* [18,39]. One mental foramen is visible along the lateral side of the mandible. It is elongated, well-marked and situated slightly posterior to the symphysis.

The outline of individualized alveoli can be distinguished in the alveolar border (Fig. 3C). The number of counted alveoli along the left dentary is 17, which is probably a slight underestimation, because of the eroded and fractured surface of the alveolar border of the most apical parts of the symphysis. It is not possible to assess the presence of a diastema between the tusk and the rest of the alveolar groove. The alveoli are oval, transversely compressed like in the *Messapicetus* spp. [15,16]. However, they are much more reduced than in the latter and much shallower compared to long snouted stem ziphiids and the species *Tasmacetus shepherdi* [17,18]. In the 3D reconstruction of the lower jaw, in lateral view, the position of the bone fragments posterior to the alveolar groove suggests the presence of a precoronoid crest. However, the dorsal surface of the acoustic window is too fractured to be certain of it.

Teeth—Together with the mandibles of ~~genus and sp. indet.~~ NHMD 189993, 14 isolated teeth were retrieved (Fig. 6G-N). Their crown is approximately as developed as the root dorsoventrally and curves lingually. The crown progressively widens ventrally and projects posteroventrally. The section at the base of the crown is circular, whereas the transverse section of the root is more oval. An oval root is present in *Messapicetus* spp., unlike the circular section observed in *Tasmacetus shepherdi* and the squared root observed in the species *Dagonodum mojnum* and *Ninoziphius platyrostris*. A distinct mesial keel is present in some of the smallest teeth of ~~genus and sp. indet.~~ NHMD 189993. Despite the presence of individualized alveoli, the reduced size of the teeth and the particularly shallow alveoli suggest that the teeth of ~~genus and sp. indet.~~ NHMD 189993 were not as robust as in other known toothed beaked whales, perhaps still embedded in the gum, as observed in some specimens of extant ziphiids (e.g. *Hyperoodon ampullatus*, *Mesoplodon grayi*; [40,41]).

An enlarged tooth interpreted as a tusk was found with ~~genus and sp. indet.~~ NHMD 189993. This tooth is more massive than the other reduced teeth (Fig. 6G-H). The tusk is triangular, with a root more developed dorsoventrally than the crown. The root of the tooth is transversely compressed with an oval outline (Fig. 5H). As suggested by the outline of the apex of the mandible, the tusk most likely fitted in apical or subapical position of the mandible. The slightly rounded tip of the crown also suggests that the tusk is slightly worn and was originally erupted. The tusk resembles those of long snouted ziphiids, *Berardius* spp. and *T. shepherdi* due to their transverse compression. It differs from the apical tusk present in males *Hyperoodon ampullatus* and *Ziphius cavirostris*, which is more conical. It also differs from the genus *Mesoplodon* where the tusk is heavily compressed transversely, including in species where the tusk is in apical or subapical position (*M. mirus* USNM 504612; *M. hectori* NMNZ MM0002901; *M. perrini* USNM 504260).

4.2.2 Postcranial Elements

Hyoid apparatus—The right stylohyal is 238 mm long, 48 mm wide and 26 mm thick (Fig. 6O-Q). The length of this bone is almost twice longer than in *Mesoplodon layardi* (NMNZ 1899: 109 mm; NMNZ 2917: 166 mm). The stylohyal length of ~~genus and sp. indet.~~ resemble more the one observed in *Hyperoodon planifrons* (NMNZ 1806: 272 mm; NMNZ DM 1878: 246 mm) and *Ziphius cavirostris* (SNM CN1: 248 mm). A constriction is present on the most anterior part of the stylohyal, at the level of the articulation with the epiphyal (Fig 5O). This constriction is observed in ~~the species~~ *Berardius armuxii* (MNHN A3244) and *Tasmacetus shepherdi* (MM 2908). In lateral view, the stylohyal progressively widens from anterior to posterior and reach its maximum transverse width in the posterior part of the bone. The posterior margin of the bone that

articulates with the tympanohyal is pointed (Fig. 5O). The shape of the stylohyal of ~~genus and sp. indet.~~ NHMD 189993 resembles those of *Z. cavirostris* and *H. planifrons*, even though in those species, the transverse widening is more pronounced (between 29 % and 68 % wider), with a flatter dorsal surface. This shape is also observed in *Mesoplodon europaeus* [27]. It differs from the stylohyal observed in several other species of *Mesoplodon* ~~consulted~~ (*M. bidens* MNHN 1963-259, MNHN 1963-111; *M. europaeus*, NMNZ 550390; *M. layardii* NMNZ 2917), where the lateral and medial margins of the bone stay straight, without transverse widening. A ridge goes along the lateral side of the stylohyal of ~~genus and sp. indet.~~ NHMD 189993 (Fig. 6O), as observed in the ziphiids *Mesoplodon europaeus* and *M. mirus* [27]. This ridge gives a triangular transverse section to the bone.

Humerus—The right humerus is fully preserved (Fig. 6A-C). Further measurements are available in Table 1. It is 204 mm long and 92 mm wide at the level of the deltoid ridge. The ratio between the humeral length and the estimated bitygomatic width (posterior width of the mandible used in the specimen) is similar to *Messapicetus gregarius* (in *M. gregarius*: 0.48; in ~~genus and sp. indet.~~ NHMD 189993: 0.50). Both species display a proportionally longer humerus than most extant ziphiids [42]. The head of the humerus is hemispherical. In lateral view (Fig. 6A), the humeral head represents a quarter of the total length of the humerus. In *M. gregarius*, the head is more prominent and anterolaterally oriented: it represents almost a third of the total length of the humerus. In lateral view, the deltoid ridge is well developed along the anterior margin of the humerus (Fig 6A). It develops approximately at mid length of the humerus, and over a third of its length. The presence of a developed deltoid ridge is characteristic of extant Ziphiidae, even though it does not develop as much as in Physteridae [43]. The posterior part of the humerus of ~~genus and sp. indet.~~ NHMD 189993 does not widen, unlike many odontocetes [43]. This feature is characteristic of the ziphiid humeri [43]. In posterior view, the articular facets for the radius and the ulna are well separated by a crest (Fig. 6C). Each facet occupies approximately half of the posterior surface of the humerus.

Radius—The associated right radius was originally found articulated with the humerus (Fig. 5D-F). ~~Further measurements are available in Table 2. The radius curves anteroposteriorly. It measures~~ 186 mm long, 63 mm wide at mid-length. The facet for articulation of the humerus is oriented anterodorsally (Fig. 5D-E). Posteriorly, the articulations for the scaphoid and the lunate are well defined; they occupy approximately half of the posterior width of the radius. The articulation for the scaphoid is straight in lateral view, whereas the articulation for the lunate is more oblique and face posterodorsally. In all Ziphiidae, and contrary to other odontocetes, the posterior part of the radius is not widened [43]. The overall shape of the radius does not significantly differ from extant ziphiids ~~consulted~~ (e.g. *Berardius arnuxii* NMNZ 415, MNHN A3244; *Mesoplodon layardii* NMNZ 2917; *Tasmacetus shepherdii* NMNZ MM 2908; *Ziphius cavirostris* SNM CN1). However, its radius is wider than in *Messapicetus gregarius* where the ratio between the length and the width of radius equals 0.25 (0.34 in ~~genus and sp. indet.~~ NHMD 189993).

Ribs—two partial ribs are preserved (Fig. 2C). Their body is heavily fractured and fragmented. Judging from the similar outline of each rib, they were likely from the same pair. They are tentatively inferred to be the pair 2, because of their thick, yet flattened body. Both are double-headed with a marked neck separating the capitulum from the tuberculum.

4.2.1 Size estimates of the specimen

Condylbasal length and bitygomatic width were strongly correlated across the dataset ($R^2=0.78$; Fig. 7). The combination of the two linear measurements was sufficient to separate the four size categories (p -value < 0.0001), and each size category were well-distinguished.

The four size categories were better separated using the bitygomatic width, particularly in the case of the medium-sized and large-sized ziphiids. Indeed, species from these two categories displayed similar range of variation in condylbasal length, because of the strong variability of the rostrum anteroposterior length in those species. The large-sized ziphiid *Ziphius cavirostris* displayed a condylbasal length similar to other medium-sized ziphiids, such as *Mesoplodon mirus* and *M. europaeus*. This species was easily distinguished from the three other large-sized ziphiids *Mesoplodon layardii*, *Tasmacetus shepherdii* and *Hyperoodon planifrons* (Fig. 7).

On the opposite, the long-snouted medium-sized ziphiids *Messapicetus gregarius*, *Ninoziphius platyrostris*, *Dagonodum mojunum*, and *Mesoplodon grayi* displayed condylbasal length matching some large-sized ziphiids (*Mesoplodon layardii*). One specimen of *M. gregarius* almost reached the condylbasal length of some of the smallest specimens of very large-sized ziphiids. The long-snouted stem ziphiids were well separated from other ziphiids from their size category, including *Mesoplodon grayi*, which also display a strong elongation of the rostrum, but is also characterized by a smaller bitygomatic width. Small-sized ziphiids were easily distinguished from other size categories. They consist in the species *Mesoplodon peruvianus* and the fossil species *Nazcacetus urbinaei* that possess a slightly longer condylbasal length.

~~Medium-sized ziphiids were better differentiated from large-sized ziphiids based on bitygomatic width.~~ Based on the estimate of the bitygomatic width and condylbasal length, ~~genus and sp. indet.~~ NHMD 189993 would be a large ziphiid, between 5.5 and 7.5 m. Its condylbasal length and bitygomatic width is within the range of the extant Ziphiidae *Mesoplodon layardii* whose size generally ranged between 5.5 and 6 m [44]. Based on the similar condylbasal length, the degree of elongation of the rostrum in ~~genus and sp. indet.~~ NHMD 189993 is likely more similar to *M. layardii* than to the shorter rostrum of *Z. cavirostris* or the extremely elongated rostrum of long-snouted stem ziphiids. The size of ~~genus and sp. indet.~~ NHMD 189993 clearly differs from the other fossil ziphiid found in the Gram Formation, *Dagonodum mojunum*, a medium-sized ziphiid whose size likely ranged between 4 and 4.5 m.

5. Discussion

5.1 Foraging Ecology

5.1.1 Suction Feeding

All extant beaked whales are specialized to use suction: they generate powerful suction pressures with their tongue acting like a piston, to capture and engulf their prey [4]. Many odontocetes can use suction feeding for capturing and/or transporting prey [45], but extant Ziphiidae are obligate suction feeders, due to the absence of functional teeth to capture their prey [4]. Furthermore, they exhibit a lateral closure of the intraoral cavity combined with a wider and thicker hyoid apparatus compared to odontocetes relying on a more raptorial feeding strategy [4]. The only extant ziphiid species that is perhaps not an obligate suction feeder is *Tasmacetus shepherdi*. Unlike other ziphiids, this species retains a set of erupted teeth likely functional [46]. Based on one stomach content containing mostly the fish species *Merluccius hubbsi*, MacLeod et al. [13] speculated that this species may specialize in feeding on deep-water fish rather than cephalopods, thus limiting the competition with other species of beaked whales in the southern oceans where it occurs.

Several lines of evidence suggest that ~~genus and sp. indet.~~ NHMD 189993 was capable of using suction feeding to a larger extent than other toothed ziphiids, ~~perhaps already an obligate suction feeding species~~. First, the stylohyal is strongly thickened and elongated. Its anteroposterior length is similar to the large ziphiids *Hyperoodon planifrons* and *Ziphius cavirostris*, whereas it is almost twice longer than in *Mesoplodon layardii*, a species close in size to NHMD 189993. A thickened stylohyal would be necessary to support strong tongue muscles. The styloglossus and the hyoglossus are the two main muscles responsible for the retraction of the tongue in a piston-like manner during suction [27]. The styloglossus originates on the lateral surface of the stylohyal, which is particularly thickened in the ziphiids *Mesoplodon mirus* and *M. europaeus*. Both species also possess a strong styloglossus: Reidenberg and Laitman [27] noticed that they developed the largest styloglossus relative to total body length from their sample. The thickening of the stylohyal is accompanied by a pronounced ridge along the lateral surface of the bone giving the stylohyal a triangular shape in transverse view [27]. This ridge was observed in several other ziphiid species (e.g. *Hyperoodon* spp., *Berardius* spp., *Mesoplodon bidens*, *M. layardii*, *Ziphius cavirostris*) and seems characteristic of the ~~genus and sp. indet.~~  NHMD 189993.

The thickening of the stylohyal is accompanied by a reduction of the teeth in ~~genus and sp. indet.~~ NHMD 189993. For each dentary, the specimen possessed at least 17 alveoli, a number similar to *T. shepherdi* (18-28) [46], but largely inferior to long snouted stem ziphiids (*Dagonodum mojnium*, 29; *Messapicetus gregarius*, 25-26; *Ninoziphius platyrostris*, 40-42) [16-18]. Additionally, several features of the teeth and the alveoli differ between the aforementioned species and ~~genus and sp. indet.~~ NHMD 189993. The alveoli of the specimen, although individualized, are particularly shallow, greatly differing from the condition observed in other toothed beaked whales where the alveolar groove is deep, but the septa not necessarily well differentiated (e.g. *M. gregarius*; Bianucci et al., 2010). The teeth themselves are also well reduced compared to other toothed ziphiids including *T. shepherdi*. The longest tooth of ~~genus and sp. indet.~~ NHMD 189993 (excepting the tusk) measures 20 mm with a maximum diameter of 11 mm. The tooth measurements of the specimen are even smaller than in the medium-sized ziphiids *D. mojnium*, *M. gregarius*, and *N. platyrostris*. Furthermore, in ziphiids with functional teeth, the robust crown shows apical wear or interlocking facets suggesting that their dentition was functional. This is not the case in ~~genus and sp. indet.~~ NHMD 189993 where the small crown does not show sign of interlocking. In many teeth, the apex is broken off and does not allow an estimation of the degree of apical wear, but the few teeth with a preserved apex do not show signs of wear. Therefore, we hypothesize that the small teeth of NHMD 189993 were either embedded from the gum or too small to be used as a regular method of capture. We do not discard the possibility that NHMD 189993 could occasionally use its reduced teeth (if erupted) to manipulate or capture some of its prey.

5.1.2 A potential case of niche separation

Several morphological evidences and the different size estimates of *Dagonodum mojnium* and ~~genus and sp. indet.~~ NHMD 189993 suggest that these two species occupied two different ecological niches. It is possible that the two species were not contemporary: despite the similar age estimation for the two species, the error margin of ± 7 Ma could mean that they did not live at the same period of time. Assuming that the two species were contemporary, the ~~co-~~ -occurrence of two different sized species of Ziphiidae at the same location suggest a case of niche separation.

Cases of niche separation are known in extant ziphiids: *Mesoplodon* species consistently feed on smaller prey type (generally, cephalopods under 500 g) compared to *Hyperoodon* and *Ziphius* species (cephalopods over 1 kg) [13]. The difference of prey size targeted may explain why species of *Mesoplodon* are often sympatric with the latter [47-50]. Size is not the only component, even though an important one [12], allowing niche separation between ziphiid species. In the case of *D. mojnium* and ~~genus and sp. indet.~~ NHMD 189993, the difference in specialization to suction feeding reinforces this hypothesis. The species *D. mojnium* possesses some adaptations to suction feeding (transverse thickening of the basyhyal and thyrohyal; presence of a precoronoid crest) [18], but not to the extent of ~~genus and sp. indet.~~ NHMD 189993, that probably relied more prominently on this feeding strategy. Obligate suction feeders with a reduced tooth count are often more teuthophagous [4,51], even though some ziphiids can still feed on fish [13]. Perhaps, the more specialized oral apparatus of ~~genus and sp. indet.~~ NHMD 189993 is more indicative of a more predominant teuthophagous diet than *D.*

mojnum. Interestingly, the fossil of a cuttlefish (Sepiida) was found in the Gram Formation (MSM DK718; unpublished data), an ideal prey type for ~~genus and sp. indet.~~ NHMD 189993.

~~Alternatively, the two species may have segregated geographically: whereas *D. mojnum* is assumed to be a local [18], the other specimen genus and sp. indet. NHMD 189993 may have been more adapted to deep diving like its modern representatives. No morphological features preserved allow the evaluation of the diving abilities of genus and sp. indet. NHMD 189993. Therefore, this hypothesis cannot be fully ruled out.~~

Other cases of niche separation between fossil ziphiids probably occurred at other locations where they show a diversity of sizes or feeding strategies. Bianucci et al. [52] already proposed this interpretation to explain the high diversity of fossil ziphiids trawled from the seafloor off South Africa. Fossil ziphiids from the Neogene of Antwerp and fished from the Atlantic Ocean floor off the Iberian Peninsula also show a great diversity of skull sizes, that could be indicative of ecological niche segregation [31,33]. In absence of precise datation for these three localities, it is unclear whether the different species were living during the same time period.

Conclusion

Despite the rich fossil record of beaked whales, the discovery of postcranial material remains a rare finding [18,37,53]. A new fossil of Ziphiidae, ~~genus and sp. indet.~~ NHMD 189993, consisting in the mandible, earbones, the stylohyal, isolated teeth including the tusk, the humerus and associated radius is described here. The fossil is dated to the mid- to late Tortonian (ca. 9.9-7.2 Ma). Despite the lack of cranial material for comparing with other similarly sized fossil ziphiids, ~~genus and sp. indet.~~ NHMD 189993 clearly differs from the other species found in the Gram Formation during the same time period, *Dagonodum mojnum*.

Unlike *D. mojnum* and other long-snouted stem ziphiids, the morphology of the oral apparatus of ~~genus and sp. indet.~~ NHMD 189993 suggests that it was mostly relying on suction feeding. The reduced teeth perhaps still embedded in the gum, and morphological features of the thickened stylohyal supports this interpretation.

The two fossil species *D. mojnum* and ~~genus and sp. indet.~~ NHMD 189993 likely occupied different ecological niches with ~~genus and sp. indet.~~ NHMD 189993 likely feeding more predominantly on evasive prey such as cephalopods. Assuming the two species were chronologically concomitant, the spatial co-occurrence of the species *D. mojnum* and ~~genus and sp. indet.~~ NHMD 189993 can be illustrative of a case of niche separation. Together with sexual dimorphism [54], the specialization toward specific ecological niches in Ziphiidae may partly explain the rich specific diversity of this family.

Acknowledgments

We thank Mette Steeman for her suggestions and support for the completion of the project. We are also indebted to Frank Osbæk and Trine Sørensen that were in charge of the preparation of the specimen.

We thank the following colleagues for kindly allowing us to access some of the comparative material we used in this study: Morten Tange Olsen and Daniel Klingberg Johansson to access the SNM collection, Anne-Lise Folie for the IRSNB collection, Christian de Muizon and Christine Lefèvre for the MNHN collection, Chiara Sorbini for the MSNUP collection, Thomas Schultz for the NMNZ, and Charles Potter for the USNM collection.

Funding Statement

The project was funded by the Dansk Slots- og Kulturstyrelsen (FORM.2016-0021).

Data Accessibility

The datasets supporting this article have been uploaded as part of the Supplementary Material at the following address:

<https://figshare.com/s/c75c55bdca6ef3ae3a6f>

Competing Interests

'We have no competing interests.'

Authors' Contributions

B.R. identified and interpreted the ziphiid remains. H.L. created the digital reconstruction and the associated figures. B.R. made calculations for the size estimates and collected the measurements. Both authors participated in the preparation of the illustrations and the writing of the manuscript. Both discussed the results and commented on the manuscript at all stages.

References

1. Dalebout ML *et al.* 2014 Resurrection of *Mesoplodon hotaula* Deraniyagala 1963: A new species of beaked whale in the tropical Indo-Pacific. *Marine Mammal Science* **30**, 1081–1108. (doi:10.1111/mms.12113)
2. Morin PA *et al.* 2017 Genetic structure of the beaked whale genus *Berardius* in the North Pacific, with genetic evidence for a new species. *Marine Mammal Science* **33**, 96–111. (doi:10.1111/mms.12345)
3. Clarke MR. 1996 Cephalopods as prey. III. Cetaceans. *Philosophical Transactions of the Royal Society of London. Series B: Biological Sciences* **351**, 1053–1065. (doi:10.1098/rstb.1996.0093)
4. Heyning JE, Mead JG. 1996 Suction feeding in beaked whales: morphological and observational evidence. *Natural History Museum of Los Angeles County Contributions in Science* **464**, 1–12.
5. Hooker Sascha K., Baird Robin W. 1999 Deep-diving behaviour of the northern bottlenose whale, *Hyperoodon ampullatus* (Cetacea: Ziphiidae). *Proceedings of the*

- Royal Society of London. Series B: *Biological Sciences* **266**, 671–676. (doi:10.1098/rspb.1999.0688)
6. MacLeod CD *et al.* 2006 Known and inferred distributions of beaked whales species (Cetacea: Ziphiidae). *J. Cetacean Res. Manage.* **7**, 271–286.
7. Johnson Mark, Madsen Peter T., Zimmer Walter M. X., Aguilar de Soto Natacha, Tyack Peter L. 2004 Beaked whales echolocate on prey. *Proceedings of the Royal Society of London. Series B: Biological Sciences* **271**, S383–S386. (doi:10.1098/rsbl.2004.0208)
8. Tyack PL, Johnson M, Soto NA, Sturlese A, Madsen PT. 2006 Extreme diving of beaked whales. *J. Exp. Biol.* **209**, 4238. (doi:10.1242/jeb.02505)
9. Minamikawa S, Iwasaki T, Kishiro T. 2007 Diving behaviour of a Baird's beaked whale, *Berardius bairdii*, in the slope water region of the western North Pacific: first dive records using a data logger. *Fisheries Oceanography* **16**, 573–577. (doi:10.1111/j.1365-2419.2007.00456.x)
10. Schorr GS, Falcone EA, Moretti DJ, Andrews RD. 2014 First Long-Term Behavioral Records from Cuvier's Beaked Whales (*Ziphius cavirostris*) Reveal Record-Breaking Dives. *PLoS ONE* **9**, e92633. (doi:10.1371/journal.pone.0092633)
11. Moore JC. 1968 Relationships among the living genera of beaked whales. *Fieldiana: Zoology* **53**, 209–298.
12. MacLeod CD, Santos MB, López A, Pierce GJ. 2006 Relative prey size consumption in toothed whales: implications for prey selection and level of specialisation. *Mar Ecol Prog Ser* **326**, 295–307. (doi:10.3354/meps326295)
13. MacLeod CD, Santos MB, Pierce GJ. 2003 Review of Data on Diets of Beaked Whales: Evidence of Niche Separation and Geographic Segregation. *Journal of the Marine Biological Association of the United Kingdom* **83**, 651–665. (doi:10.1017/S0025315403007616h)
14. Hocking David P., Marx Felix G., Park Travis, Fitzgerald Erich M. G., Evans Alistair R. 2017 A behavioural framework for the evolution of feeding in predatory aquatic mammals. *Proceedings of the Royal Society B: Biological Sciences* **284**, 20162750. (doi:10.1098/rspb.2016.2750)
15. Bianucci G, Landini W, Varola A. 1994 Relationships of *Messapicetus longirostris* (Cetacea, Ziphiidae) from the Miocene of South Italy. *Bollettino della Società Paleontologica Italiana* **33**, 231–241.
16. Bianucci G, Lambert O, Post K. 2010 High concentration of long-snouted beaked whales (genus *Messapicetus*) from the Miocene of Peru. *Palaeontology, Wiley* **53**, 1077–1098.
17. Lambert O, de Muizon C, Bianucci G. 2013 The most basal beaked whale *Ninoziphius platyrostris* Muizon, 1983: clues on the evolutionary history of the family Ziphiidae (Cetacea: Odontoceti). *Zoological Journal of the Linnean Society* **167**, 569–598. (doi:10.1111/zoj.12018)
18. Ramassamy B. 2016 Description of a new long-snouted beaked whale from the Late Miocene of Denmark: evolution of suction feeding and sexual dimorphism in the Ziphiidae (Cetacea: Odontoceti). *Zoological Journal of the Linnean Society* **178**, 381–409. (doi:10.1111/zoj.12418)
19. Lambert O, Bianucci G, Post K. 2009 A new beaked whale (Odontoceti, Ziphiidae) from the middle Miocene of Peru. *Journal of Vertebrate Paleontology* **29**, 910–922. (doi:10.1671/039.029.0304)
20. Bianucci G, Lambert O, Post K. 2007 A high diversity in fossil beaked whales (Mammalia, Odontoceti, Ziphiidae) recovered by trawling from the sea floor off South Africa. *Geodiversitas* **29**, 561–618.
21. Jolicoeur P. 1963 193. Note: the multivariate generalization of the allometry equation. *Biometrics* **19**, 497–499.
22. Marcus LF. 1990 Chapter 4: Traditional Morphometrics. In *Proceedings of the Michigan morphometrics workshop* (eds FJ Rohlf, FL Bookstein), pp. 77–122. Michigan: University of Michigan Museum of Zoology.
23. R Core Team. 2019 *R: A language and environment for statistical computing*. Vienna, Austria: R Foundation for Statistical Computing. See <https://www.R-project.org/>.
24. Mead JG, Fordyce RE. 2009 *The Therian Skull: a Lexicon with Emphasis on the Odontocetes*. Washington D.C.: Smithsonian Institution Scholarly Press.
25. Fitzgerald EM. 2016 A late Oligocene waipatiid dolphin (Odontoceti: Waipatiidae) from Victoria, Australia. *Memoirs of Museum Victoria* **74**, 117–136.
26. Marx F, Lambert O, Uhen MD. 2016 *Cetacean Paleobiology*. Chichester: John Wiley & Sons.
27. Reidenberg JS, Laitman JT. 1994 Anatomy of the hyoid apparatus in odontoceli (toothed whales): Specializations of their skeleton and musculature compared with those of terrestrial mammals. *The Anatomical Record* **240**, 598–624. (doi:10.1002/ar.1092400417)
28. Rasmussen LB. 1966 Molluscan faunas and biostratigraphy of the marine younger Miocene formations in Denmark. Part I: Geology and biostratigraphy. *Geological survey of Denmark* **88**, 1–358.
29. Piasecki S. 2005 Dinoflagellate cysts of the Middle-Upper Miocene Gram Formation, Denmark. *Palaeontos* **7**, 29–45.
30. Beyer C. 2005 A magnetic analysis of the Late Miocene Gram Formation, Denmark. *Palaeontos* **7**, 19–28.
31. Lambert O. 2005 Systematics and phylogeny of the fossil beaked whales *Ziphirostrum du Bus*, 1868 and *Choneziphius Duvernoy*, 1851 (Mammalia, Cetacea, Odontoceti) from the Neogene of Antwerp (North of Belgium). *Geodiversitas* **27**, 443–497.
32. Lambert O, Louwye S. 2006 *Archaeoziphius microglenoideus*, a new primitive beaked whale (Mammalia, Cetacea, odontoceti) from the Middle Miocene of Belgium. *Journal of Vertebrate Paleontology* **26**, 182–191. (doi:10.1671/0272-4634(2006)26[182:AMANPB]2.0.CO;2)
33. Bianucci G, Miján I, Lambert O, Post K, Mateus O. 2013 Bizarre fossil beaked whales (Odontoceti, Ziphiidae) fished from the Atlantic Ocean floor off the Iberian Peninsula. *Geodiversitas* **35**, 105–153.
34. Barnes LG. 1976 Outline of eastern North Pacific fossil cetacean assemblages. *Systematic Zoology* **25**, 321–343.
35. Fordyce RE, de Muizon C. 2001 Evolutionary history of cetaceans: a review. Secondary Adaptation of Tetrapods to Life in Water. In *Secondary Adaptation of Tetrapods to Life in Water* (eds J-M Mazin, V de Buffrénil), pp. 169–212. München.
36. Galatius A, Kinze CC. 2003 Ankylosis patterns in the postcranial skeleton and hyoid bones of the harbour porpoise (*Phocoena phocoena*) in the Baltic and North Sea. *Canadian Journal of Zoology* **81**, 1851–1861.
37. Bianucci G, Collareta A, Post K, Varola A, Lambert O. 2016 A New Record of *Messapicetus* from the Pietra Leccese (Late Miocene, Southern Italy): Antitropical Distribution in a Fossil Beaked Whale (Cetacea, Ziphiidae). *Rivista Italiana di Paleontologia Stratigrafia* **122**, 63–73.
38. Dalebout ML, Mead JG, Baker CS, Baker AN, van Helden AL. 2002 A New Species of beaked whale *Mesoplodon perini* sp. n. (Cetacea: Ziphiidae) discovered through Phylogenetic Analyses of Mitochondrial DNA sequences. *Marine Mammal Science* **18**, 577–608.
39. Cope ED. 1869 Two extinct Mammalia from the United States. *Proceedings of the American Philosophical Society* **11**, 188–190.
40. Flower WH. 1882 On the Whales of the Genus *Hyperoodon*. In *Proceedings of the Zoological Society of London*, pp. 722–726. Oxford, UK: Blackwell Publishing Ltd.
41. Boschma H. 1951 Rows of small teeth in ziphioid whales. *Zoologische Mededelingen* **31**, 130–148.
42. Lambert O, Collareta A, Landini W, Post K, Ramassamy B, Di Celma C, Urbina M, Bianucci G. 2015 No deep diving: evidence of predation on epipelagic fish for a stem beaked whale from the Late Miocene of Peru. *Proceedings of the Royal Society B: Biological Sciences* **282**, 20151530. (doi:10.1098/rspb.2015.1530)

43. Benke H. 1993 Investigations on the osteology and the functional morphology of the flipper of whales and dolphins (Cetacea). *Investigations on Cetacea* **24**, 9–252.
44. Ross JGB. 1984 The smaller cetaceans of the south east coast of Southern Africa. *Annals of the Cape Provincial Museums* **15**, 173–410.
45. Werth AJ. 2006 Mandibular and Dental Variation and the Evolution of Suction Feeding in Odontoceti. *Journal of Mammalogy* **87**, 579–588. (doi:10.1644/05-MAMM-A-279R1.1)
46. Mead JG, Payne RS. 1975 A specimen of the Tasman beaked whale, *Tasmacetus shepherdi*, from Argentina. *Journal of Mammalogy*, 213–218.
47. Heyning JE, Ridgway SH, Harrison R. 1989 Cuvier's beaked whale *Ziphius cavirostris* G. Cuvier, 1823. In *Handbook of marine mammals*, pp. 289–320. London: Academic Press.
48. Mead JG. 1989 Bottlenose whales *Hyperoodon ampullatus* (Forster, 1770) and *Hyperoodon planifrons* Flower 1882. In *Handbook of marine mammals* (eds SH Ridgway, R Harrison), pp. 321–348. London: Academic Press.
49. Mead JG. 1989 Beaked whales of the genus *Mesoplodon*. In *Handbook of marine mammals* (eds SH Ridgway, R Harrison), pp. 349–430. London: Academic Press.
50. MacLeod CD. 2000 Distribution of beaked whales of the genus *Mesoplodon* in the North Atlantic. *Mammal Review* **30**, 1–8.
51. Clarke MR. 1986 Cephalopods in the diets of odontocetes. In *Research on Dolphins* (eds MM Bryden, R Harrison), pp. 281–321. Oxford: Clarendon Press.
52. Bianucci G, Lambert O, Post K. 2008 Beaked whale mysteries revealed by seafloor fossils trawled off South Africa. *South African Journal of Science* **104**.
53. de Muizon C. 1984 *Les vertébrés fossiles de la Formation Pisco (Pérou). deuxième partie: les Odontocètes (Cetacea, Mammalia) du Pliocène inférieur de Sud-Sacaco*. Travaux de l'Institut Français d'Etudes Andines.
54. Dalebout ML, Steel D, Baker CS. 2008 Phylogeny of the Beaked Whale Genus *Mesoplodon* (Ziphiidae: Cetacea) Revealed by Nuclear Introns: Implications for the Evolution of Male Tusks. *Systematic Biology* **57**, 857–875. (doi:10.1080/10635150802559257)

Tables

TABLE 1. Measurements of the mandible, cranial remains, and forelimb bones of the specimen genus and sp. indet. NHMD 189993.

Feature	NHMD 189993
Mandibles	
Anteroposterior length as preserved	1032
Maximum posterior width as preserved	412
Symphyseal portion anteroposterior length	289
Symphyseal portion maximal transverse width	61
Humerus	
Anteroposterior humeral head diameter	73
Maximal humeral length	204
Maximal distal width	80
Width at the level of the deltoid ridge	92
Radius	
Maximal radius length	186
Width at mid-length	63
Proximal width	67
Distal width	73
Measurements in mm	

TABLE 2. Measurements of the periotic and the tympanic bone of genus and sp. indet. NHMD 189993.

Feature	NHMD 189993
Right Periotic	
Maximal anteroposterior length	47
Maximal transverse width	33

Pars cochlearis maximum anteroposterior length	25
Pars cochlearis maximum transverse width	27
Anterior process maximum anteroposterior length	21
Anterior process maximum transverse width	20
Posterior process maximum anteroposterior length	18
Posterior process maximum transverse width	21
Lateral tuberosity in lateral view transverse width	10
Right Tympanic bulla	
Tympanic anteroposterior length	43
Tympanic maximum transverse width	26
Inner posterior prominence maximum transverse width	11
Outer posterior prominence maximum width	15
Dorsoventral height in lateral view as preserved	21
Involucrum indentation on the tympanic	
Dorsoventral height in medial view	12

All measurements are in mm and taken in ventral view unless noted otherwise.

Figures

Figure 1. Current extension of the Gram Formation in Denmark (shaded area). The type locality is situated in the Gram claypit, 1.5 km north of Gram. Modified from Rasmussen [28].

Figure 2. Mandibles, cranial, and postcranial remains of *genus and sp. indet.*, NHMD 189993. **A**, ventral view; **B**, corresponding drawing; **C**, detail of the preserved nasal and other postcranial remains.

Figure 3. Digital reconstruction of the mandible and postcranial remains of *genus and sp. indet.*, NHMD 189993. **A**, dorsal view; **B**, lateral view; **C**, detail of the anterodorsal part of the mandible.

Figure 4. Right periotic of *genus and sp. indet.*, NHMD 189993. **A**, dorsal view; **B**, lateral view; **C**, medial view; **D**, ventral view; **E-H**, corresponding drawings.

Figure 5. Right tympanic of *genus and sp. indet.*, NHMD 189993. **A**, dorsal view; **B**, medial view; **C**, ventral view; **D**, lateral view.

Figure 6. Postcranial elements of *genus and sp. indet.*, NHMD 189993. Right humerus in **A**, lateral view; **B**, dorsal view; **C**, posterior view. Associated right radius **D**, in lateral view; **E**, medial view; **F**, dorsal view; isolated tusk in **G**, lateral view; **H**, ventral view; three isolated teeth in **I**, **K**, **M**, medial view; **J**, **L**, **N**, lateral view; right stylohyal in **A**, lateral view; **B**, medial view; **C**, dorsal view.

Figure 7. Log-transformed bizygomatic width plotted against condylobasal length in Ziphiidae. **Abbreviations:** **Bear:** *Berardius arnuxii*; **Beba:** *Berardius bairdii*; **Hyam:** *Hyperoodon ampullatus*; **HypI:** *Hyperoodon planifrons*; **Megr:** *Mesoplodon grayi*; **Mela:** *Mes. layardii*; **Mepe:** *M. peruvianus*; **Messgr:** *Messapicetus gregarius*; **Naur:** *Nazacetus urbanai*; **Nipl:** *Ninoziphius platyrostris*; **Tash:** *Tasmacetus shepherdii*; **Zica:** *Ziphius cavirostris*. The cluster of medium ziphiids corresponds to the species *M. bidens*, *M. europaeus*, *M. ginkgodens*, and *M. mirus*.

Supplementary Data

Supplementary data S1. Condylobasal lengths and postorbital widths from specimens of Ziphiidae used in the analysis.

Supplementary data S2. Pdf file containing a 3D reconstruction of the *genus and sp. indet.*, NHMD 189993.

1
2
3
4
5
6
7
8
9
10
11
12
13
14
15
16
17
18
19
20
21
22
23
24
25
26
27
28
29
30
31
32
33
34
35
36
37
38
39
40
41
42
43
44
45
46
47
48
49
50
51
52
53
54
55
56
57
58
59
60

Figure 1. Current extension of the Gram Formation in Denmark (shaded area). The type locality is situated in the Gram claypit, 1.5 km north of Gram. Modified from Rasmussen [28].

Figure 2. Mandibles, cranial, and postcranial remains of genus and sp. indet. NHMD 189993. A, ventral view; B, corresponding drawing; C. detail of the preserved nasal and other postcranial remains.

Figure 3. Digital reconstruction of the mandible and postcranial remains of genus and sp. Indet. NHMD 189993. A, dorsal view; B, lateral view; C, detail of the anterodorsal part of the mandible.

Figure 4. Right periotic of genus and sp. indet. NHMD 189993. A, dorsal view; B, lateral view; C, medial view; D, ventral view; E-H, corresponding drawings.

Figure 5. Right tympanic of genus and sp. indet. NHMD 189993. A, dorsal view; B, medial view; C, ventral view; D, lateral view.

Figure 6. Postcranial elements of genus and sp. indet. NHMD 189993. Right humerus in A, lateral view; B, dorsal view; C, posterior view. Associated right radius D, in lateral view; E, medial view; F, dorsal view; isolated tusk in G, lateral view; H, ventral view; three isolated teeth in I, K, M, medial view; J, L, N, lateral view; right stylohyal in A, lateral view; B, medial view; C, dorsal view.

35 Figure 7. Log-transformed bizygomatic width plotted against condylobasal length in Ziphiidae.
36 Abbreviations: Bear: *Berardius arnuxii*; Beba: *Berardius bairdii*; Hyam: *Hyperoodon ampullatus*; Hypl:
37 *Hyperoodon planifrons*; Megr: *Mesoplodon grayi*; Mela: *Mes. layardii*; Mepe: *M. peruvianus*; Messgr:
38 *Messapicetus gregarius*; Naur: *Nazcacetus urbinai*; Nipl: *Ninoziphius platyrostris*; Tash: *Tasmacetus*
39 *shepherdii*; Zica: *Ziphius cavirostris*. The cluster of medium ziphiids corresponds to the species *M. bidens*, *M.*
40 *europaeus*, *M. ginkgodens*, and *M. mirus*.

Appendix B**ROYAL SOCIETY
OPEN SCIENCE****A new specimen of Ziphiidae (Cetacea, Odontoceti) from the late Miocene of Denmark with morphological evidences for suction feeding behaviour**

Journal:	Royal Society Open Science
Manuscript ID	RSOS-191347
Article Type:	Research
Date Submitted by the Author:	03-Aug-2019
Complete List of Authors:	Ramassamy, Benjamin; Museum of Southern Jutland, Department of Natural History and Palaeontology; Lauridsen, Henrik; Aarhus University Hospital Skejby, The Department of Clinical Medicine, Comparative Medicine Lab
Subject:	palaeontology < BIOLOGY, taxonomy and systematics < BIOLOGY, evolution < BIOLOGY
Keywords:	Feeding Strategy, Ecological niche, Systematics, Ziphiidae
Subject Category:	Biology (whole organism)

Author-supplied statements

Relevant information will appear here if provided.

Ethics

Does your article include research that required ethical approval or permits?:

This article does not present research with ethical considerations

Statement (if applicable):

CUST_IF_YES_ETHICS :No data available.

Data

It is a condition of publication that data, code and materials supporting your paper are made publicly available. Does your paper present new data?:

Yes

Statement (if applicable):

CUST_IF_YES_DATA :No data available.

Conflict of interest

I/We declare we have no competing interests

Statement (if applicable):

CUST_STATE_CONFLICT :No data available.

Authors' contributions

This paper has multiple authors and our individual contributions were as below

Statement (if applicable):

B.R. identified and interpreted the ziphiid remains. H.L. created the digital reconstruction and the associated figures. B.R. made calculations for the size estimates and collected the measurements. Both authors participated in the preparation of the illustrations and the writing of the manuscript. Both discussed the results and commented on the manuscript at all stages.

A new specimen of Ziphiidae (Cetacea, Odontoceti) from the late Miocene of Denmark with morphological evidences for suction feeding behaviour

BENJAMIN RAMASSAMY*,¹ and HENRIK LAURIDSEN²

¹Department of Natural History and Palaeontology, the Museum of Southern Jutland, Lergravsvej 2, Gram, 6510, Denmark;

²Comparative Medicine Lab, Department of Clinical Medicine, Aarhus University, Palle Juul-Jensens Boulevard 99, Aarhus, 8200, Denmark;

Keywords: Feeding Strategy; Ecological niche; Systematics; Ziphiidae

1. Summary

A new fossil of Ziphiidae from the upper Miocene Gram Formation (ca. 9.9-7.2 Ma) is described. Computed Tomographic scanning of the specimen was performed to visualize the mandibles and to obtain a digital reconstruction. It possesses several derived characters in the Ziphiidae, such as the dorsoventral thickening of the anterior process of the periotic, the dorsoventral compression of the pars cochlearis, and the short unfused symphysis. The specimen cannot be identified beyond family level, because of the unusual nature of the preserved parts consisting in mandibles, earbones and postcranial remains. It differs from the other species of ziphiid from the Gram Formation, *Dagonodum mojnium*, based on its larger size and the more derived morphology of its mandibles and earbones. Its long and thickened stylohyal, combined with its teeth reduction suggests that this new specimen relied primarily on suction feeding. By contrast, the other ziphiid species from the Gram Formation, *D. mojnium*, was adapted to a more raptorial feeding strategy. Assuming the two species were chronologically concomitant, their co-occurrence at the same locality with two different feeding strategies, may represent a case of niche separation. They may have hunted different types of prey, thus avoiding competition for a same food resource.

2. Introduction

Beaked whales (Ziphiidae) represent a diversified family of echolocating toothed whales (Odontoceti), currently represented by at least 22 species in 6 genera [1] with a potential new species of *Berardius* suspected in the North Pacific [2]. Their best-known modern representatives are capable of regular deep dives beyond 1000 meters to reach their foraging grounds, where they prey mostly on cephalopods and more occasionally on bathypelagic fish and crustaceans [3–10]. Most extant ziphiids are typified by a strong reduction of their teeth count reduced to one or two pairs, often only erupted in adult males [11]. Beaked whales do not use them to capture or manipulate their prey; instead, they use suction feeding as their main feeding strategy, except perhaps for the toothed ziphiid *Tasmacetus shepherdi* which retains a set of functional teeth [4]. Suction feeding forces them to be more selective relative to the size of their prey, thus allowing different species of beaked whales to be sympatric without competing for the same food resource [12,13].

Recently, Hocking et al. [14] proposed a new framework to understand the evolution of feeding in predatory aquatic mammals. Instead of thinking the different feeding styles as rigid categories, they argue that feeding strategies of aquatic mammals follow a particular evolutionary sequence that can be used to predict the origin of particular feeding styles. Under this framework, the specialization to suction feeding of extant beaked whales should arise from ancestors using a more raptorial feeding strategy. The fossil record of Ziphiidae confirm this prediction: some of the most basal beaked whales possessed elongated jaws and numerous functional interlocking teeth potentially used to capture their prey [15–18].

*Author for correspondence (benjamin.ramassamy@laposte.net).

†Present address: Department of Natural History and Palaeontology, the Museum of Southern Jutland, Lergravsvej 2, Gram, 6510, Denmark

However, morphological evidences suggest that some of them were also capable to use suction feeding at least in the most posterior part of the mandibles [17,18]. *D. mojunum*, a new fossil genus and species with close affinities to *M. gregarius*, was described from the Gram Formation in Denmark [18]. This species was interpreted as a more raptorial feeder than extant ziphiids based on its numerous interlocking teeth and elongated jaws, despite moderate adaptations to suction feeding.

A new fossil Ziphiidae from the same locality is described here. The preserved parts of the specimen consist in the lower jaw, earbones, part of the hyoid apparatus, and forelimb elements. This project aims at describing the specimen and at proposing a palaeoecological reconstruction based on morphological features. Elements relative to feeding strategies and the ecological niches occupied by the ziphiids from the Gram Formation are discussed.

3. Materials and Methods

3.1 Specimen Preparation and Computed Tomography

The specimen was discovered in 2007 and prepared by means of mechanical tools at the curatorial department of the Museum of Southern Jutland. A co-polymer of acrylates (MA/EMA Paraloid B72) was used as an adhesive to keep the fragments of the lower jaw together. Photos of the specimen were taken using a Fujifilm FinePix HS10 with a focal length of 4.2-126.0 mm.

Specimens coming from the Gram Formation are fragile, difficult to handle, and prepare. Furthermore, the preparation sometimes results in the loss of information in relation with the original placement of the bone structures. To alleviate the preparation work and avoid extensive manipulation of the specimen, the lower jaw was scanned using a clinical computed tomography (CT) system (Siemens Somatom; Siemens Medical Solutions, Forchheim, Germany) using the following parameters: $0.98 \times 0.98 \times 0.60$ mm³ voxel size; 140 kVp tube voltage; 185 μ As tube charge, resulting in an acquisition time of ~60 s. Data was reconstructed using a B45s convolution kernel. The CT-scans unveiled the dorsal and lateral side of the lower jaw as preserved that otherwise, would have not been accessible without extensive preparation. Visualizations of the scanned fossil were made in the DICOM-viewer OsiriX (Pixmeo SARL) and image segmentation and construction of an interactive model of the fossil was made in Amira 5.6 (FEI, Visualization Sciences Group). The digital reconstruction is available in the Supplementary Data (Supplementary Figure S1).

3.2 Size Estimation and Evaluation of Trophic level

Cetaceans, particularly obligate suction feeders, are known to select their prey relative to their own size [12]. Therefore, assessing the size of a ziphiid individual and comparing it with other species may help estimating the trophic level at which the specimen fed.

To do so, two cranial measurements were collected from different ziphiids specimens: the bizygomatic width and the condylobasal length (available in Supplementary Dataset S2). Many fossil forms had to be discarded, because their partial skull did not allow a good estimation of the condylobasal length and/or the bizygomatic width. The fossil species *Ninoziphius platyrostris*, *Nazcacetus urbinai*, and *Messapicetus gregarius* were included based on the measurements provided in their respective descriptions [16,17,19]. In absence of a preserved skull for the specimen NHMD 189993 described herein, such measurements were not available. The anteroposterior length and posterior transverse width of the mandibles were used instead of, respectively, the bizygomatic width and the condylobasal length. The posterior transverse width of the specimen is a good estimator of the bizygomatic width but in this case, is slightly underestimated due to the lack of the most posterior parts of the mandibles. Anteroposterior length of the mandibles is significantly shorter than bizygomatic width in odontocetes and is therefore only an indicator of minimum size rather than a precise estimator. Cranial measurements were selected for other ziphiids, because the mandibles of beaked whales are often disarticulated and the posterior width of the mandibles is therefore not always measurable.

Ziphiid species were placed in four size categories: very large-sized ziphiids (8-10 m), large-sized ziphiids (5.5-7.5 m), medium-sized ziphiids (4-4.5 m), and small-sized ziphiids (3-4 m). Those categories were defined in Bianucci et al. [20] based on a regression of the postorbital width relative to the body length of different ziphiid species.

A natural logarithmic transformation was applied to the cranial measurements to attenuate the effect of allometry and correct for heteroscedasticity [21,22]. A MANOVA (Multivariate analysis of variance) was performed to evaluate whether the cranial measurements were sufficient to assess each size category. It was followed by a Tukey's HSD (honest significant difference) test on each variable to compare differences between the size categories. Linear regression was also performed on the dataset to assess the relationship between the two cranial measurements. All Analyses were performed with the software R 3.6.0 [23].

3.3 Nomenclature

Institutional Abbreviations—**IRSNB**, Institut Royal des Sciences Naturelles de Belgique, Brussels, Belgium; **MNHN**, Muséum National d'Histoire Naturelle, Paris, France; **MSM**, Museum Sønderjylland Naturhistorie og Palæontologi, Gram

Lergrav, Gram, Denmark; **MSNUP**, Museo di Storia Naturale dell'Università di Pisa, Italy; **MUSM**, Museo de Historia Natural, Lima, Peru; **NMNZ**, National Museum of New Zealand Te Papa Tongarewa, Wellington, New Zealand; **NHMD**, Statens Naturhistoriske Museum, Copenhagen, Denmark; **USNM**, United States National Museum of Natural History, Smithsonian Institute, Washington D.C., U.S.A.

Terminology—The anatomical terminology of the skull and earbones follows Mead and Fordyce [24]. The terminology used by Fitzgerald [25] and Marx et al. [26] was followed for the postcranial remains. The nomenclature of Reidenberg and Laitman [27] was used for describing elements of the hyoid apparatus.

4. Results

4.1 Systematic Palaeontology

Order CETACEA Brisson, 1762
Suborder ODONTOCETI Flower, 1867
Family ZIPHIIDAE Gray, 1850
Genus and species indet.

Referred Material—NHMD 189993, subcomplete mandibles, the right stylohyal, 14 isolated teeth including a tusk, two periotics and the right tympanic, the right humerus and associated radius, parts of the nasal.

Horizon and Locality—The locality is situated 1.5 km north of the town of Gram, Southern Jutland, Denmark (55°18'02.67"N, 9°30'32.51"E; Fig. 1). The specimen was dated based on the mollusc fauna identified in association with genus and sp. indet. NHMD 189993. The high percentage of *Carinastarte vetula reimersi* (55% of the specimens identified) and the co-occurrence of the species *Gemmula badensis* and *Turitella tricarinata* suggest that the specimen was originally found in the assemblage Zone V [28].

The assemblage Zone V belongs to the upper part of the Gram Formation dated from the Tortonian age, based on the co-occurrence of the dinoflagellate cysts *Hystichosphaeris obscura*, *Spiniferis solidago*, and *Labyrinthodinium truncatum* [29]. The minimum age of the Assemblage Zones from the Gram Formation is estimated to 9.9 Ma based on the presence of a polarity zone of less than 70 000 years [30]. The specimen genus and sp. indet. NHMD 189993 can be dated from the mid- to late Tortonian, ca. 9.9–7.2 Ma.

Systematic Attribution of the Specimen—The specimen is identified as a member of the family Ziphiidae based on the following combination of characters: the enlargement of the apical or subapical mandibular tooth; the reduction of the dorsal keel on the posterior process of the periotic; the mediolateral thickening of the anterior process of the periotic; in dorsal view, the anterior shift of the pars cochlearis of the periotic.

Genus and sp. indet. NHMD 189993 clearly differs from the other species found in the Gram Formation, *Dagonodum mojnium*, based on the following combination of characters: the reduction of the mandibular teeth, the shorter unfused symphysis, the dorsoventral thickening of the anterior process of the periotic, the dorsoventral compression of the pars cochlearis, the presence of a cochlear spine.

Identification beyond the genus was not possible, because of the unusual nature of the preserved material. Most fossil of ziphiids are represented by cranial remains, mostly the rostral, preauricular, and vertex region [15,20,31–33]. Mandibles, earbones, and postcranial remains are more rarely preserved. Despite the unusual features present on the periotics (presence of a cochlear spine, depression along the medial surface of the posterior process) and the peculiar size of the specimen, the lack of cranial remains makes it nearly impossible to compare it with many similarly sized ziphiids only known from the preauricular region (e.g. *Africanacetes*, *Globicetus*, *Tusciziphius*). Genus and sp. indet. NHMD 189993 possesses several derived features thought to be characteristic of a crown Ziphiidae [e.g. 16,33,33]: the dorsoventral thickening of the anterior process of the periotic bone, the dorsoventral compression of the pars cochlearis, the short and unfused symphysis. However, a recent phylogenetic analysis proposed that some members of the more basal *Messapicetus* clade displayed derived characters indicative of a convergent evolution between stem and crown Ziphiidae (Bianucci et al., 2016a). Mandibles, earbones and postcranial material of the most derived members of this clade, *Globicetus*, *Tusciziphius*, and *Imocetus* are not known [33]. It is therefore impossible to assess whether genus and sp. indet. NHMD 189993 was a crown ziphiid or a member of the *Messapicetus* clade.

By measure of caution, the advice of Barnes [34] and Fordyce and Muizon [35], which suggest that the identification of a new species should at least include skull and rostrum, is followed until more cranial material is available.

4.2 Description and Comparisons

4.2.1 Cranium and Mandible

Overview and ontogeny—The specimen is interpreted as an adult based on the complete fusion of the humeral head to the humeral shaft and the epiphyseal ankylosis of each epiphysis of the radius. In the porpoise *Phocoena phocoena*, extensive ankylosis of the postcranial skeleton characterizes adult specimens [36].

The most robust parts of the mandibles and postcranial elements of ~~genus and sp. indet~~ NHMD 189993 are well preserved compared to the more fragmentary cranial remains and ribs (Fig. 2-3). The earbones were found with the lower jaw, the right periotic still having the stapes firmly attached to it (Fig. 4-5; only right periotic illustrated). The humerus and radius were originally still articulated (Fig. 6A-F), whereas the stylohyal lied along the right lateral side of the symphysis (Fig. 6O-Q). Teeth were collected around the specimen out of their mandibular sockets (Fig. 6G-N). The preserved parts of the mandibles are 1032 mm long and 412 mm wide. More measurements of the specimen are available in Table 1 and 2.

Nasal—Because the nasal is the only piece identifiable from the shattered cranium of the specimen, its orientation reveals difficult. The dorsal exposure is flat and rectangular (Fig. 2C). The ratio between the width and length of the visible surface is 0.70 (60 mm long and 86 mm wide). No excavation is visible on the surface of the nasal bone.

Periotic—Measurements of the right periotic are available in Table 2. The anterior process of the periotic is transversely thickened, with a large rounded protuberance along the dorsomedial surface of the periotic (Fig. 4A-C). This strong thickening both lateromedial and dorsoventral, is observed in all crown ziphiids, but not in the stem-ziphiids *Dagonodum mojnum*, *Messapicetus gregarius*, and *Ninoziphius platyrostris* [16–18]. In the latter, the thickening occurs only lateromedially. The tip of the anterior process is pointed. In ventral view, the anterior bullar facet is anteroposteriorly elongated and elliptical. Posteromedially to this facet, the accessory ossicle is still articulated in the fovea epitubaria (Fig. 4D). It extends along the dorsomedial margin of the anterior process. It is less rounded and developed than in *Berardius*, *Hyperoodon*, some species of *Mesoplodon* (*M. carlhubbsi*, *M. europaeus*, *M. grayi*, *M. mirus*), *Nazcacetus*, and *Tasmacetus*. In ventral view, a sulcus extends anteroposteriorly along the accessory ossicle, and separates it in two (Figure 4D). A similar sulcus is also observed in *H. ampullatus*, *M. densirostris*, and *T. shepherdi*, although less developed in those species. The sulcus observed in ~~genus and sp. indet~~ NHMD 189993 could refer to the origin of the tendon of m. tensor tympani [24]. Posteriorly to the fovea epitubaria and the accessory ossicle, the malleolar fossa develop along the medial margin of the lateral tuberosity. In ventral view, the anterior process is separated from the lateral tuberosity by the anteroexternal sulcus, which can also be seen in lateral view. The anteroexternal sulcus is also present in the periotic of *D. mojnum*. In ventral view, the lateral tuberosity is lateromedially elongated, a character observed in all ziphiids, except *D. mojnum*, *M. gregarius*, and *N. platyrostris* [16–18]. The fenestra ovalis is rounded. Posteroventrally to the fenestra ovalis, a deep hiatus epitympanic separates the posterior process from the lateral tuberosity. In ventral view, the posterior process of the periotic is fan-shaped: it is rounded and widens abruptly from anterior to posterior (Fig. 4D). A fan-shaped posterior bullar facet is characteristic of all ziphiids, except *D. mojnum*, *N. platyrostris*, and *M. gregarius* [16–18]. In medial view, the posterior process is oriented posteroventrally. ~~Genus and sp. indet~~ NHMD 189993 lacks a distinct keel along the whole posterior process, a feature present in all ziphiids [17]. A deep depression excavates the anteromedial side of the posterior process, just posterior to the pars cochlearis (Fig. 4C). This depression seems unique to ~~genus and sp. indet~~ NHMD 189993 and was not observed in the periotic bone of other ziphiids.

In dorsolateral view, the pars cochlearis is anteriorly shifted, a feature that distinguishes ~~an eurhinodelphid from a ziphiid~~ periotic (Fig. 4A) [16]. In ventromedial view, the pars cochlearis is rectangular, because of its straight anteromedial corner. It is also dorsoventrally compressed, a feature observed in crown ziphiids, but absent in *N. platyrostris* and members of the *Messapicetus* clade [17]. In ventral view, the pars cochlearis bears a triangular depression similar in shape to *D. mojnum* [18]. This depression is also visible in ~~the species~~ *Mesoplodon mirus* (USNM 504612, USNM 550351, USNM 572961, USNM KLC112) and *M. bidens* (MNHN 1975.112, SNM CN5x), but is elliptical, more elongated anteroposteriorly in both species. The tear-shaped fenestra rotunda is oriented posteroventrally. Posterodorsally to the internal acoustic meatus, the periotic bears a large cochlear spine. This unusual feature in Ziphiidae is present in ~~the species~~ *N. platyrostris* and *Berardius armuxii* [17]. Its presence was also observed in ~~the species~~ *B. bairdii* (USNM 571524). The cochlear spine in ~~genus and sp. indet~~ NHMD 189993 is moderately developed dorsally, a condition similar to ~~the genus~~ *Berardius* and differing from the well-marked cochlear spine of *N. platyrostris*. In dorsomedial view, the internal acoustic meatus is elliptical; this feature is also observed in ~~the periotic of~~ *N. platyrostris* and is connected to the presence of the cochlear spine. A thick crest separates the internal acoustic meatus from the aperture for the vestibular aqueduct (Fig. 4C, G). Inside the internal acoustic meatus, the dorsal vestibular meatus is separated from its ventral counterpart by a transverse crest. The ventral vestibular area occupies almost two thirds of the surface of the internal acoustic meatus. Posteriorly to the vestibular area of the internal acoustic meatus, the aperture for the vestibular aqueduct is anteroposteriorly compressed. Ventrally to the vestibular aqueduct, the aperture for the cochlear aqueduct is reduced to a small opening (Fig. 4C).

Tympanic bulla—The right tympanic bulla is partially preserved (Fig. 5). ~~Further measurements~~ are available in Table 2. It lacks the base of the pedicle, the sigmoid process, and the dorsal part of the outer lip. In ventral view, the bulla is heart-shaped, because of the interprominential notch well marked posteriorly that separates the inner and outer posterior prominences. ~~In ventral view,~~ the inner posterior prominence is compressed transversely and is less developed posteriorly than the twice-larger outer posterior prominence (Fig. 5C). The compression of the inner posterior prominence is similar to some species of *Mesoplodon* (e.g. *M. bidens* SNM CN5x, *M. bowdoini* NMNZ MM2653, *M. europaeus* USNM 504349), *Messapicetus gregarius*, and *N. platyrostris*. In *Hyperoodon* spp. and *Z. cavirostris*, the inner posterior prominence is more reduced and is even shorter posteriorly. In ventral view, the interprominential notch connects to the deep median furrow. The median furrow extends roughly until the first third of the bulla (Fig. 5C). The median furrow is less developed in *Hyperoodon* spp. and *Z. cavirostris*, but more extended in the stem ziphiids *D. mojnum*, *M. gregarius*, and *N. platyrostris*. In ventral view, a keel extends along the whole anteroposterior length of the bulla.

The involucre is indented, a feature visible both in dorsal and medial view (Fig. 5A-B). In medial view, the ventral part of the bulla is incurved, but does not reach the dorsalmost margin of the posterior portion of the involucre, as in *D. mojnum* (Fig. 5B). The anterior margin of the tympanic bulla is too damaged to assess the degree of development of the

1 tympanic spine, if present. However, the broken anterolateral margin of the bulla develops anteriorly into a thin bone plate
2 (Fig. 5A), a condition similar to *N. platyrostris*, where the tympanic spine is absent [17].

3 **Stapes**—The right stapes is still firmly attached to the periotic in the fenestra ovalis and could not be removed (Fig. 4C).
4 The stapes is conical, widening at its oval base, as observed in several ziphiids species (Lambert et al., 2009). The head of
5 the stapes has a circular outline. The small and circular vestigial stapelial foramen opening is situated approximately at
6 mid-length of the stapes. The muscular process is well developed and situated at the level of the head of the stapes.

7 **Mandible**—The mandible of ~~genus and sp. indet. NHMD 189993~~ lacks the posterior part of the acoustic window and the
8 **mandibular condyle** (Fig. 2-3). Further measurements are available in Table 1. The symphyseal portion of the mandible is
9 unfused (Fig. 3C). It is not ankylosed as in the long snouted stem ziphiids *Ninoziphius platyrostris*, *Messapicetus* spp., and
10 *Dagonodum mojnum* [16–18,37]. The symphysis is 289 mm long and represents at most 28 percent of the total length of
11 the mandible (the total length of the preserved parts is 1032 mm). This value is much lower than in long snouted ziphiids
12 and *Tasmacetus shepherdii*, where the symphysis extends at least along 36 percent of the mandible total length [17]. The
13 **symphyseal portion** of ~~genus and sp. indet. NHMD 189993~~ is triangular, differing from the half-circled section of *Berardius*
14 spp., *D. mojnum*, *Messapicetus* spp., *N. platyrostris*, and *Tasmacetus shepherdii* [16–18]. The symphyseal portion of the
15 mandible is turned upwards. This feature is also present in the species *H. ampullatus*, *M. bidens*, *M. grayi*, *M. mirus*, *N.*
16 *urbinaei*, and *Z. cavirostris* [19].

17 The apex of the mandibles is heavily fractured, but the fragments conserved their original position, allowing an estimation
18 of the original outline. In ventral view, the apex is rounded and likely possessed an enlarged alveolus for the tusk. This
19 interpretation fits with the shape and size of the preserved tusk that is similar to those of several long snouted stem beaked
20 whales (*D. mojnum*, *Mess. gregarius*) possessing a pair of tusks in apical position (Fig. 6G-H). Furthermore, no other
21 alveolus along the alveolar groove is sufficiently developed to support the tusk. The apex of the mandible is too fractured
22 to identify precisely whether the tusk was positioned apically or subapically, as observed by Dalebout et al. [38] in
23 *Mesoplodon perrini*. It is also possible that ~~genus and sp. indet. NHMD 189993~~ possessed two pairs of tusks, even though
24 only one tusk is preserved with the specimen (Fig. 6G-H). This character is observed in *Berardius* spp., *Anoploussa*
25 *forcipata*, and *D. mojnum* [18,39]. One mental foramen is visible along the lateral side of the mandible. It is elongated,
26 well-marked and situated slightly posterior to the symphysis.

27 The outline of individualized alveoli can be distinguished in the alveolar border (Fig. 3C). The number of counted alveoli
28 along the left dentary is 17, which is probably a slight underestimation, because of the eroded and fractured surface of the
29 alveolar border of the most apical parts of the symphysis. It is not possible to assess the presence of a diastema between the
30 tusk and the rest of the alveolar groove. The alveoli are oval, transversely compressed like in the *Messapicetus* spp. [15,16].
31 However, they are much more reduced than in the latter and much shallower compared to long snouted stem ziphiids and
32 ~~the species~~ *Tasmacetus shepherdii* [17,18]. In the 3D reconstruction of the lower jaw, in lateral view, the position of the
33 bone fragments posterior to the alveolar groove suggests the presence of a precoronoid crest. However, the dorsal surface
34 of the acoustic window is too fractured to be certain of it.

35 **Teeth**—Together with the mandibles of ~~genus and sp. indet. NHMD 189993~~, 14 isolated teeth were retrieved (Fig. 6G-N).
36 Their crown is approximately as developed as the root dorsoventrally and curves lingually. The crown progressively widens
37 ventrally and projects posteroventrally. The section at the base of the crown is circular, whereas the transverse section of
38 the root is more oval. An oval root is present in *Messapicetus* spp., unlike the circular section observed in *Tasmacetus*
39 *shepherdii* and the squared root observed in the species *Dagonodum mojnum* and *Ninoziphius platyrostris*. A discrete mesial
40 keel is present in some of the smallest teeth of ~~genus and sp. indet. NHMD 189993~~. Despite the presence of individualized
41 alveoli, the reduced size of the teeth and the particularly shallow alveoli suggest that the teeth of ~~genus and sp. indet. NHMD~~
42 189993 were not as robust as in other known toothed beaked whales, perhaps still embedded in the gum, as observed in
43 some specimens of extant ziphiids (e.g. *Hyperoodon ampullatus*, *Mesoplodon grayi*; [40,41]).

44 An enlarged tooth interpreted as a tusk was found with ~~genus and sp. indet. NHMD 189993~~. This tooth is more massive
45 than the other reduced teeth (Fig. 6G-H). The tusk is triangular, with a root more developed dorsoventrally than the crown.
46 The root of the tooth is transversely compressed with an oval outline (Fig. 5H). As suggested by the outline of the apex of
47 the mandible, the tusk most likely fitted in apical or subapical position of the mandible. The slightly rounded tip of the
48 crown also suggests that the tusk is slightly worn and was originally erupted. The tusk resembles those of long snouted
49 ziphiids, *Berardius* spp. and *T. shepherdii* due to their transverse compression. It differs from the apical tusk present in
50 males *Hyperoodon ampullatus* and *Ziphius cavirostris*, which is more conical. It also differs from the genus *Mesoplodon*
51 where the tusk is heavily compressed transversely, including in species where the tusk is in apical or subapical position (*M.*
52 *mirus* USNM 504612; *M. hectori* NMNZ MM0002901; *M. perrini* USNM 504260).

4.2.2 Postcranial Elements

53 **Hyoid apparatus**—The right stylohyal is 238 mm long, 48 mm wide and 26 mm thick (Fig. 6O-Q). The length of this bone
54 is almost twice longer than in *Mesoplodon layardi* (NMNZ 1899: 109 mm; NMNZ 2917: 166 mm). ~~The stylohyal length of~~
55 ~~genus and sp. indet. resemble more the one observed in~~ *Hyperoodon planifrons* (NMNZ 1806: 272 mm; NMNZ DM 1878:
56 246 mm) and *Ziphius cavirostris* (SNM CN1: 248 mm). A constriction is present on the most anterior part of the stylohyal,
57 at the level of the articulation with the epiphyal (Fig 5O). This constriction is observed in the species *Berardius arnuxii*
58 (MNHN A3244) and *Tasmacetus shepherdii* (MM 2908). In lateral view, the stylohyal progressively widens from anterior
59 to posterior and reach its maximum transverse width in the posterior part of the bone. The posterior margin of the bone that
60

articulates with the tympanohyal is pointed (Fig. 5O). The shape of the stylohyal of ~~genus and sp. indet.~~ NHMD 189993 resembles those of *Z. cavirostris* and *H. planifrons*, even though in those species, the transverse widening is more pronounced (between 29 % and 68 % wider), with a flatter dorsal surface. This shape is also observed in *Mesoplodon europaeus* [27]. It differs from the stylohyal observed in several other species of *Mesoplodon* ~~consulted~~ (*M. bidens* MNHN 1963-259, MNHN 1963-111; *M. europaeus*, NMNZ 550390; *M. layardii* NMNZ 2917), where the lateral and medial margins of the bone stay straight, without transverse widening. A ridge goes along the lateral side of the stylohyal of genus and sp. indet. NHMD 189993 (Fig. 6O), as observed in the ziphiids *Mesoplodon europaeus* and *M. mirus* [27]. This ridge gives a triangular transverse section to the bone.

Humerus—The right humerus is fully preserved (Fig. 6A-C). Further measurements are available in Table 1. It is 204 mm long and 92 mm wide at the level of the deltoid ridge. The ratio between the humeral length and the estimated bitygomatic width (posterior width of the mandible used in the specimen) is similar to *Messapicetus gregarius* (in *M. gregarius*: 0.48; ~~in genus and sp. indet.~~ NHMD 189993: 0.50). Both species display a proportionally longer humerus than most extant ziphiids [42]. The head of the humerus is hemispherical. In lateral view (Fig. 6A), the humeral head represents a quarter of the total length of the humerus. In *M. gregarius*, the head is more prominent and anterolaterally oriented: it represents almost a third of the total length of the humerus. In lateral view, the deltoid ridge is well developed along the anterior margin of the humerus (Fig 6A). It develops approximately at mid length of the humerus, and over a third of its length. The presence of a developed deltoid ridge is characteristic of extant Ziphiidae, even though it does not develop as much as in Physeteridae [43]. The posterior part of the humerus of ~~genus and sp. indet.~~ NHMD 189993 does not widen, unlike many odontocetes [43]. This feature is characteristic of the ziphiid humeri [43]. In posterior view, the articular facets for the radius and the ulna are well separated by a crest (Fig. 6C). Each facet occupies approximately half of the posterior surface of the humerus.

Radius—The associated right radius was originally found articulated with the humerus (Fig. 5D-F). Further measurements are available in Table 2. The radius curves anteroposteriorly. It measures 186 mm long, 63 mm wide at mid-length. The facet for articulation of the humerus is oriented anterodorsally (Fig. 5D-E). Posteriorly, the articulations for the scaphoid and the lunate are well defined; they occupy approximately half of the posterior width of the radius. The articulation for the scaphoid is straight in lateral view, whereas the articulation for the lunate is more oblique and face posterodorsally. In all Ziphiidae, and contrary to other odontocetes, the posterior part of the radius is not widened [43]. The overall shape of the radius does not significantly differ from extant ziphiids ~~consulted~~ (e.g. *Berardius arnuxii* NMNZ 415, MNHN A3244; *Mesoplodon layardii* NMNZ 2917; *Tasmacetus shepherdii* NMNZ MM 2908; *Ziphius cavirostris* SNM CN1). However, its radius is wider than in *Messapicetus gregarius* where the ratio between the length and the width of radius equals 0.25 (0.34 ~~in genus and sp. indet.~~ NHMD 189993).

Ribs—two partial ribs are preserved (Fig. 2C). Their body is heavily fractured and fragmented. Judging from the similar outline of each rib, they were likely from the same pair. They are tentatively inferred to be the pair 2, because of their thick, yet flattened body. Both are double-headed with a marked neck separating the capitulum from the tuberculum.

4.2.1 Size estimates of the specimen

Condylobasal length and bitygomatic width were strongly correlated across the dataset ($R^2=0.78$; Fig. 7). The combination of the two linear measurements was sufficient to separate the four size categories (p -value < 0.0001), and each size category were well-distinguished.

The four size categories were better separated using the bitygomatic width, particularly in the case of the medium-sized and large-sized ziphiids. Indeed, species from these two categories displayed similar range of variation in condylobasal length, because of the strong variability of the rostrum anteroposterior length in those species. The large-sized ziphiid *Ziphius cavirostris* displayed a condylobasal length similar to other medium-sized ziphiids, such as *Mesoplodon mirus* and *M. europaeus*. This species was easily distinguished from the three other large-sized ziphiids *Mesoplodon layardii*, *Tasmacetus shepherdii* and *Hyperoodon planifrons* (Fig. 7).

On the opposite, the long-snouted medium-sized ziphiids *Messapicetus gregarius*, *Ninoziphius platyrostris*, *Dagonodum mojunum*, and *Mesoplodon grayi* displayed condylobasal length matching some large-sized ziphiids (*Mesoplodon layardii*). One specimen of *M. gregarius* almost reached the condylobasal length of some of the smallest specimens of very large-sized ziphiids. The long-snouted stem ziphiids were well separated from other ziphiids from their size category, including *Mesoplodon grayi*, which also display a strong elongation of the rostrum, but is also characterized by a smaller bitygomatic width. Small-sized ziphiids were easily distinguished from other size categories. They consist in the species *Mesoplodon peruvianus* and the fossil species *Nazcacetus urbinai* that possess a slightly longer condylobasal length.

Medium-sized ziphiids were better differentiated from large-sized ziphiids based on bitygomatic width. Based on the estimate of the bitygomatic width and condylobasal length, genus and sp. indet. NHMD 189993 would be a large ziphiid, between 5.5 and 7.5 m. Its condylobasal length and bitygomatic width is within the range of the extant Ziphiidae *Mesoplodon layardii* whose size generally ranged between 5.5 and 6 m [44]. Based on the similar condylobasal length, the degree of elongation of the rostrum in genus and sp. indet. NHMD 189993 is likely more similar to *M. layardii* than to the shorter rostrum of *Z. cavirostris* or the extremely elongated rostrum of long snouted stem ziphiids. The size of genus and sp. indet. NHMD 189993 clearly differs from the other fossil ziphiid found in the Gram Formation, *Dagonodum mojunum*, a medium-sized ziphiid whose size likely ranged between 4 and 4.5 m.

5. Discussion

5.1 Foraging Ecology

5.1.1 Suction Feeding

All extant beaked whales are specialized to use suction: they generate powerful suction pressures with their tongue acting like a piston, to capture and engulf their prey [4]. Many odontocetes can use suction feeding for capturing and/or transporting prey [45], but extant Ziphiidae are obligate suction feeders, due to the absence of functional teeth to capture their prey [4]. Furthermore, they exhibit a lateral closure of the intraoral cavity combined with a wider and thicker hyoid apparatus compared to odontocetes relying on a more raptorial feeding strategy [4]. The only extant ziphiid species that is perhaps not an obligate suction feeder is *Tasmacetus shepherdi*. Unlike other ziphiids, this species retains a set of erupted teeth likely functional [46]. Based on one stomach content containing mostly the fish species *Merluccius hubbsi*, MacLeod et al. [13] speculated that this species may specialize in feeding on deep-water fish rather, than cephalopods, thus limiting the competition with other species of beaked whales in the southern oceans where it occurs.

Several lines of evidence suggest that ~~genus and sp. indet.~~ NHMD 189993 was capable of using suction feeding to a larger extent than other toothed ziphiids, perhaps already an obligate suction feeding species. First, the stylohyal is strongly thickened and elongated. Its anteroposterior length is similar to the large ziphiids *Hyperoodon planifrons* and *Ziphius cavirostris*, whereas it is almost twice longer than in *Mesoplodon layardii*, a species close in size to NHMD 189993. A thickened stylohyal would be necessary to support strong tongue muscles. The styloglossus and the hyoglossus are the two main muscles responsible for the retraction of the tongue in a piston-like manner during suction [27]. The styloglossus originates on the lateral surface of the stylohyal, which is particularly thickened in the ziphiids *Mesoplodon mirus* and *M. europaeus*. Both species also possess a strong styloglossus: Reidenberg and Laitman [27] noticed that they developed the largest styloglossus relative to total body length from their sample. The thickening of the stylohyal is accompanied by a pronounced ridge along the lateral surface of the bone giving the stylohyal a triangular shape in transverse view [27]. This ridge was observed in several other ziphiid species (e.g. *Hyperoodon* spp., *Berardius* spp., *Mesoplodon bidens*, *M. layardii*, *Ziphius cavirostris*) and seems characteristic of the group. The same ridge is present in the ~~genus and sp. indet.~~ NHMD 189993.

The thickening of the stylohyal is accompanied by a reduction of the teeth in ~~genus and sp. indet.~~ NHMD 189993. For each dentary, the specimen possessed at least 17 alveoli, a number similar to *T. shepherdi* (18–28) [46], but largely inferior to long snouted stem ziphiids (*Dagonodum mojnium*, 29; *Messapicetus gregarius*, 25–26; *Ninoziphius platyrostris*, 40–42) [16–18]. Additionally, ~~several features of the teeth and the alveoli differ between the aforementioned species and~~ ~~genus and sp. indet.~~ NHMD 189993. The alveoli of the specimen, although individualized, are particularly shallow, greatly differing from the condition observed in other toothed beaked whales where the alveolar groove is deep, but the septa not necessarily well differentiated (e.g. *M. gregarius*; Bianucci et al., 2010). The teeth themselves are also well reduced compared to other toothed ziphiids including *T. shepherdi*. The longest tooth of ~~genus and sp. indet.~~ NHMD 189993 (excepting the tusk) measures 20 mm with a maximum diameter of 11 mm. The tooth measurements of the specimen are even smaller than in the medium-sized ziphiids *D. mojnium*, *M. gregarius*, and *N. platyrostris*. Furthermore, in ziphiids with functional teeth, the robust crown shows apical wear or interlocking facets suggesting that their dentition was functional. This is not the case in ~~genus and sp. indet.~~ NHMD 189993 where the small crown does not show sign of interlocking. In many teeth, the apex is broken off and does not allow an estimation of the degree of apical wear, but the few teeth with a preserved apex do not show signs of wear. Therefore, we hypothesize that the small teeth of NHMD 189993 were either embedded from the gum or too small to be used as a regular method of capture. We do not discard the possibility that NHMD 189993 could occasionally use its reduced teeth (if erupted) to manipulate or capture some of its prey.

5.1.2 A potential case of niche separation

Several morphological evidences and the different size estimates of *Dagonodum mojnium* and ~~genus and sp. indet.~~ NHMD 189993 suggest that these two species occupied two different ecological niches. It is possible that the two species were not contemporary: despite the similar age estimation for the two species, the error margin of 2.7 Ma could mean that they did not live at the same period of time. Assuming that the two species were contemporary, the co-occurrence of two different sized species of Ziphiidae at the same location suggest a case of niche separation.

Cases of niche separation are known in extant ziphiids: *Mesoplodon* species consistently feed on smaller prey type (generally, cephalopods under 500 g) compared to *Hyperoodon* and *Ziphius* species (cephalopods over 1 kg) [13]. The difference of prey size targeted may explain why species of *Mesoplodon* are often sympatric with the latter [47–50]. Size is not the only component, even though an important one [12], allowing niche separation between ziphiid species. In the case of *D. mojnium* and ~~genus and sp. indet.~~ NHMD 189993, the difference in specialization to suction feeding reinforces this hypothesis. The species *D. mojnium* possesses some adaptations to suction feeding (transverse thickening of the basyhyal and thyrohyal; presence of a precoronoid crest) [18], but not to the extent of ~~genus and sp. indet.~~ NHMD 189993, that probably relied more prominently on this feeding strategy. Obligate suction feeders with a reduced tooth count are often more teuthophagous [4,51], even though some ziphiids can still feed on fish [13]. Perhaps, the more specialized oral apparatus of ~~genus and sp. indet.~~ NHMD 189993 is more indicative of a more predominant teuthophagous diet than *D.*

mojnum. Interestingly, the fossil of a cuttlefish (Sepiida) was found in the Gram Formation (MSM DK718; unpublished data), an ideal prey type for ~~genus and sp. indet.~~ NHMD 189993.

Alternatively, the two species may have segregated geographically: whereas *D. mojnum* is assumed to be a local [18], the other specimen ~~genus and sp. indet.~~ NHMD 189993 may have been more adapted to deep diving like its modern representatives. No morphological features preserved allow the evaluation of the diving abilities of ~~genus and sp. indet.~~ NHMD 189993. Therefore, this hypothesis cannot be fully ruled out.

Other cases of niche separation between fossil ziphiids probably occurred at other locations where they show a diversity of sizes or feeding strategies. Bianucci et al. [52] already proposed this interpretation to explain the high diversity of fossil ziphiids trawled from the seafloor off South Africa. Fossil ziphiids from the Neogene of Antwerp and fished from the Atlantic Ocean floor off the Iberian Peninsula also show a great diversity of skull sizes, that could be indicative of ecological niche segregation [31,33]. In absence of precise datation for these three localities, it is unclear whether the different species were living during the same time period.

Conclusion

Despite the rich fossil record of beaked whales, the discovery of postcranial material remains a rare finding [18,37,53]. A new fossil of Ziphiidae, genus and sp. indet. NHMD 189993, consisting in the mandible, earbones, the stylohyal, isolated teeth including the tusk, the humerus and associated radius is described here. The fossil is dated to the mid- to late Tortonian (ca. 9.9-7.2 Ma). Despite the lack of cranial material for comparing with other similarly sized fossil ziphiids, ~~genus and sp. indet.~~ NHMD 189993 clearly differs from the other species found in the Gram Formation during the same time period, *Dagonodum mojnum*.

Unlike *D. mojnum* and other long snouted stem ziphiids, the morphology of the oral apparatus of ~~genus and sp. indet.~~ NHMD 189993 suggests that it was mostly relying on suction feeding. The reduced teeth perhaps still embedded in the gum, and morphological features of the thickened stylohyal supports this interpretation.

The two fossil species *D. mojnum* and ~~genus and sp. indet.~~ NHMD 189993 likely occupied different ecological niches with ~~genus and sp. indet.~~ NHMD 189993 likely feeding more predominantly on evasive prey such as cephalopods. Assuming the two species were chronologically concomitant, the spatial co-occurrence of the species *D. mojnum* and ~~genus and sp. indet.~~ NHMD 189993 can be illustrative of a case of niche separation. Together with sexual dimorphism [54], the specialization toward specific ecological niches in Ziphiidae may partly explain the rich specific diversity of this family.

Acknowledgments

We thank Mette Steeman for her suggestions and support for the completion of the project. We are also indebted to Frank Osbæk and Trine Sørensen that were in charge of the preparation of the specimen.

We thank the following colleagues for kindly allowing us to access some of the comparative material we used in this study: Morten Tange Olsen and Daniel Klingberg Johansson to access the SNM collection, Anne-Lise Folie for the IRSNB collection, Christian de Muizon and Christine Lefèvre for the MNHN collection, Chiara Sorbini for the MSNUP collection, Thomas Schultz for the NMNZ, and Charles Potter for the USNM collection.

Funding Statement

The project was funded by the Dansk Slots- og Kulturstyrelsen (FORM.2016-0021).

Data Accessibility

The datasets supporting this article have been uploaded as part of the Supplementary Material at the following address:

<https://figshare.com/s/c75c55bdca6ef3ae3a6f>

Competing Interests

'We have no competing interests.'

Authors' Contributions

B.R. identified and interpreted the ziphiid remains. H.L. created the digital reconstruction and the associated figures. B.R. made calculations for the size estimates and collected the measurements. Both authors participated in the preparation of the illustrations and the writing of the manuscript. Both discussed the results and commented on the manuscript at all stages.

References

- Dalebout ML et al. 2014 Resurrection of *Mesoplodon hotaula* Deraniyagala 1963: A new species of beaked whale in the tropical Indo-Pacific. *Marine Mammal Science* **30**, 1081–1108. (doi:10.1111/mms.12113)
- Morin PA et al. 2017 Genetic structure of the beaked whale genus *Berardius* in the North Pacific, with genetic evidence for a new species. *Marine Mammal Science* **33**, 96–111. (doi:10.1111/mms.12345)
- Clarke MR. 1996 Cephalopods as prey. III. Cetaceans. *Philosophical Transactions of the Royal Society of London. Series B: Biological Sciences* **351**, 1053–1065. (doi:10.1098/rstb.1996.0093)
- Heyning JE, Mead JG. 1996 Suction feeding in beaked whales: morphological and observational evidence. *Natural History Museum of Los Angeles County Contributions in Science* **464**, 1–12.
- Hooker Sascha K., Baird Robin W. 1999 Deep-diving behaviour of the northern bottlenose whale, *Hyperoodon ampullatus* (Cetacea: Ziphiidae). *Proceedings of the*

- Royal Society of London. Series B: *Biological Sciences* **266**, 671–676. (doi:10.1098/rspb.1999.0688)
6. MacLeod CD *et al.* 2006 Known and inferred distributions of beaked whales species (Cetacea: Ziphiidae). *J. Cetacean Res. Manage.* **7**, 271–286.
7. Johnson Mark, Madsen Peter T., Zimmer Walter M. X., Aguilar de Soto Natacha, Tyack Peter L. 2004 Beaked whales echolocate on prey. *Proceedings of the Royal Society of London. Series B: Biological Sciences* **271**, S383–S386. (doi:10.1098/rsbl.2004.0208)
8. Tyack PL, Johnson M, Soto NA, Sturlese A, Madsen PT. 2006 Extreme diving of beaked whales. *J. Exp. Biol.* **209**, 4238. (doi:10.1242/jeb.02505)
9. Minamikawa S, Iwasaki T, Kishiro T. 2007 Diving behaviour of a Baird's beaked whale, *Berardius bairdii*, in the slope water region of the western North Pacific: first dive records using a data logger. *Fisheries Oceanography* **16**, 573–577. (doi:10.1111/j.1365-2419.2007.00456.x)
10. Schorr GS, Falcone EA, Moretti DJ, Andrews RD. 2014 First Long-Term Behavioral Records from Cuvier's Beaked Whales (*Ziphius cavirostris*) Reveal Record-Breaking Dives. *PLoS ONE* **9**, e92633. (doi:10.1371/journal.pone.0092633)
11. Moore JC. 1968 Relationships among the living genera of beaked whales. *Fieldiana: Zoology* **53**, 209–298.
12. MacLeod CD, Santos MB, López A, Pierce GJ. 2006 Relative prey size consumption in toothed whales: implications for prey selection and level of specialisation. *Mar Ecol Prog Ser* **326**, 295–307. (doi:10.3354/meps326295)
13. MacLeod CD, Santos MB, Pierce GJ. 2003 Review of Data on Diets of Beaked Whales: Evidence of Niche Separation and Geographic Segregation. *Journal of the Marine Biological Association of the United Kingdom* **83**, 651–665. (doi:10.1017/S0025315403007616h)
14. Hocking David P., Marx Felix G., Park Travis, Fitzgerald Erich M. G., Evans Alistair R. 2017 A behavioural framework for the evolution of feeding in predatory aquatic mammals. *Proceedings of the Royal Society B: Biological Sciences* **284**, 20162750. (doi:10.1098/rspb.2016.2750)
15. Bianucci G, Landini W, Varola A. 1994 Relationships of *Messapicetus longirostris* (Cetacea, Ziphiidae) from the Miocene of South Italy. *Bollettino della Società Paleontologica Italiana* **33**, 231–241.
16. Bianucci G, Lambert O, Post K. 2010 High concentration of long-snouted beaked whales (genus *Messapicetus*) from the Miocene of Peru. *Palaeontology*, *Wiley* **53**, 1077–1098.
17. Lambert O, de Muizon C, Bianucci G. 2013 The most basal beaked whale *Ninoziphius platyrostris* Muizon, 1983: clues on the evolutionary history of the family Ziphiidae (Cetacea: Odontoceti). *Zoological Journal of the Linnean Society* **167**, 569–598. (doi:10.1111/zoj.12018)
18. Ramassamy B. 2016 Description of a new long-snouted beaked whale from the Late Miocene of Denmark: evolution of suction feeding and sexual dimorphism in the Ziphiidae (Cetacea: Odontoceti). *Zoological Journal of the Linnean Society* **178**, 381–409. (doi:10.1111/zoj.12418)
19. Lambert O, Bianucci G, Post K. 2009 A new beaked whale (Odontoceti, Ziphiidae) from the middle Miocene of Peru. *Journal of Vertebrate Paleontology* **29**, 910–922. (doi:10.1671/039.029.0304)
20. Bianucci G, Lambert O, Post K. 2007 A high diversity in fossil beaked whales (Mammalia, Odontoceti, Ziphiidae) recovered by trawling from the sea floor off South Africa. *Geodiversitas* **29**, 561–618.
21. Jolicoeur P. 1963 193. Note: the multivariate generalization of the allometry equation. *Biometrics* **19**, 497–499.
22. Marcus LF. 1990 Chapter 4: Traditional Morphometrics. In *Proceedings of the Michigan morphometrics workshop* (eds FJ Rohlf, FL Bookstein), pp. 77–122. Michigan: University of Michigan Museum of Zoology.
23. R Core Team. 2019 *R: A language and environment for statistical computing*. Vienna, Austria: R Foundation for Statistical Computing. See <https://www.R-project.org/>.
24. Mead JG, Fordyce RE. 2009 *The Therian Skull: a Lexicon with Emphasis on the Odontocetes*. Washington D.C.: Smithsonian Institution Scholarly Press.
25. Fitzgerald EM. 2016 A late Oligocene waipatiid dolphin (Odontoceti: Waipatiidae) from Victoria, Australia. *Memoirs of Museum Victoria* **74**, 117–136.
26. Marx F, Lambert O, Uhen MD. 2016 *Cetacean Paleobiology*. Chichester: John Wiley & Sons.
27. Reidenberg JS, Laitman JT. 1994 Anatomy of the hyoid apparatus in odontoceli (toothed whales): Specializations of their skeleton and musculature compared with those of terrestrial mammals. *The Anatomical Record* **240**, 598–624. (doi:10.1002/ar.1092400417)
28. Rasmussen LB. 1966 Molluscan faunas and biostratigraphy of the marine younger Miocene formations in Denmark. Part I: Geology and biostratigraphy. *Geological survey of Denmark* **88**, 1–358.
29. Piasecki S. 2005 Dinoflagellate cysts of the Middle-Upper Miocene Gram Formation, Denmark. *Palaeontos* **7**, 29–45.
30. Beyer C. 2005 A magnetic analysis of the Late Miocene Gram Formation, Denmark. *Palaeontos* **7**, 19–28.
31. Lambert O. 2005 Systematics and phylogeny of the fossil beaked whales *Ziphirostrum du Bus*, 1868 and *Choneziphius Duvernoy*, 1851 (Mammalia, Cetacea, Odontoceti) from the Neogene of Antwerp (North of Belgium). *Geodiversitas* **27**, 443–497.
32. Lambert O, Louwye S. 2006 Archaeoziphius microglenoideus, a new primitive beaked whale (Mammalia, Cetacea, odontoceti) from the Middle Miocene of Belgium. *Journal of Vertebrate Paleontology* **26**, 182–191. (doi:10.1671/0272-4634(2006)26[182:AMANPB]2.0.CO;2)
33. Bianucci G, Miján I, Lambert O, Post K, Mateus O. 2013 Bizarre fossil beaked whales (Odontoceti, Ziphiidae) fished from the Atlantic Ocean floor off the Iberian Peninsula. *Geodiversitas* **35**, 105–153.
34. Barnes LG. 1976 Outline of eastern North Pacific fossil cetacean assemblages. *Systematic Zoology* **25**, 321–343.
35. Fordyce RE, de Muizon C. 2001 Evolutionary history of cetaceans: a review. Secondary Adaptation of Tetrapods to Life in Water. In *Secondary Adaptation of Tetrapods to Life in Water* (eds J-M Mazin, V de Buffrénil), pp. 169–212. München.
36. Galatius A, Kinze CC. 2003 Ankylosis patterns in the postcranial skeleton and hyoid bones of the harbour porpoise (*Phocoena phocoena*) in the Baltic and North Sea. *Canadian Journal of Zoology* **81**, 1851–1861.
37. Bianucci G, Collareta A, Post K, Varola A, Lambert O. 2016 A New Record of *Messapicetus* from the Pietra Leccese (Late Miocene, Southern Italy): Antitropical Distribution in a Fossil Beaked Whale (Cetacea, Ziphiidae). *Rivista Italiana di Paleontologia Stratigrafia* **122**, 63–73.
38. Dalebout ML, Mead JG, Baker CS, Baker AN, van Helden AL. 2002 A New Species of beaked whale *Mesoplodon perrini* sp. n. (Cetacea: Ziphiidae) discovered through Phylogenetic Analyses of Mitochondrial DNA sequences. *Marine Mammal Science* **18**, 577–608.
39. Cope ED. 1869 Two extinct Mammalia from the United States. *Proceedings of the American Philosophical Society* **11**, 188–190.
40. Flower WH. 1882 On the Whales of the Genus *Hyperoodon*. In *Proceedings of the Zoological Society of London*, pp. 722–726. Oxford, UK: Blackwell Publishing Ltd.
41. Boschma H. 1951 Rows of small teeth in ziphioid whales. *Zoologische Mededelingen* **31**, 130–148.
42. Lambert O, Collareta A, Landini W, Post K, Ramassamy B, Di Celma C, Urbina M, Bianucci G. 2015 No deep diving: evidence of predation on epipelagic fish for a stem beaked whale from the Late Miocene of Peru. *Proceedings of the Royal Society B: Biological Sciences* **282**, 20151530. (doi:10.1098/rspb.2015.1530)

43. Benke H. 1993 Investigations on the osteology and the functional morphology of the flipper of whales and dolphins (Cetacea). *Investigations on Cetacea* **24**, 9–252.
44. Ross JGB. 1984 The smaller cetaceans of the south east coast of Southern Africa. *Annals of the Cape Provincial Museums* **15**, 173–410.
45. Werth AJ. 2006 Mandibular and Dental Variation and the Evolution of Suction Feeding in Odontoceti. *Journal of Mammalogy* **87**, 579–588. (doi:10.1644/05-MAMM-A-279R1.1)
46. Mead JG, Payne RS. 1975 A specimen of the Tasman beaked whale, *Tasmacetus shepherdi*, from Argentina. *Journal of Mammalogy*, 213–218.
47. Heyning JE, Ridgway SH, Harrison R. 1989 Cuvier's beaked whale *Ziphius cavirostris* G. Cuvier, 1823. In *Handbook of marine mammals*, pp. 289–320. London: Academic Press.
48. Mead JG. 1989 Bottlenose whales *Hyperoodon ampullatus* (Forster, 1770) and *Hyperoodon planifrons* Flower 1882. In *Handbook of marine mammals* (eds SH Ridgway, R Harrison), pp. 321–348. London: Academic Press.
49. Mead JG. 1989 Beaked whales of the genus *Mesoplodon*. In *Handbook of marine mammals* (eds SH Ridgway, R Harrison), pp. 349–430. London: Academic Press.
50. MacLeod CD. 2000 Distribution of beaked whales of the genus *Mesoplodon* in the North Atlantic. *Mammal Review* **30**, 1–8.
51. Clarke MR. 1986 Cephalopods in the diets of odontocetes. In *Research on Dolphins* (eds MM Bryden, R Harrison), pp. 281–321. Oxford: Clarendon Press.
52. Bianucci G, Lambert O, Post K. 2008 Beaked whale mysteries revealed by seafloor fossils trawled off South Africa. *South African Journal of Science* **104**.
53. de Muizon C. 1984 *Les vertébrés fossiles de la Formation Pisco (Pérou). deuxième partie: les Odontocètes (Cetacea, Mammalia) du Pliocène inférieur de Sud-Sacaco*. Travaux de l'Institut Français d'Etudes Andines.
54. Dalebout ML, Steel D, Baker CS. 2008 Phylogeny of the Beaked Whale Genus *Mesoplodon* (Ziphiidae: Cetacea) Revealed by Nuclear Introns: Implications for the Evolution of Male Tusks. *Systematic Biology* **57**, 857–875. (doi:10.1080/10635150802559257)

Tables

TABLE 1. Measurements of the mandible, cranial remains, and forelimb bones of the specimen genus and sp. indet. NHMD 189993.

Feature	NHMD 189993
Mandibles	
Anteroposterior length as preserved	1032
Maximum posterior width as preserved	412
Symphyseal portion anteroposterior length	289
Symphyseal portion maximal transverse width	61
Humerus	
Anteroposterior humeral head diameter	73
Maximal humeral length	204
Maximal distal width	80
Width at the level of the deltoid ridge	92
Radius	
Maximal radius length	186
Width at mid-length	63
Proximal width	67
Distal width	73
Measurements in mm	

TABLE 2. Measurements of the periotic and the tympanic bone of genus and sp. indet. NHMD 189993.

Feature	NHMD 189993
Right Periotic	
Maximal anteroposterior length	47
Maximal transverse width	33

Pars cochlearis maximum anteroposterior length	25
Pars cochlearis maximum transverse width	27
Anterior process maximum anteroposterior length	21
Anterior process maximum transverse width	20
Posterior process maximum anteroposterior length	18
Posterior process maximum transverse width	21
Lateral tuberosity in lateral view transverse width	10
Right Tympanic bulla	
Tympanic anteroposterior length	43
Tympanic maximum transverse width	26
Inner posterior prominence maximum transverse width	11
Outer posterior prominence maximum width	15
Dorsoventral height in lateral view as preserved	21
Involucrum indentation on the tympanic	
Dorsoventral height in medial view	12

All measurements are in mm and taken in ventral view unless noted otherwise.

Figures

Figure 1. Current extension of the Gram Formation in Denmark (shaded area). The type locality is situated in the Gram claypit, 1.5 km north of Gram. Modified from Rasmussen [28].

Figure 2. Mandibles, cranial, and postcranial remains of genus and sp. indet. NHMD 189993. **A**, ventral view; **B**, corresponding drawing; **C**, detail of the preserved nasal and other postcranial remains.

Figure 3. Digital reconstruction of the mandible and postcranial remains of genus and sp. indet. NHMD 189993. **A**, dorsal view; **B**, lateral view; **C**, detail of the anterodorsal part of the mandible.

Figure 4. Right periotic of genus and sp. indet. NHMD 189993. **A**, dorsal view; **B**, lateral view; **C**, medial view; **D**, ventral view; **E-H**, corresponding drawings.

Figure 5. Right tympanic of genus and sp. indet. NHMD 189993. **A**, dorsal view; **B**, medial view; **C**, ventral view; **D**, lateral view.

Figure 6. Postcranial elements of genus and sp. indet. NHMD 189993. Right humerus in **A**, lateral view; **B**, dorsal view; **C**, posterior view. Associated right radius **D**, in lateral view; **E**, medial view; **F**, dorsal view; isolated tusk in **G**, lateral view; **H**, ventral view; three isolated teeth in **I**, **K**, **M**, medial view; **J**, **L**, **N**, lateral view; right stylohyal in **A**, lateral view; **B**, medial view; **C**, dorsal view.

Figure 7. Log-transformed bizygomatic width plotted against condylobasal length in Ziphiidae. **Abbreviations:** **Bear:** Berardius arnuxii; **Beba:** Berardius bairdii; **Hyam:** Hyperoodon ampullatus; **HypI:** Hyperoodon planifrons; **Megr:** Mesoplodon grayi; **Mela:** Mes. layardii; **Mepe:** M. peruvianus; **Messgr:** Messapicetus gregarius; **Naur:** Nazzacetus urbanai; **Nipl:** Ninoziphius platyrostris; **Tash:** Tasmacetus shepherdii; **Zica:** Ziphius cavirostris. The cluster of medium ziphiids corresponds to the species *M. bidens*, *M. europaeus*, *M. ginkgodens*, and *M. mirus*.

Supplementary Data

Supplementary data S1. Condylobasal lengths and postorbital widths from specimens of Ziphiidae used in the analysis.

Supplementary data S2. Pdf file containing a 3D reconstruction of the genus and sp. indet. NHMD 189993.

Figure 1. Current extension of the Gram Formation in Denmark (shaded area). The type locality is situated in the Gram claypit, 1.5 km north of Gram. Modified from Rasmussen [28].

Figure 2. Mandibles, cranial, and postcranial remains of genus and sp. indet. NHMD 189993. A, ventral view; B, corresponding drawing; C. detail of the preserved nasal and other postcranial remains.

Figure 3. Digital reconstruction of the mandible and postcranial remains of genus and sp. Indet. NHMD 189993. A, dorsal view; B, lateral view; C, detail of the anterodorsal part of the mandible.

Figure 4. Right periotic of genus and sp. indet. NHMD 189993. A, dorsal view; B, lateral view; C, medial view; D, ventral view; E-H, corresponding drawings.

Figure 5. Right tympanic of genus and sp. indet. NHMD 189993. A, dorsal view; B, medial view; C, ventral view; D, lateral view.

Figure 6. Postcranial elements of genus and sp. indet. NHMD 189993. Right humerus in A, lateral view; B, dorsal view; C, posterior view. Associated right radius D, in lateral view; E, medial view; F, dorsal view; isolated tusk in G, lateral view; H, ventral view; three isolated teeth in I, K, M, medial view; J, L, N, lateral view; right stylohyal in A, lateral view; B, medial view; C, dorsal view.

34 Figure 7. Log-transformed bizygomatic width plotted against condyl basal length in Ziphiidae.
 35 Abbreviations: Bear: *Berardius arnuxii*; Beba: *Berardius bairdii*; Hyam: *Hyperoodon ampullatus*; Hypl:
 36 *Hyperoodon planifrons*; Megr: *Mesoplodon grayi*; Mela: *Mes. layardii*; Mepe: *M. peruvianus*; Messgr:
 37 *Messapicetus gregarius*; Naur: *Nazcacetus urbinai*; Nipl: *Ninoziphius platyrostris*; Tash: *Tasmacetus*
 38 *shepherdii*; Zica: *Ziphius cavirostris*. The cluster of medium ziphiids corresponds to the species *M. bidens*, *M.*
 39 *europaeus*, *M. ginkgodens*, and *M. mirus*.

A new specimen of Ziphiidae (Cetacea, Odontoceti) from the late Miocene of Denmark with morphological evidences for suction feeding behaviour

BENJAMIN RAMASSAMY*,¹ and HENRIK LAURIDSEN²

¹Department of Natural History and Palaeontology, the Museum of Southern Jutland, Lergravsvej 2, Gram, 6510, Denmark;

²Comparative Medicine Lab, Department of Clinical Medicine, Aarhus University, Palle Juul-Jensens Boulevard 99, Aarhus, 8200, Denmark;

Keywords: Feeding Strategy; Ecological niche; Systematics; Ziphiidae

1. Summary

A new fossil of Ziphiidae from the upper Miocene Gram Formation (ca. 9.9-7.2 Ma) is described herein. Computed Tomographic scanning of the specimen was performed to visualize the mandibles and to obtain a 3D digital reconstruction. It possesses several ~~derived~~ characters ~~ofn~~ the ~~derived~~ Ziphiidae, such as the dorsoventral thickening of the anterior process of the petiotic, the dorsoventral compression of the pars cochlearis, and the short unfused symphysis. The specimen cannot be identified beyond the family level, because of the unusual nature of the preserved parts consisting of the mandibles, earbones and postcranial remains. It differs from the other Ziphiidae species from the Gram Formation, *Dagonodum mojunum*, ~~inbased on~~ its larger size and the more derived morphology of its mandibles and earbones. Its long and thickened stylohyal, combined with its ~~reduced~~ teeth ~~reduction~~ suggests that this new specimen relied primarily on suction feeding. By contrast, the other ziphiid species from the Gram Formation, *D. mojunum*, ~~was shows~~ ~~adaptionsed~~ ~~forte~~ a more raptorial feeding strategy. Assuming the two species were ~~chronologically concomitant~~ ~~coeval~~, their co-occurrence at the same locality with two different feeding strategies, may represent a case of niche separation. They may have hunted different types of prey, thus avoiding ~~direct~~ competition for the same food resource.

2. Introduction

Beaked whales (Ziphiidae) represent a diversified family of echolocating toothed whales (Odontoceti), currently represented by at least 22 species in 6 genera [1] with a potential new species of *Berardius* suspected in the North Pacific [2]. Their best-known modern representatives are capable of regular deep dives beyond 1000 meters to reach their foraging grounds, where they prey mostly on cephalopods and more occasionally on bathypelagic fish and crustaceans [3–10]. Most extant ziphiids are typified by a strong reduction of their tooth count ~~reduced~~ to one or two ~~mandibular~~ pairs, often only erupted in adult males [11]. Beaked whales do not use them to capture or manipulate their prey; instead, they use suction feeding as their main feeding strategy, except perhaps for the toothed ziphiid *Tasmacetus shepherdi* which retains a set of functional teeth ~~in both the upper and lower jaws~~ [4]. Suction feeding forces ~~them-ziphiids~~ to be more selective ~~relative with respect~~ to the size of their prey, thus allowing different species of beaked whales to be sympatric without competing for the same food resource [12,13].

Recently, Hocking et al. [14] proposed a new framework to understand the evolution of feeding in predatory aquatic mammals. Instead of thinking the different feeding styles as rigid categories, they argue that feeding strategies of aquatic mammals follow a particular evolutionary sequence that can be used to predict the origin of particular feeding styles. Under this framework, the specialization ~~forte~~ suction feeding of extant beaked whales should arise from ancestors ~~that using-used~~ a more raptorial feeding strategy. The fossil record of Ziphiidae confirm this prediction: some of the most basal beaked

*Author for correspondence (benjamin.ramassamy@laposte.net).

†Present address: Department of Natural History and Palaeontology, the Museum of Southern Jutland, Lergravsvej 2, Gram, 6510, Denmark

whales possessed elongated jaws and numerous functional interlocking teeth potentially used to capture their prey [15–18]. However, morphological evidences suggest that some of them were also capable of using suction feeding—at least in the most posterior part of the mandibles [17,18]. For example, *Dagonodum mojunum*, a new fossil genus and species late Miocene ziphiid from the Gram Formation of Denmark with close affinities to *M. gregarius*, was described from the Gram Formation in Denmark and was interpreted as a more raptorial feeder than extant beaked whales based on its numerous interlocking teeth and elongated jaws, despite moderate adaptations to suction feeding [18]. This species was interpreted as a more raptorial feeder than extant ziphiids based on its numerous interlocking teeth and elongated jaws, despite moderate adaptations to suction feeding.

A new fossil Ziphiidae from the same locality is described here. The preserved parts of the specimen consist of the lower jaw mandibles, earbones, part of the hyoid apparatus, and forelimb elements. This project paper aims at describing the specimen and at proposing a palaeoecological reconstruction of its autecology based on morphological features. Elements relative aspects of feeding strategies and the ecological niches occupied by the ziphiids from the Gram Formation are also discussed.

3. Materials and Methods

3.1 Specimen Preparation and Computed Tomography

The specimen was discovered in 2007 and prepared by means of mechanical tools at the curatorial department of the Museum of Southern Jutland (Denmark). A co-polymer of acrylates (MA/EMA Paraloid B72) was used as an adhesive to keep the fragments of the lower jaw together. Photos of the specimen were taken using a Fujifilm FinePix HS10 with a focal length of 4.2–126.0 mm.

Specimens coming from the Gram Formation are fragile, difficult to handle, and prepare. Furthermore, the preparation sometimes results in the loss of information about in relation with the original placement of the bone structures. Similar use of computed tomography (CT) analysis has already been applied to fossil Ziphiidae with great success [19].

To alleviate the preparation work and avoid extensive manipulation of the specimen, the lower jaws were scanned using a clinical computed tomography (CT) system (Siemens Somatom; Siemens Medical Solutions, Forchheim, Germany) using the following parameters: 0.98×0.98×0.60 mm³ voxel size; 140 kVp tube voltage; 185 μAs tube charge, resulting in an acquisition time of ~60 s. Data were reconstructed using a B45s convolution kernel. The CT scans 3D reconstruction unveiled the dorsal and lateral side of the lower jaws as preserved that otherwise, would have not been accessible without extensive preparation. Visualizations of the scanned fossil were made done using the DICOM-viewer OsiriX (Pixmeo SARL) and image segmentation and construction of an interactive model of the fossil was done made in Amira 5.6 (FEI, Visualization Sciences Group). The digital reconstruction is available in the Supplementary Data (Supplementary Figure S1).

3.2 Geological and palaeoenvironmental setting

Originally, three members were recognised in the Gram Formation: the lowermost glaucony-rich clay, the Gram clay and the Gram Sand member [20]. The Glauconite clay member is now recognized as part of the Ørnhøj Formation and the Gram Sand member as the Marbæk Formation [21]. The type section is found at the Gram Formation where a 13.1 m thick section of Gram clay is exposed [21]. Neither the base nor the top is visible, the reference section being 16 m thick [22]. The Gram Formation consists of dark brown clay with siderite concretions in the lower part and a few fine-grained wave rippled sand beds in the upper part [21]. The Gram Formation was deposited in a fully marine environment with water depth up to 100 m [23]. The occurrence of storm beds in the upper part suggests a progradation of the shoreline [21].

Estimation of the age of the formation is based on several lines of evidence. Rasmussen [20] identified five biozones based on the abundance of mollusc fauna found in the Gram Formation. A more recent biostratigraphy based on dinoflagellate cysts suggests that the Gram Formation was deposited between the late Serravalian and Tortonian age with the consistent occurrences of the dinoflagellate cysts *Hystrichosphaeris obscura*, *Spiniferites solidago* and *Labyrinthodinium truncatum* [24]. The most precise age estimation of the Gram Formation is indicated by a magnetic analysis of a 16 m vertical profile [25]. Beyer [25] identified a reverse polarity zone of less than 70 000 years at 14.8 m deep, approximately the basis of the mollusc biozones identified by Rasmussen [22]. This leaves three possible datations for the Tortonian stage: 7.1 Mya, 7.4 Mya, and 9.9 Mya [25]. Furthermore, the analysis of the accumulation rates indicates that the Gram Formation was deposited during 120 000 years with a much faster deposition rate in the uppermost 8 m of the formation (approximately 20 000 years) [25].

Based on these multiple lines of evidence, the Gram Formation can be dated from the Tortonian age with a maximum age of 9.9 Mya (based on the reverse polarity zone) and a minimum age of 7.2 Mya corresponding to the Tortonian-Messinian boundary (based on the dinoflagellate cysts biostratigraphy).

3.2.3 Size Estimation and Evaluation of Trophic level

Cetaceans, particularly obligate suction feeders, are known to select their prey relative to their own size [12]. Therefore, assessing the size of a ziphiid individual and comparing it with other species may help estimating the trophic level at which ~~the specimen~~ used to feed.

To do so, two cranial measurements were collected from different ziphiids specimens: the bizygomatic width and the condylobasal length (data available in Supplementary Dataset S2). Many fossil ~~specimen~~ forms had to be discarded, because their partial skull did not allow a good estimation of the condylobasal length and/or the bizygomatic width. The fossil species *Ninoziphius platyrostris*, *Nazcacetus urbinai*, and *Messapicetus gregarius* were included based on the measurements provided in their respective descriptions [16,17,2649]. In absence of a preserved skull for the specimen NHMD 189993 described herein, such measurements were not available. The anteroposterior length and posterior transverse width of the mandibles were used instead of, respectively, the bizygomatic width and the condylobasal length. The posterior transverse width of the ~~specimen~~ mandibles is a good estimator of the bizygomatic width but in this case, ~~is slightly results in a slight underestimation of the latter dimension~~ due to the lack of the most posterior parts of the mandibles. Anteroposterior length of the mandibles is significantly shorter than bizygomatic width in odontocetes; ~~as such and the latter dimension is therefore should only be taken as~~ an indicator of minimum size rather than a precise estimator/proxy. Cranial measurements were ~~nonetheless~~ selected for other ziphiids, because the mandibles of beaked whales are often disarticulated and the posterior width of the mandibles is therefore not always measurable.

Ziphiid species were ~~placed~~ regarded as representatives of ~~four~~ four size categories: very large-sized ziphiids (8-10 m), large-sized ziphiids (5.5-7.5 m), medium-sized ziphiids (4-4.5 m), and small-sized ziphiids (3-4 m). Those categories were defined in Bianucci et al. [270] based on a regression of the postorbital width relative to the body length of different ziphiid species.

A natural logarithmic transformation was applied to the cranial measurements to attenuate the effect of allometry and correct for heteroscedasticity [281,292]. A MANOVA (Multivariate analysis of variance) was performed to evaluate whether the cranial measurements were sufficient to assess each size category. It was followed by a Tukey's HSD (honest significant difference) test on each variable to compare differences between the size categories. Linear regression was also performed on the dataset to assess the relationship between the two cranial measurements. All ~~a~~Analyses were performed with the software R 3.6.0 [230].

3.33.4 Nomenclature

Institutional Abbreviations—**IRSNB**, Institut Royal des Sciences Naturelles de Belgique, Brussels, Belgium; **MNHN**, Muséum National d'Histoire Naturelle, Paris, France; **MSM**, Museum Sønderjylland Naturhistorie og Palæontologi, Gram Lergav, Gram, Denmark; **MSNUP**, Museo di Storia Naturale dell'Università di Pisa, Italy; **MUSM**, Museo de Historia Natural, Lima, Peru; **NMNZ**, National Museum of New Zealand Te Papa Tongarewa, Wellington, New Zealand; **NHMD**, Statens Naturhistoriske Museum, Copenhagen, Denmark; **USNM**, United States National Museum of Natural History, Smithsonian Institute, Washington D.C., U.S.A.

Anatomical Terminology—The anatomical terminology of the skull ~~and earbones~~ follows Mead and Fordyce [3124]. The terminology used by Fitzgerald [3225] and Marx et al. [3326] was followed for the postcranial remains. The nomenclature of Reidenberg and Laitman [3427] was used for describing elements of the hyoid apparatus.

4. Results

4.1 Systematic Palaeontology

Order CETACEA Brisson, 1762
Suborder ODONTOCETI Flower, 1867
Family ZIPHIIDAE Gray, 1850
Genus and species indet.

Referred Material—NHMD 189993, subcomplete mandibles, the right stylohyal, 14 isolated teeth including ~~one~~ tusk, two periotics and the right tympanic, the right humerus and associated radius, parts of the nasal (~~unambiguous identification of a side is impossible~~).

Horizon and Locality—The ~~finding~~ locality is situated 1.5 km north of the town of Gram, Southern Jutland, Denmark (55°18'02.67"N, 9°30'32.51"E; Fig. 1). The specimen ~~was dated based on the mollusc fauna identified in association with genus and sp. indet.~~ NHMD 189993 ~~was dated on the basis of the associated mollusc fauna assemblage~~. The high percentage of *Carinastarte vetula reimersi* (~~accounting for~~ 55% of the specimens identified) and the co-occurrence of the species *Gemmula badensis* and *Turritella tricarinata* suggest that the specimen was originally found in the assemblage Zone V [208].

The assemblage Zone V belongs to the upper part of the Gram Formation dated from the Tortonian age, based on the co-occurrence of the dinoflagellate cysts *Hystichosphaeris obscura*, *Labyrinthodinium truncatum* and *Spiniferis solidago*, ~~and~~

Labyrinthodinium truncatum [249]. The ~~minimum-maximum~~ age of the ~~a~~Assemblage ~~z~~Zones from the Gram Formation is estimated to 9.9 Ma based on the presence of a polarity zone ~~of less~~shorter than 70 000 years [2530]. ~~The specimen genus and sp. indet~~ NHMD 189993 can ~~thus~~ be dated from the mid- to late Tortonian, ca. 9.9-7.2 Ma.

Systematic Attribution of the Specimen—The specimen is ~~identified as a member of~~assigned to the family Ziphiidae based on the following combination of characters: the enlargement of the apical or subapical mandibular tooth; the reduction of the dorsal keel on the posterior process of the periotic; the mediolateral thickening of the anterior process of the periotic; in dorsal view, the anterior shift of the pars cochlearis of the periotic.

~~Genus and sp. indet~~ NHMD 189993 clearly differs from the other species found in the Gram Formation, *Dagonodum mojnum*, based on the following ~~combination of~~ characters: the reduction of the mandibular teeth; the shorter unfused symphysis; the dorsoventral thickening of the anterior process of the periotic; the dorsoventral compression of the pars cochlearis; the presence of a cochlear spine.

Identification beyond the genus was not possible, because of the unusual nature of the preserved material. Most fossil of ziphiids are represented by cranial remains, mostly the rostral, preauricular, and vertex region [15,250,351–373]. Mandibles, earbones, and postcranial remains are more rarely preserved. Despite the unusual features present on the periotics (presence of a cochlear spine, depression along the medial surface of the posterior process) and the ~~relatively peculiar-large~~ size of the specimen, the lack of cranial remains makes it nearly impossible to compare ~~it~~ with many similarly sized ziphiids ~~only known from the preauricular region whose mandibles are not preserved~~ (e.g. *Africanacetus*, *Globicetus*, *Tusciziphius*). Genus and sp. indet. NHMD 189993 possesses several derived ~~crown Ziphiidae~~ features ~~thought to be characteristic of a crown Ziphiidae~~ [e.g. 16,17,26,36]–16,33,33]—the dorsoventral thickening of the anterior process of the periotic bone, the dorsoventral compression of the pars cochlearis ~~of the periotic~~, the short and unfused ~~mandibular~~ symphysis. However, a recent phylogenetic analysis proposed that some members of the more basal *Messapicetus* clade displayed derived characters indicative of a convergent evolution between stem and crown Ziphiidae [38] (Bianucci et al., 2016a). Mandibles, earbones and postcranial material of the most derived members of this clade, *Globicetus*, *Tusciziphius*, and *Imocetus* are not known [363]. It is therefore impossible to assess whether NHMD 189993 was a crown ziphiid or a member of the *Messapicetus* clade.

By measure of caution, the advice of Barnes [394] and Fordyce and Muizon [4035], which suggest that the identification of a new cetacean species should at least include skull and rostrum, is followed until more cranial material is available.

4.2 Description and Comparisons

4.2.1 Cranium and Mandible

Overview and ontogeny—~~The specimen~~ NHMD 189993 is interpreted as an adult based on the complete fusion of the humeral head to the humeral shaft and the epiphyseal ankylosis of ~~each-both~~ epiphyses of the radius. In the porpoise *Phocoena phocoena*, extensive ankylosis of the postcranial skeleton characterizes adult specimens [4136].

The most robust parts of the mandibles and postcranial elements of ~~genus and sp. indet~~ NHMD 189993 are well preserved compared to the more fragmentary cranial remains and ribs (Fig. 2-3). The earbones were found ~~close to~~with the lower jaw, the right periotic still having the stapes firmly attached to it (Fig. 4-5; only right periotic illustrated). The humerus and radius were originally still articulated (Fig. 6A-F), whereas the stylohyal lied along the right lateral side of the symphysis (Fig. 6O-Q). ~~The teeth~~ were collected ~~around the specimen~~ out of their mandibular sockets ~~around the bones~~ (Fig. 6G-N). The preserved parts of the mandibles are 1032 mm long and 412 mm wide. More measurements of the specimen are available in Tables 1 and 2.

Nasal—Because the nasal is the only piece identifiable from the shattered cranium of the specimen, its orientation reveals difficult. ~~Unambiguous identification of a side is impossible~~. The dorsal exposure is flat and rectangular (Fig. 2C). The ratio between the width and length of the visible ~~dorsal~~ surface is 0.70 (60 mm long and 86 mm wide). No excavation is visible on the surface of the nasal bone.

Periotic—Measurements of the right periotic are available in Table 2. The anterior process of the periotic is transversely thickened, with a large rounded protuberance along the dorsomedial surface of the periotic (Fig. 4A-C). ~~This~~ ~~Such~~ strong ~~thickening both~~ lateromedial and dorsoventral ~~thickening~~, is observed in all crown ziphiids, but not in the stem-ziphiids *Dagonodum mojnum*, *Messapicetus gregarius*, and *Ninoziphius platyrostris* [16–18]. In the latter, the thickening occurs only lateromedially. The tip of the anterior process is pointed. In ventral view, the anterior bullar facet is anteroposteriorly elongated and elliptical. Posteromedially to this facet, the accessory ossicle is still articulated in the fovea epitubaria (Fig. 4D). It extends along the dorsomedial margin of the anterior process. It is less rounded and developed than in *Berardius*, *Hyperoodon*, some species of *Mesoplodon* (*M. carlhubbsi*, *M. europaeus*, *M. grayi*, *M. mirus*), *Nazcacetus*, and *Tasmacetus*. In ventral view, a sulcus extends anteroposteriorly along the accessory ossicle, and separates it in two ~~portions~~ (Figure 4D). A similar sulcus is also observed in ~~Hyperoodon- ampullatus~~, ~~Mesoplodon- densirostris~~, and ~~Tasmacetus- shepherdii~~, although less developed in ~~these~~ species. The sulcus observed in ~~genus and sp. indet~~ NHMD 189993 could ~~indicate refer to~~ the origin of the tendon of m. tensor tympani [3124]. Posteriorly to the fovea epitubaria and the accessory ossicle, the malleolar fossa develops ~~along~~ along the medial margin of the lateral tuberosity. In ventral view, the anterior process is separated from the lateral tuberosity by the anteroexternal sulcus, which can also be seen in lateral view. The anteroexternal sulcus is also present in the periotic of *D. mojnum*. In ventral view, the lateral tuberosity is lateromedially elongated, a character observed in all ziphiids, except *D. mojnum*, *M. gregarius*, and *N. platyrostris* [16–18]. The fenestra ovalis is rounded. Posteroventrally to the fenestra ovalis, a deep hiatus epitympanicus separates the posterior process from the lateral tuberosity. In ventral view, the posterior process of the periotic is fan-shaped: it is rounded and widens abruptly ~~from~~

anterior to posteriorly (Fig. 4D). A fan-shaped posterior bullar facet is characteristic of all ziphiids, except *D. mojnium*, *N. platyrostris*, and *M. gregarius* [16–18]. In medial view, the posterior process is oriented posteroventrally. ~~Genus and sp. indet.~~-NHMD 189993 lacks a distinct keel along the whole posterior process, a feature present in all ziphiids [17]. A deep depression excavates the anteromedial side of the posterior process, just posterior to the pars cochlearis (Fig. 4C). This depression seems unique to ~~genus and sp. indet.~~-NHMD 189993 and was not observed ~~in the periotic bone of among~~ ~~to~~ other ziphiids ~~for which the periotic portion is known~~.

In dorsolateral view, the pars cochlearis is anteriorly shifted, a feature that distinguishes ~~an eurhinodelphida ziphiid~~ from a ~~ziphiidn eurhinodelphid~~ periotic (Fig. 4A) [16]. In ventromedial view, the pars cochlearis is rectangular, because of its straight anteromedial corner. It is also dorsoventrally compressed, a feature observed in crown ziphiids, but absent in *N. platyrostris* and members of the *Messapicetus* clade [17]. In ventral view, the pars cochlearis bears a triangular depression similar in shape to *D. mojnium* [18]. This depression is also visible in the species *Mesoplodon mirus* (USNM 504612, USNM 550351, USNM 572961, USNM KLC112) and *M. bidens* (MNHN 1975.112, SNM CN5x), but is elliptical ~~and~~; more elongated anteroposteriorly ~~in both species than in NHMD 189993~~. The tear-shaped fenestra rotunda is oriented posteroventrally. Posterodorsally to the internal acoustic meatus, the periotic bears a large cochlear spine. This unusual feature in Ziphiidae is present in ~~the species~~-*N. platyrostris* and *Berardius arnuxii* [17]. Its presence was also observed in ~~the species~~-*Berardius bairdii* (USNM 571524). The cochlear spine in ~~genus and sp. indet.~~-NHMD 189993 is moderately developed dorsally, a condition similar to ~~that of~~ the genus *Berardius* and differing from the well-marked cochlear spine of *N. platyrostris*. In dorsomedial view, the internal acoustic meatus is elliptical; this feature is also observed in ~~the periotic of~~-*N. platyrostris* and is connected to the presence of the cochlear spine. A thick crest separates the internal acoustic meatus from the aperture for the vestibular aqueduct (Fig. 4C, G). Inside the internal acoustic meatus, the dorsal vestibular meatus is separated from its ventral counterpart by a transverse crest. The ventral vestibular area occupies almost two thirds of the surface of the internal acoustic meatus. Posteriorly to the vestibular area of the internal acoustic meatus, the aperture for the vestibular aqueduct is anteroposteriorly compressed. Ventrally to the vestibular aqueduct, the aperture for the cochlear aqueduct is reduced to a small opening (Fig. 4C).

Tympanic bulla—The right tympanic bulla is partially preserved (Fig. 5). ~~M~~Further measurements are available in Table 2. It lacks the base of the pedicle, the sigmoid process, and the dorsal part of the outer lip. In ventral view, the bulla is heart-shaped, because of the interprominential notch well marked posteriorly that separates the inner and outer posterior prominences. In ventral view, the inner posterior prominence is compressed transversely. ~~The outer posterior prominence is twice larger than the inner posterior prominence and is less developed posteriorly than the twice larger outer posterior prominence~~ (Fig. 5C). The ~~degree of~~ compression of the inner posterior prominence ~~is similar to recalls~~ some species of *Mesoplodon* (e.g. *M. bidens* SNM CN5x, *M. bowdoini* NMNZ MM2653, *M. europaeus* USNM 504349), *Messapicetus gregarius*, and *N. platyrostris*. In *Hyperoodon* spp. and *Z. cavirostris*, the inner posterior prominence is more reduced and is even shorter posteriorly. In ventral view, the interprominential notch connects to the deep median furrow. The median furrow extends roughly until the first third of the bulla (Fig. 5C). The median furrow is ~~moreless~~ developed ~~than~~ in *Hyperoodon* spp. and *Z. cavirostris*, but ~~lessmore~~ extended ~~than~~ in the stem ziphiids *D. mojnium*, *M. gregarius*, and *N. platyrostris*. In ventral view, a keel extends along the whole anteroposterior length of the bulla.

The involucrum is indented, a feature visible both in dorsal and medial view (Fig. 5A–B). In medial view, the ventral part of the bulla is incurved, but does not reach the dorsalmost margin of the posterior portion of the involucrum, as in *D. mojnium* (Fig. 5B). The anterior margin of the tympanic bulla is too damaged to assess the degree of development of the tympanic spine, if present. However, the broken anterolateral margin of the bulla develops anteriorly into a thin bone plate (Fig. 5A), a condition similar to *N. platyrostris*, where the tympanic spine is absent [17].

Stapes—The right stapes is still firmly attached to the periotic in the fenestra ovalis and could not be removed (Fig. 4C). The stapes is conical, widening at its oval base, as observed in several ziphiids species ~~[26]~~ ~~(Lambert et al., 2009)~~. The head of the stapes has a circular outline. The small and circular vestigial stapedia foramen opening is situated approximately at mid-length of the stapes. The muscular process is well developed and situated at the level of the head of the stapes.

Mandible—~~The Both~~ mandibles of ~~genus and sp. indet.~~-NHMD 189993 lacks the posterior part of the acoustic window and the mandibular condyle (Fig. 2-3). ~~Further measurements are available in Table 1~~. The symphyseal portion of the mandible is unfused (Fig. 3C). It is not ankylosed as in the long snouted stem ziphiids *Dagonodum mojnium*, *Ninoziphius platyrostris*, *Messapicetus* spp., *Ninoziphius platyrostris* and ~~genus and sp. indet.~~ MSUM 3237 *Dagonodum mojnium* [16–18, 38, 4237]. The symphysis is 289 mm long and represents at most 28 percent of the total length of the mandible (the total length of the preserved parts is 1032 mm). This value is much lower than in ~~the~~ long snouted ziphiids *Dagonodum mojnium*, *Messapicetus* spp., *Ninoziphius platyrostris*, and ~~in the extant species~~ *Tasmacetus shepherdii*, where the symphysis extends at least along 36 percent of the mandible total length [17]. The ~~transverse section of the~~ symphyseal portion of ~~the mandibles~~ ~~genus and sp. indet.~~-NHMD 189993 is triangular, differing from the half-circled section of *Berardius* spp., *D. mojnium*, *Messapicetus* spp., *N. platyrostris*, ~~and~~ *Tasmacetus shepherdii* ~~and MUSM 3237~~ [16–18, 42]. The symphyseal portion of the mandible is turned upwards. This feature is also present in ~~the species~~-*H. ampullatus*, *M. bidens*, *M. grayi*, *M. mirus*, *N. urbinai*, and *Z. cavirostris* [2619]. ~~The short unfused triangular symphysis of NHMD 189993 is close to~~ *Chavinziphius maxillocristatus* ~~whose mandible exhibits similar features~~ [38].

The apex of the mandibles is heavily fractured, but the fragments ~~conserved~~ ~~preserved~~ their original position, ~~thus~~ allowing an estimation of the original outline. In ventral view, the apex is rounded and likely possessed an enlarged alveolus for the tusk. This interpretation fits with the shape and size of the preserved tusk that is similar to those of several long snouted stem beaked whales (*D. mojnium*, *Mess. gregarius*) possessing a pair of tusks in apical position (Fig. 6G–H). Furthermore, no other alveolus along the alveolar groove is sufficiently developed to support the tusk. The apex of the mandible is too fractured to identify precisely whether the tusk was positioned apically or subapically, as observed by Dalebout et al. [4338]

in *Mesoplodon perrini*. It is also possible that ~~genus and sp. indet.~~ NHMD 189993 possessed two pairs of tusks, even though only one tusk is preserved with the specimen (Fig. 6G-H). This character is observed in *Berardius* spp., *Anoplonyssa forcipata*, and *D. mojnium* [18,4439]. One mental foramen is visible along the lateral side of the mandible. It is elongated, well-marked ~~individualised~~ and situated slightly posterior to the symphysis.

The outline of individualised alveoli can be distinguished in the alveolar border (Fig. 3C). The number of ~~counted-detected~~ alveoli along the left dentary is 17, which is probably a slight underestimation, because of the eroded and fractured surface of the alveolar border of the most apical parts of the symphysis. It is not possible to assess the presence of a diastema between the tusk and the rest of the alveolar groove. The alveoli are oval, transversely compressed like in the *Messapicetus* spp. [15,16]. However, they are much more reduced than in the latter and much shallower compared to long snouted stem ziphiids, *C. cristatus* and the species *Tasmacetus shepherdii* [17,18, 38].

In the 3D reconstruction of the lower jaw, in lateral view, the position of the bone fragments posterior to the alveolar groove suggests the presence of a precoronoid crest. However, the dorsal surface of the acoustic window is too fractured to ~~be certain of it~~ draw definitive conclusions of this issue. Further measurements are available in Table 1.

Teeth—~~Together Alongwith~~ the mandibles of ~~genus and sp. indet.~~ NHMD 189993, 14 isolated teeth were ~~retrieved recovered~~ (Fig. 6G-N). Their crown is approximately as developed as the root dorsoventrally and curves lingually. The crown progressively widens ventrally and projects posteroventrally. The section at the base of the crown is circular, whereas the transverse section of the root is more oval. An oval root is present in *Messapicetus* spp., unlike the circular section observed in *Tasmacetus shepherdii* and the squared root observed in the species *Dagonodum mojnium* and *Ninziphius platyrostris*. A ~~discrete-faint~~ mesial keel is present in some of the smallest teeth of ~~genus and sp. indet.~~ NHMD 189993. Despite the presence of individualized alveoli, the reduced size of the teeth and the particularly shallow alveoli suggest that the teeth of ~~genus and sp. indet.~~ NHMD 189993 were not as robust as in other known toothed beaked whales, perhaps still embedded in the gum, as observed in some specimens of extant ziphiids (e.g. *Hyperoodon ampullatus*, *Mesoplodon grayi*; [450,461]).

An enlarged tooth interpreted as a tusk was ~~also found with genus and sp. indet. NHMD 189993~~. This tooth is more massive than the other reduced teeth (Fig. 6G-H). The tusk is triangular, with a root more developed dorsoventrally than the crown. The root of the tooth is transversely compressed with an oval outline (Fig. 5H). As suggested by the outline of the apex of the mandible, the tusk most likely fitted in apical or subapical position ~~onf~~ the mandible. The slightly rounded tip of the crown also suggests that the tusk is slightly worn and ~~as such~~, was originally erupted. The tusk resembles those of long snouted ziphiids, *Berardius* spp. and *T. shepherdii* due to their transverse compression. It differs from the apical tusk present in males *Hyperoodon H. ampullatus* and *Ziphius Z. cavirostris*, which is more conical. It also differs from the genus *Mesoplodon* where the tusk is heavily compressed transversely, ~~including even~~ in species ~~in where which~~ the tusk is in apical or subapical position (*M. mirus* USNM 504612; *M. hectori* NMNZ MM0002901; *M. perrini* USNM 504260).

4.2.2 Postcranial Elements

Hyoid apparatus—The right stylohyal is 238 mm long, 48 mm wide and 26 mm thick (Fig. 6O-Q). The length of this bone is almost twice longer than in *Mesoplodon layardii* (NMNZ 1899: 109 mm; NMNZ 2917: 166 mm). The stylohyal length of ~~NHMD 189993 genus and sp. indet.~~ resemble more the one observed in *Hyperoodon planifrons* (NMNZ 1806: 272 mm; NMNZ DM 1878: 246 mm) and *Ziphius cavirostris* (SNM CN1: 248 mm). ~~The ratio between length and width of the stylohyoid of NHMD 189993 is closer to *Messapicetus gregarius* than *Nazcacetus urbinai* (4.96 in NHMD 189993; 4.63 in *M. gregarius*; 4.10 in *N. urbinai*). This suggests that the stylohyal of *N. urbinai* is wider than long compared to NHMD 189993 and *M. gregarius*. However, the stylohyal of NHMD 189993 is significantly thicker than the one of *M. gregarius*: the ratio between width and thickness equals 1.84 in NHMD 189993 whereas it is 1.39 in *M. gregarius*.~~

A constriction is present on the most anterior part of the stylohyal, at the level of the articulation with the epihyal (Fig 5O). This constriction is observed in the species *Berardius arnuxii* (MNHN A3244) and *Tasmacetus shepherdii* (MM 2908). In lateral view, the stylohyal progressively widens from anterior to posterior and reach its maximum transverse width in the posterior part of the bone. The posterior margin of the bone that articulates with the tympanohyal is pointed (Fig. 5O). The shape of the stylohyal of ~~genus and sp. indet.~~ NHMD 189993 resembles those of *Z. cavirostris* and *H. planifrons*, even though in those species, the transverse widening is more pronounced (between 29 % and 68 % wider), with a flatter dorsal surface. This shape is also observed in *Mesoplodon europaeus* [3427]. It differs from the stylohyal observed in several other species of *Mesoplodon* ~~consulted-examined~~ (*M. bidens* MNHN 1963-259, MNHN 1963-111; *M. europaeus*, NMNZ 550390; *M. layardii* NMNZ 2917), where the lateral and medial margins of the bone ~~stay-are~~ straight, without transverse widening. A ridge ~~run~~ goes along the lateral side of the stylohyal of ~~genus and sp. indet.~~ NHMD 189993 (Fig. 6O), as observed in the ziphiids *Mesoplodon M. europaeus* and *M. mirus* [3427]. This ridge gives a triangular transverse section to the bone.

Humerus—The right humerus is fully preserved (Fig. 6A-C). Further measurements are available in Table 1. It is 204 mm long and 92 mm wide at the level of the deltoid ridge. The ratio between the humeral length and the estimated bizygomatic width (or posterior width of the mandible ~~as~~ used in the specimen) is similar to *Messapicetus gregarius* (in *M. gregarius*: 0.48; in ~~genus and sp. indet.~~ NHMD 189993: 0.50). Both species display a proportionally longer humerus than most extant ziphiids [472]. The head of the humerus is hemispherical. In lateral view (Fig. 6A), the humeral head represents a quarter of the total length of the humerus. In *M. gregarius*, the head is more prominent and anterolaterally oriented: it represents almost a third of the total length of the humerus. In lateral view, the deltoid ridge is well developed along the anterior margin of the humerus (Fig 6A). It develops approximately at mid length of the humerus, and over a third of its length. The presence of a developed deltoid ridge is characteristic of extant Ziphiidae, even though ~~it does not~~ ~~as much~~ developed ~~as~~

much as in Physeteridae [483]. The posterior part of the humerus of ~~genus and sp. indet.~~ NHMD 189993 does not widen, ~~unlike thus differing from the condition observed in~~ many odontocetes [483]. This feature is characteristic of the ziphiid humeri [483]. In posterior view, the articular facets for the radius and the ulna are well separated by a crest (Fig. 6C). Each facet occupies approximately half of the posterior surface of the humerus.

Radius—The associated right radius was originally found articulated with the humerus (Fig. 5D-F). ~~Further measurements are available in Table 2.~~ The radius curves anteroposteriorly. It measures 186 mm long, 63 mm wide at mid-length. The facet for articulation of the humerus is oriented anterodorsally (Fig. 5D-E). Posteriorly, the articulations for the scaphoid and the lunate are well defined; they occupy approximately half of the posterior width of the radius. The articulation for the scaphoid is straight in lateral view, whereas the articulation for the lunate is more oblique and face posterodorsally. In all Ziphiidae, and ~~contrary to differing from~~ other odontocetes, the posterior part of the radius is not widened [483]. The overall shape of the radius does not significantly differ from extant ziphiids ~~consulted-examined~~ (e.g. *Berardius arnuxii* NMNZ 415, MNHN A3244; *Mesoplodon layardii* NMNZ 2917; *Tasmacetus shepherdi* NMNZ MM 2908; *Ziphius cavirostris* SNM CN1). However, its radius is wider than in *Messapicetus gregariusgregarius*, ~~where-in which~~ the ratio between the length and the width of radius equals 0.25 (~~versus~~ 0.34 in ~~genus and sp. indet.~~ NHMD 189993). ~~Further measurements are available in Table 2.~~

Ribs—two partial ribs ~~of NHMD 189993~~ are preserved (Fig. 2C). Their body is heavily fractured and fragmented. Judging from the similar outline of each rib, they were likely from the same pair. They are tentatively inferred to be the pair 2, because of their thick, yet flattened body. Both are double-headed with a marked neck separating the capitulum from the tuberculum.

4.2.3 Size estimates of the specimen

Condylbasal length and bitygomatic width were strongly correlated across the dataset ($R^2=0.78$; Fig. 7). The combination of the two linear measurements was sufficient to separate the four size categories (p -value < 0.0001), and each size category ~~wasere~~ well-distinguishable.

The four size categories were better separated using the bitygomatic width, particularly in the case of the medium-sized and large-sized ziphiids. ~~Indeed, s~~Species from these two categories displayed similar range of variation in condylbasal length, ~~because of due to~~ the strong variability ~~of the of the rostrum~~ anteroposterior length ~~of the rostrum~~ in those species. ~~For example, t~~The large-sized ziphiid *Ziphius cavirostris* displayed a condylbasal length similar to other medium-sized ziphiids, such as *Mesoplodon M. mirus* and *M. europaeus*. ~~This species was easily distinguished from the three other large-sized ziphiids Mesoplodon layardii, Tasmacetus shepherdi and Hyperoodon planifrons~~ (Fig. 7).

On the opposite, the long-snouted medium-sized ziphiids *Messapicetus gregarius*, *Ninoziphius platyrostris*, *Dagonodum moynum*, and *Mesoplodon grayi* displayed condylbasal length matching some large-sized ziphiids (e.g. *Mesoplodon layardii*). ~~One specimen of M. gregarius almost reached the condylbasal length of some of the smallest specimens of very large-sized ziphiids.~~ The long-snouted stem ziphiids were well separated from other ziphiids ~~from species of~~ their size category, including *Mesoplodon grayi*, ~~which~~ ~~The latter also~~ display a strong elongation of the rostrum, but is also characterized by a smaller bitygomatic width. Small-sized ziphiids were easily distinguished from other size categories. They consist ~~of in~~ the ~~living species~~ *Mesoplodon peruvianus* and the fossil ~~species~~ *Nazcacetus urbinai* ~~that possess a slightly longer condylbasal length.~~

~~Medium-sized ziphiids were better differentiated from large-sized ziphiids based on bitygomatic width.~~ Based on the estimate of the bitygomatic width and condylbasal length, ~~genus and sp. indet.~~ NHMD 189993 would be a large ziphiid, between 5.5 and 7.5 m. Its condylbasal length and bitygomatic width is within the range of the extant Ziphiidae *Mesoplodon layardii* whose size generally ranges ~~s~~ between 5.5 and 6 m [494]. Based on the similar condylbasal length, the degree of elongation of the rostrum in ~~genus and sp. indet.~~ NHMD 189993 is likely more similar to *M. layardii* than to the shorter rostrum of *Z. cavirostris* or the extremely elongated rostrum of long-snouted stem ziphiids. The size of ~~genus and sp. indet.~~ NHMD 189993 clearly differs from ~~that of~~ the other fossil ziphiid found in the Gram Formation, *Dagonodum moynum*, a medium-sized ziphiid whose size likely ranged between 4 and 4.5 m.

5. Discussion

5.1 Foraging Ecology

5.1.1 Suction Feeding

All extant beaked whales are specialized ~~to use~~ suction ~~feeders~~: they generate powerful suction pressures with their tongue acting like a piston, to capture and engulf their prey [4]. Many odontocetes can use suction feeding for capturing and/or transporting ~~the~~ prey [450], but extant Ziphiidae are obligate suction feeders, due to the absence of functional teeth to capture their prey [4]. Furthermore, they exhibit a lateral closure of the intraoral cavity combined with a wider and thicker hyoid apparatus compared to odontocetes relying on a more raptorial feeding strategy [4]. The only extant ziphiid species that is perhaps not an obligate suction feeder is *Tasmacetus shepherdi*. Unlike other ziphiids, this species retains a set of erupted teeth likely functional [5146]. Based on one stomach content ~~containing~~ mostly ~~consisting of~~ the fish species

Merluccius hubbsi, MacLeod et al. [13] speculated that this species may be specialized in feeding on deep-water fish rather than cephalopods, thus limiting the competition with other species of beaked whales in the southern oceans where it occurs. Several lines of evidence suggest that ~~genus and sp. indet.~~ NHMD 189993 was capable of using suction feeding to a larger extent than other toothed ziphiids, ~~perhaps already an obligate suction feeding species~~. First, the stylohyal is strongly thickened and elongated. Its anteroposterior length is similar to ~~that of~~ the large ziphiids *Hyperoodon planifrons* and *Ziphius cavirostris*, whereas it is almost twice longer than in *Mesoplodon layardii*, a species close in ~~body~~ size to NHMD 189993. A thickened stylohyal ~~would be~~ necessary to support strong tongue muscles. The styloglossus and the hyoglossus are the two main muscles responsible for the retraction of the tongue in a piston-like manner during suction [3427]. The styloglossus originates on the lateral surface of the stylohyal, which is particularly thickened in the ziphiids *Mesoplodon mirus* and *M. europaeus*. Both species also possess a strong styloglossus: Reidenberg and Laitman [3427] noticed that ~~they~~ ~~these species developed~~ possessed the largest styloglossus relative to total body length from their sample. The thickening of the stylohyal is ~~accompanied by~~ associated with the development ~~pronounced of~~ a ridge along the lateral surface of the bone giving the stylohyal a triangular shape in transverse view [3427]. This ridge was observed in several other ziphiid species (e.g. *Hyperoodon* spp., *Berardius* spp., *Mesoplodon bidens*, *M. layardii*, *Ziphius cavirostris*) and seems characteristic of ~~the ziphiids species specialized to suction feeding~~ the group. The same ridge is present in ~~the genus and sp. indet.~~ NHMD 189993.

The thickening of the stylohyal is accompanied by a reduction of the teeth in ~~genus and sp. indet.~~ NHMD 189993. ~~For~~ On each dentary, the specimen possessed at least 17 alveoli, a number similar to *T. shepherdii* (18–28) [5146], but largely inferior to long snouted stem ziphiids (*Dagonodum mojunum*, 29; *Messapicetus gregarius*, 25–26; *Ninziphius platyrostris*, 40–42) [16–18] ~~or the fossil ziphiid *Chavinziphius maxillocristatus* (at least 50)~~[38]. Additionally, several features of the teeth and the alveoli differ between the aforementioned species and ~~genus and sp. indet.~~ NHMD 189993. The alveoli of the ~~specimen~~ latter, although individualized, are particularly shallow, greatly differing from the condition observed in other toothed beaked whales where the alveolar groove is deep, but the septa are not necessarily well differentiated (e.g. *M. gregarius*; Bianucci et al., 2010)[16]. The teeth themselves are also ~~well~~ reduced in size compared to other toothed ziphiids including *T. shepherdii*. ~~With the exclusion of the tusk,~~ the longest tooth of ~~genus and sp. indet.~~ NHMD 189993 (~~excepting the tusk~~) measures 20 mm with a maximum diameter of 11 mm. The tooth measurements of the specimen are even smaller than in the medium-sized ziphiids *D. mojunum*, *M. gregarius*, and *N. platyrostris*. Furthermore, in ziphiids with functional teeth, the robust crown shows apical wear or interlocking facets suggesting that their dentition was functional. This is not the case in ~~genus and sp. indet.~~ NHMD 189993 where the small crown does not show sign of interlocking. In many teeth, the apex is broken off and does not allow ~~for~~ an estimation of the degree of apical wear, but the few teeth with a preserved apex do not show signs of wear. Therefore, we hypothesize that the small teeth of NHMD 189993 were either embedded ~~from within~~ the gum or too small to be used ~~as a regular~~ ordinarily method ~~of~~ for capturing the prey. ~~Nevertheless,~~ ~~w~~We do not discard the possibility that NHMD 189993 could ~~have~~ occasionally used its reduced teeth (if erupted) to manipulate or capture some of its prey.

5.1.2.2 A potential case of niche separation

Several morphological evidences and the different size estimates of *Dagonodum mojunum* and ~~genus and sp. indet.~~ NHMD 189993 suggest that these two species occupied two different ecological niches. ~~Despite the relatively inaccurate age estimation of the Gram Formation (7.2–9.9 Ma), study of the variation of accumulation rates suggests that sediments were deposited during approximately 120 000 years [25]. Therefore, *D. mojunum* and NHMD 189993, both found in the Gram claypit, were likely coeval. The mollusc faunas found in association with NHMD 189993 indicates that it was found in the biozone V (the uppermost part of the Gram Formation), whereas *D. mojunum* was found in the biozone III, IV, or V[18]. Since Beyer [25] observed a significant increase in the uppermost 8 m of the formation, there is a possibility that the two specimens were not separated from more than 20 000 years. It is possible that the two species were not contemporary: despite the similar age estimation for the two species, the error margin of 2.7 Ma could mean that they did not live at the same period of time.~~ Assuming that the two species were contemporary, the co-occurrence of two different sized species of Ziphiidae at the same location suggests a case of niche separation.

Cases of niche separation are known in extant ziphiids: *Mesoplodon* species consistently feed on smaller prey type (generally, cephalopods under 500 g) compared to *Hyperoodon* and *Ziphius* species (cephalopods over 1 kg) [13]. The difference of prey size targeted may explain why species of *Mesoplodon* are often sympatric with the latter [5247–550]. Size is not the only component, even though an important one [12], allowing niche separation between ziphiid species. In the case of *D. mojunum* and ~~genus and sp. indet.~~ NHMD 189993, the difference in specialization to suction feeding reinforces this hypothesis. The species *D. mojunum* possesses some adaptations to suction feeding (transverse thickening of the basyhyal and thyrohyal; presence of a precoronoid crest) [18], but not to the extent of ~~genus and sp. indet.~~ NHMD 189993, that probably relied more prominently on this feeding strategy. Obligate suction feeders with a reduced tooth count are often more teuthophagous [4,564], even though some ziphiids can still feed on fish [13]. Perhaps, the more specialized oral apparatus of ~~genus and sp. indet.~~ NHMD 189993 is more indicative of a more predominantly teuthophagous diet than *D. mojunum*. Interestingly, the fossil of a cuttlefish (Sepiida) was found in the Gram Formation (MSM DK718; unpublished data), an ~~ideal~~ possible prey type for ~~genus and sp. indet.~~ NHMD 189993.

~~Alternatively, the two species may have segregated geographically: whereas *D. mojunum* is assumed to be a local [18], the other specimen ~~genus and sp. indet.~~ NHMD 189993 may have been more adapted to deep diving like its modern representatives. No morphological features preserved allow the evaluation of the diving abilities of ~~genus and sp. indet.~~ NHMD 189993. Therefore, this hypothesis cannot be fully ruled out.~~

Other cases of niche separation between fossil ziphiids probably occurred at other locations where they show a diversity of sizes or feeding strategies. Bianucci et al. [572] already proposed this interpretation to explain the high diversity of fossil ziphiids trawled from the seafloor off South Africa. Fossil ziphiids from the Neogene of Antwerp and fished from the Atlantic Ocean floor off the Iberian Peninsula also exhibit a great diversity-range of skull sizes, that-which could be indicative of ecological niche segregation [354,337]. In absence of precise datation for these three localities, it is unclear whether the different species were living during the same time spanperiod.

Conclusion

Despite the rich fossil record of beaked whales, the discovery of postcranial material remains still represents a rare finding [18,4237,583]. A new fossil of Ziphiidae, genus and sp. indet. NHMD 189993, consisting of the mandible, earbones, the stylohyal, isolated teeth including the tusk, the right humerus and associated radius is described here. This fossil is dated to the mid- to late Tortonian (ca. 9.9-7.2 Ma). Despite the lack of cranial material for comparing with other similarly sized fossil ziphiids, genus and sp. indet. NHMD 189993 (here referred to Ziphiidae gen. and sp. indet.) clearly differs from the other species found in known from the Gram Formation during the same time period, *Dagonodum mojnum*.

Unlike *D. mojnum* and other long-snouted stem ziphiids, the morphology of the oral apparatus of genus and sp. indet. NHMD 189993 suggests that it was mostly relying on well adapted for suction feeding. The reduced teeth perhaps were possibly still embedded in the gum, and morphological features of the thickened stylohyal supports this interpretation.

The two fossil species *Dagonodum mojnum* and genus and sp. indet. NHMD 189993 likely occupied different ecological niches with genus and sp. indet. NHMD 189993 likely feeding relying more predominantly on evasive prey such as cephalopods. Assuming the two species that D. mojnum and NHMD 189993 were chronologically concomitant, the spatial co-occurrence of the species D. mojnum and genus and sp. indet. NHMD 189993 se two species can be illustrative of a case of niche separation. Together with sexual dimorphism [594], the specialization toward specific ecological niches in Ziphiidae may partly explain the rich specific diversity of this family.

Acknowledgments

We thank Mette Steeman for her suggestions and support for the completion of the project. We are also indebted to Frank Osbæk and Trine Sørensen that were in charge of the preparation of the specimen.

We thank the following colleagues for kindly allowing us to access some of the comparative material we used in this study: Morten Tange Olsen and Daniel Klingberg Johansson to access the SNM collection, Anne-Lise Folie for the IRSNB collection, Christian de Muizon and Christine Lefèvre for the MNHN collection, Chiara Sorbini for the MSNUP collection, Thomas Schultz for the NMNZ, and Charles Potter for the USNM collection.

We are also grateful for the useful feedbacks of the associate editor of RSOS, Giovanni Bianucci and the suggestions of an anonymous reviewer.

Ethics statements

This does not concern this research.

Funding Statement

The project was funded by the Dansk Slots- og Kulturstyrelsen (FORM.2016-0021).

Data Accessibility

The datasets supporting this article have been uploaded are available as part of the Supplementary Material at the following address from Figshare: <https://figshare.com/s/c75c55bdca6ef3ae3a6f>

Competing Interests

We have no competing interests. 'We have no competing interests.'

Authors' Contributions

B.R. identified and interpreted the ziphiid remains. H.L. created the digital reconstruction and the associated figures. B.R. made calculations for the size estimates and collected the measurements. Both authors participated in the preparation of the illustrations and the writing of the manuscript. Both discussed the results and commented on the manuscript at all stages.

References

1. Dalebout ML et al. 2014 Resurrection of *Mesoplodon hotaula* Deraniyagala 1963: A new species of beaked whale in the tropical Indo-Pacific. *Marine Mammal Science* **30**, 1081–1108. (doi:10.1111/mms.12113)
2. Morin PA et al. 2017 Genetic structure of the beaked whale genus *Berardius* in the North Pacific, with genetic evidence for a new species. *Marine Mammal Science* **33**, 96–111. (doi:10.1111/mms.12345)
3. Clarke MR. 1996 Cephalopods as prey. III. Cetaceans. *Philosophical Transactions of the Royal Society of London. Series B: Biological Sciences* **351**, 1053–1065. (doi:10.1098/rstb.1996.0093)
4. Heyning JE, Mead JG. 1996 Suction feeding in beaked whales: morphological and observational evidence. *Natural History Museum of Los Angeles County Contributions in Science* **464**, 1–12.
5. Hooker Sascha K., Baird Robin W. 1999 Deep-diving behaviour of the northern bottlenose whale, *Hyperoodon ampullatus*

- (Cetacea: Ziphiidae). *Proceedings of the Royal Society of London. Series B: Biological Sciences* **266**, 671–676. (doi:10.1098/rspb.1999.0688)
6. MacLeod CD *et al.* 2006 Known and inferred distributions of beaked whales species (Cetacea: Ziphiidae). *J. Cetacean Res. Manage.* **7**, 271–286.
7. Johnson Mark, Madsen Peter T., Zimmer Walter M. X., Aguilar de Soto Natacha, Tyack Peter L. 2004 Beaked whales echolocate on prey. *Proceedings of the Royal Society of London. Series B: Biological Sciences* **271**, S383–S386. (doi:10.1098/rsbl.2004.0208)
8. Tyack PL, Johnson M, Soto NA, Sturlese A, Madsen PT. 2006 Extreme diving of beaked whales. *J. Exp. Biol.* **209**, 4238. (doi:10.1242/jeb.02505)
9. Minamikawa S, Iwasaki T, Kishiro T. 2007 Diving behaviour of a Baird's beaked whale, *Berardius bairdii*, in the slope water region of the western North Pacific: first dive records using a data logger. *Fisheries Oceanography* **16**, 573–577. (doi:10.1111/j.1365-2419.2007.00456.x)
10. Schorr GS, Falcone EA, Moretti DJ, Andrews RD. 2014 First Long-Term Behavioral Records from Cuvier's Beaked Whales (*Ziphius cavirostris*) Reveal Record-Breaking Dives. *PLoS ONE* **9**, e92633. (doi:10.1371/journal.pone.0092633)
11. Moore JC. 1968 Relationships among the living genera of beaked whales. *Fieldiana: Zoology* **53**, 209–298.
12. MacLeod CD, Santos MB, López A, Pierce GJ. 2006 Relative prey size consumption in toothed whales: implications for prey selection and level of specialisation. *Mar Ecol Prog Ser* **326**, 295–307. (doi:10.3354/meps326295)
13. MacLeod CD, Santos MB, Pierce GJ. 2003 Review of Data on Diets of Beaked Whales: Evidence of Niche Separation and Geographic Segregation. *Journal of the Marine Biological Association of the United Kingdom* **83**, 651–665. (doi:10.1017/S0025315403007616h)
14. Hocking David P., Marx Felix G., Park Travis, Fitzgerald Erich M. G., Evans Alistair R. 2017 A behavioural framework for the evolution of feeding in predatory aquatic mammals. *Proceedings of the Royal Society B: Biological Sciences* **284**, 20162750. (doi:10.1098/rspb.2016.2750)
15. Bianucci G, Landini W, Varola A. 1994 Relationships of *Messapicetus longirostris* (Cetacea, Ziphiidae) from the Miocene of South Italy. *Bollettino della Società Paleontologica Italiana* **33**, 231–241.
16. Bianucci G, Lambert O, Post K. 2010 High concentration of long-snouted beaked whales (genus *Messapicetus*) from the Miocene of Peru. *Palaeontology, Wiley* **53**, 1077–1098.
17. Lambert O, de Muizon C, Bianucci G. 2013 The most basal beaked whale *Ninoziphius platyrostris* Muizon, 1983: clues on the evolutionary history of the family Ziphiidae (Cetacea: Odontoceti). *Zoological Journal of the Linnean Society* **167**, 569–598. (doi:10.1111/zoj.12018)
18. Ramassamy B. 2016 Description of a new long-snouted beaked whale from the Late Miocene of Denmark: evolution of suction feeding and sexual dimorphism in the Ziphiidae (Cetacea: Odontoceti). *Zoological Journal of the Linnean Society* **178**, 381–409. (doi:10.1111/zoj.12418)
19. Bianucci G, Llàcer S, Cardona JQ, Collareta A, Florit AR. 2019 A new beaked whale record from the upper Miocene of Menorca, Balearic Islands, based on CT-scan analysis of limestone slabs. *Acta Palaeontologica Polonica* **64**, 291–302.
20. Rasmussen LB. 1966 Molluscan faunas and biostratigraphy of the marine younger Miocene formations in Denmark. Part I: Geology and biostratigraphy. *Geological survey of Denmark* **88**, 1–358.
21. Rasmussen ES, Dybkjær K, Piasecki S. 2010 Lithostratigraphy of the upper Oligocene–Miocene succession of Denmark. *Geological Survey of Denmark and Greenland Bulletin* **22**, 92.
22. Rasmussen LB. 1956 *The marine Upper Miocene of South Jutland and its molluscan fauna*. Denmark: Danmarks Geologiske Undersøgelse II.
23. Rasmussen ES. 2005 The geology of the upper Middle-Upper Miocene Gram Formation in the Danish area. *Paleontos*, 5–18.
24. Piasecki S. 2005 Dinoflagellate cysts of the Middle-Upper Miocene Gram Formation, Denmark. *Palaeontos* **7**, 29–45.
25. Beyer C. 2005 A magnetic analysis of the Late Miocene Gram Formation, Denmark. *Palaeontos* **7**, 19–28.
26. Lambert O, Bianucci G, Post K. 2009 A new beaked whale (Odontoceti, Ziphiidae) from the middle Miocene of Peru. *Journal of Vertebrate Paleontology* **29**, 910–922. (doi:10.1671/039.029.0304)
27. Bianucci G, Lambert O, Post K. 2007 A high diversity in fossil beaked whales (Mammalia, Odontoceti, Ziphiidae) recovered by trawling from the sea floor off South Africa. *Geodiversitas* **29**, 561–618.
28. Jolicoeur P. 1963 193. Note: the multivariate generalization of the allometry equation. *Biometrics* **19**, 497–499.
29. Marcus LF. 1990 Chapter 4: Traditional Morphometrics. In *Proceedings of the Michigan morphometrics workshop* (eds FJ Rohlf, FL Bookstein), pp. 77–122. Michigan: University of Michigan Museum of Zoology.
30. R Core Team. 2019 *R: A language and environment for statistical computing*. Vienna, Austria: R Foundation for Statistical Computing. See <https://www.R-project.org/>.
31. Mead JG, Fordyce RE. 2009 *The Therian Skull: a Lexicon with Emphasis on the Odontocetes*. Washington D.C.: Smithsonian Institution Scholarly Press.
32. Fitzgerald EM. 2016 A late Oligocene waipatiid dolphin (Odontoceti: Waipatiidae) from Victoria, Australia. *Memoirs of Museum Victoria* **74**, 117–136.
33. Marx F, Lambert O, Uhen MD. 2016 *Cetacean Paleobiology*. Chichester: John Wiley & Sons.
34. Reidenberg JS, Laitman JT. 1994 Anatomy of the hyoid apparatus in odontoceli (toothed whales): Specializations of their skeleton and musculature compared with those of terrestrial mammals. *The Anatomical Record* **240**, 598–624. (doi:10.1002/ar.1092400417)
35. Lambert O. 2005 Systematics and phylogeny of the fossil beaked whales *Ziphirostrum* du Bus, 1868 and *Choneziphius* Duvernoy, 1851 (Mammalia, Cetacea, Odontoceti) from the Neogene of Antwerp (North of Belgium). *Geodiversitas* **27**, 443–497.
36. Bianucci G, Miján I, Lambert O, Post K, Mateus O. 2013 Bizarre fossil beaked whales (Odontoceti, Ziphiidae) fished from the Atlantic Ocean floor off the Iberian Peninsula. *Geodiversitas* **35**, 105–153.
37. Lambert O, Louwyte S. 2006 *Archaeoziphius microglenoideus*, a new primitive beaked whale (Mammalia, Cetacea, odontoceti) from the Middle Miocene of Belgium. *Journal of Vertebrate Paleontology* **26**, 182–191. (doi:10.1671/0272-4634(2006)26[182:AMANPB]2.0.CO;2)
38. Bianucci G, Di Celma C, Urbina M, Lambert O. 2016 New beaked whales from the late Miocene of Peru and evidence for convergent evolution in stem and crown Ziphiidae (Cetacea, Odontoceti). *PeerJ*, e2479.
39. Barnes LG. 1976 Outline of eastern North Pacific fossil cetacean assemblages. *Systematic Zoology* **25**, 321–343.
40. Fordyce RE, de Muizon C. 2001 Evolutionary history of cetaceans: a review. Secondary Adaptation of Tetrapods to Life in Water. In *Secondary Adaptation of Tetrapods to Life in Water* (eds J-M Mazin, V de Buffrénil), pp. 169–212. München.
41. Galatius A, Kinze CC. 2003 Ankylosis patterns in the postcranial skeleton and hyoid bones of the harbour porpoise (*Phocoena phocoena*) in the Baltic and North Sea. *Canadian Journal of Zoology* **81**, 1851–1861.
42. Bianucci G, Collareta A, Post K, Varola A, Lambert O. 2016 A New Record of *Messapicetus* from the Pietra Leccese (Late Miocene, Southern Italy): Antitropical Distribution in a Fossil Beaked Whale (Cetacea, Ziphiidae). *Rivista Italiana di Paleontologia Stratigrafia* **122**, 63–73.
43. Dalebout ML, Mead JG, Baker CS, Baker AN, van Helden AL. 2002 A New Species of beaked whale *Mesoplodon perrini* sp. n. (Cetacea: Ziphiidae) discovered through Phylogenetic Analyses of Mitochondrial DNA sequences. *Marine Mammal Science* **18**, 577–608.

44. Cope ED. 1869 Two extinct Mammalia from the United States. *Proceedings of the American Philosophical Society* **11**, 188–190.
45. Flower WH. 1882 On the Whales of the Genus *Hyperoodon*. In *Proceedings of the Zoological Society of London*, pp. 722–726. Oxford, UK: Blackwell Publishing Ltd.
46. Boschma H. 1951 Rows of small teeth in ziphioid whales. *Zoologische Mededelingen* **31**, 130–148.
47. Lambert O, Collareta A, Landini W, Post K, Ramassamy B, Di Celma C, Urbina M, Bianucci G. 2015 No deep diving: evidence of predation on epipelagic fish for a stem beaked whale from the Late Miocene of Peru. *Proceedings of the Royal Society B: Biological Sciences* **282**, 20151530. (doi:10.1098/rspb.2015.1530)
48. Benke H. 1993 Investigations on the osteology and the functional morphology of the flipper of whales and dolphins (Cetacea). *Investigations on Cetacea* **24**, 9–252.
49. Ross JGB. 1984 The smaller cetaceans of the south east coast of Southern Africa. *Annals of the Cape Provincial Museums* **15**, 173–410.
50. Werth AJ. 2006 Mandibular and Dental Variation and the Evolution of Suction Feeding in Odontoceti. *Journal of Mammalogy* **87**, 579–588. (doi:10.1644/05-MAMM-A-279R1.1)
51. Mead JG, Payne RS. 1975 A specimen of the Tasman beaked whale, *Tasmacetus shepherdi*, from Argentina. *Journal of Mammalogy*, 213–218.
52. Heyning JE, Ridgway SH, Harrison R. 1989 Cuvier's beaked whale *Ziphius cavirostris* G. Cuvier, 1823. In *Handbook of marine mammals*, pp. 289–320. London: Academic Press.
53. Mead JG. 1989 Beaked whales of the genus *Mesoplodon*. In *Handbook of marine mammals* (eds SH Ridgway, R Harrison), pp. 349–430. London: Academic Press.
54. Mead JG. 1989 Bottlenose whales *Hyperoodon ampullatus* (Forster, 1770) and *Hyperoodon planifrons* (Flower 1882). In *Handbook of marine mammals* (eds SH Ridgway, R Harrison), pp. 321–348. London: Academic Press.
55. MacLeod CD. 2000 Distribution of beaked whales of the genus *Mesoplodon* in the North Atlantic. *Mammal Review* **30**, 1–8.
56. Clarke MR. 1986 Cephalopods in the diets of odontocetes. In *Research on Dolphins* (eds MM Bryden, R Harrison), pp. 281–321. Oxford: Clarendon Press.
57. Bianucci G, Lambert O, Post K. 2008 Beaked whale mysteries revealed by seafloor fossils trawled off South Africa. *South African Journal of Science* **104**.
58. de Muizon C. 1984 *Les vertébrés fossiles de la Formation Pisco (Pérou)*, deuxième partie: les Odontocètes (Cetacea, Mammalia) du Pliocène inférieur de Sud-Sacaco. Travaux de l'Institut Français d'Etudes Andines.
59. Dalebout ML, Steel D, Baker CS. 2008 Phylogeny of the Beaked Whale Genus *Mesoplodon* (Ziphiidae: Cetacea) Revealed by Nuclear Introns: Implications for the Evolution of Male Tusks. *Systematic Biology* **57**, 857–875. (doi:10.1080/10635150802559257)
1. Dalebout ML *et al.* 2014 Resurrection of *Mesoplodon hotaula* Deraniyagala 1963: A new species of beaked whale in the tropical Indo-Pacific. *Marine Mammal Science* **30**, 1081–1108. (doi:10.1111/mms.12113)
2. Morin PA *et al.* 2017 Genetic structure of the beaked whale genus *Berardius* in the North Pacific, with genetic evidence for a new species. *Marine Mammal Science* **33**, 96–111. (doi:10.1111/mms.12345)
3. Clarke MR. 1996 Cephalopods as prey. III. Cetaceans. *Philosophical Transactions of the Royal Society of London. Series B: Biological Sciences* **351**, 1053–1065. (doi:10.1098/rstb.1996.0093)
4. Heyning JE, Mead JG. 1996 Suction feeding in beaked whales: morphological and observational evidence. *Natural History Museum of Los Angeles County Contributions in Science* **464**, 1–12.
5. Hooker Sasecha K., Baird Robin W. 1999 Deep-diving behaviour of the northern bottlenose whale, *Hyperoodon ampullatus* (Cetacea: Ziphiidae). *Proceedings of the Royal Society of London. Series B: Biological Sciences* **266**, 671–676. (doi:10.1098/rspb.1999.0688)
6. MacLeod CD *et al.* 2006 Known and inferred distributions of beaked whales species (Cetacea: Ziphiidae). *J. Cetacean Res. Manage.* **7**, 271–286.
7. Johnson-Mark, Madsen Peter T., Zimmer Walter M. X., Aguilar de Soto Natacha, Tyack Peter L. 2004 Beaked whales echolocate on prey. *Proceedings of the Royal Society of London. Series B: Biological Sciences* **271**, S383–S386. (doi:10.1098/rsbl.2004.0208)
8. Tyack PL, Johnson M, Soto NA, Sturlese A, Madsen PT. 2006 Extreme diving of beaked whales. *J. Exp. Biol.* **209**, 4238. (doi:10.1242/jeb.02505)
9. Minamikawa S, Iwasaki T, Kishiro T. 2007 Diving behaviour of a Baird's beaked whale, *Berardius bairdii*, in the slope water region of the western North Pacific: first dive records using a data logger. *Fisheries Oceanography* **16**, 573–577. (doi:10.1111/j.1365-2419.2007.00456.x)
10. Schorr GS, Falcone EA, Moretti DJ, Andrews RD. 2014 First Long-Term Behavioral Records from Cuvier's Beaked Whales (*Ziphius cavirostris*) Reveal Record-Breaking Dives. *PLoS ONE* **9**, e92633. (doi:10.1371/journal.pone.0092633)
11. Moore JC. 1968 Relationships among the living genera of beaked whales. *Feldiana: Zoology* **53**, 209–298.
12. MacLeod CD, Santos MB, López A, Pierce GJ. 2006 Relative prey size consumption in toothed whales: implications for prey selection and level of specialisation. *Mar Ecol Prog Ser* **326**, 295–307. (doi:10.3354/meps326295)
13. MacLeod CD, Santos MB, Pierce GJ. 2003 Review of Data on Diets of Beaked Whales: Evidence of Niche Separation and Geographic Segregation. *Journal of the Marine Biological Association of the United Kingdom* **83**, 651–665. (doi:10.1017/S0025315403007616h)
14. Hocking David P., Marx Felix G., Park Travis, Fitzgerald Erich M. G., Evans Alistair R. 2017 A behavioural framework for the evolution of feeding in predatory aquatic mammals. *Proceedings of the Royal Society B: Biological Sciences* **284**, 20162750. (doi:10.1098/rspb.2016.2750)
15. Bianucci G, Landini W, Varola A. 1994 Relationships of *Mesaspicetus longirostris* (Cetacea, Ziphiidae) from the Miocene of South Italy. *Bollettino della Società Paleontologica Italiana* **33**, 231–241.
16. Bianucci G, Lambert O, Post K. 2010 High concentration of long-snouted beaked whales (genus *Mesaspicetus*) from the Miocene of Peru. *Palaeontology*, Wiley **53**, 1077–1098.
17. Lambert O, de Muizon C, Bianucci G. 2013 The most basal beaked whale *Ninziphius platyrostris* Muizon, 1983: clues on the evolutionary history of the family Ziphiidae (Cetacea: Odontoceti). *Zoological Journal of the Linnean Society* **167**, 569–598. (doi:10.1111/zoj.12018)
18. Ramassamy B. 2016 Description of a new long-snouted beaked whale from the Late Miocene of Denmark: evolution of suction feeding and sexual dimorphism in the Ziphiidae (Cetacea: Odontoceti). *Zoological Journal of the Linnean Society* **178**, 381–409. (doi:10.1111/zoj.12418)
19. Lambert O, Bianucci G, Post K. 2009 A new beaked whale (Odontoceti, Ziphiidae) from the middle Miocene of Peru. *Journal of Vertebrate Paleontology* **29**, 910–922. (doi:10.1671/039.029.0304)
20. Bianucci G, Lambert O, Post K. 2007 A high diversity in fossil beaked whales (Mammalia, Odontoceti, Ziphiidae) recovered by trawling from the sea floor off South Africa. *Geodiversitas* **29**, 561–618.
21. Jolicoeur P. 1963–193. Note: the multivariate generalization of the allometry equation. *Biometrics* **19**, 497–499.
22. Marcus LF. 1990 Chapter 4: Traditional Morphometrics. In *Proceedings of the Michigan morphometrics workshop* (eds FJ Rohlf, FL Bookstein), pp. 77–122. Michigan: University of Michigan Museum of Zoology.

23. R-Core Team. 2019 *R: A language and environment for statistical computing*. Vienna, Austria: R Foundation for Statistical Computing. See <https://www.R-project.org/>.
24. Mead JG, Fordyce RE. 2009 *The Therian Skull: a Lexicon with Emphasis on the Odontocetes*. Washington D.C.: Smithsonian Institution Scholarly Press.
25. Fitzgerald EM. 2016 A late Oligocene waipatiid dolphin (Odontoceti: Waipatiidae) from Victoria, Australia. *Memoirs of Museum Victoria* **74**, 117–136.
26. Marx F, Lambert O, Uhen MD. 2016 *Cetacean Paleobiology*. Chichester: John Wiley & Sons.
27. Reidenberg JS, Laitman JT. 1994 Anatomy of the hyoid apparatus in odontoceti (toothed whales): Specializations of their skeleton and musculature compared with those of terrestrial mammals. *The Anatomical Record* **240**, 598–624. (doi:10.1002/ar.1092400417)
28. Rasmussen LB. 1966 Molluscan faunas and biostratigraphy of the marine younger Miocene formations in Denmark. Part I: Geology and biostratigraphy. *Geological survey of Denmark* **88**, 1–358.
29. Piasecki S. 2005 Dinoflagellate cysts of the Middle Upper Miocene Gram Formation, Denmark. *Palaeontos* **7**, 29–45.
30. Beyer C. 2005 A magnetic analysis of the Late Miocene Gram Formation, Denmark. *Palaeontos* **7**, 19–28.
31. Lambert O. 2005 Systematics and phylogeny of the fossil beaked whales *Ziphirostrum du Bus*, 1868 and *Choneziphius Duvernoy*, 1851 (Mammalia, Cetacea, Odontoceti) from the Neogene of Antwerp (North of Belgium). *Geodiversitas* **27**, 443–497.
32. Lambert O, Louwe S. 2006 *Archaeoziphius microglenoideus*, a new primitive beaked whale (Mammalia, Cetacea, odontoceti) from the Middle Miocene of Belgium. *Journal of Vertebrate Paleontology* **26**, 182–191. (doi:10.1671/0272-4634(2006)26[182:AMANPB]2.0.CO;2)
33. Bianucci G, Miján I, Lambert O, Post K, Mateus O. 2013 Bizarre fossil beaked whales (Odontoceti, Ziphiidae) fished from the Atlantic Ocean floor off the Iberian Peninsula. *Geodiversitas* **35**, 105–153.
34. Barnes LG. 1976 Outline of eastern North Pacific fossil cetacean assemblages. *Systematic Zoology* **25**, 321–343.
35. Fordyce RE, de Muizon C. 2001 Evolutionary history of cetaceans: a review. Secondary Adaptation of Tetrapods to Life in Water. In *Secondary Adaptation of Tetrapods to Life in Water* (eds J-M Mazin, V de Bruffrénil), pp. 169–212. München.
36. Galatius A, Kinze CC. 2003 Ankylosis patterns in the postcranial skeleton and hyoid bones of the harbour porpoise (*Phocoena phocoena*) in the Baltic and North Sea. *Canadian Journal of Zoology* **81**, 1851–1861.
37. Bianucci G, Collareta A, Post K, Varola A, Lambert O. 2016 A New Record of *Messapicetus* from the Pietra Leccese (Late Miocene, Southern Italy): Antitropical Distribution in a Fossil Beaked Whale (Cetacea, Ziphiidae). *Rivista Italiana di Paleontologia-Stratigrafia* **122**, 63–73.
38. Dalebout ML, Mead JG, Baker CS, Baker AN, van Helden AL. 2002 A New Species of beaked whale *Mesoplodon perrini* sp. n. (Cetacea: Ziphiidae) discovered through Phylogenetic Analyses of Mitochondrial DNA sequences. *Marine Mammal Science* **18**, 577–608.
39. Cope ED. 1869 Two extinct Mammalia from the United States. *Proceedings of the American Philosophical Society* **11**, 188–190.
40. Flower WH. 1882 On the Whales of the Genus *Hyperoodon*. In *Proceedings of the Zoological Society of London*, pp. 722–726. Oxford, UK: Blackwell Publishing Ltd.
41. Bosehna H. 1951 Rows of small teeth in ziphioid whales. *Zoologische Mededelingen* **31**, 130–148.
42. Lambert O, Collareta A, Landini W, Post K, Ramassamy B, Di Celma C, Urbina M, Bianucci G. 2015 No deep diving: evidence of predation on epipelagic fish for a stem beaked whale from the Late Miocene of Peru. *Proceedings of the Royal Society B: Biological Sciences* **282**, 20151530. (doi:10.1098/rspb.2015.1530)
43. Benke H. 1993 Investigations on the osteology and the functional morphology of the flipper of whales and dolphins (Cetacea). *Investigations on Cetacea* **24**, 9–252.
44. Ross JGB. 1984 The smaller cetaceans of the south-east coast of Southern Africa. *Annals of the Cape Provincial Museums* **15**, 173–410.
45. Werth AJ. 2006 Mandibular and Dental Variation and the Evolution of Suction Feeding in Odontoceti. *Journal of Mammalogy* **87**, 579–588. (doi:10.1644/05-MAMM-A-279R1.1)
46. Mead JG, Payne RS. 1975 A specimen of the Tasman beaked whale, *Tasmacetus shepherdi*, from Argentina. *Journal of Mammalogy*, 213–218.
47. Heyning JE, Ridgway SH, Harrison R. 1989 Cuvier's beaked whale *Ziphius cavirostris* G. Cuvier, 1823. In *Handbook of marine mammals*, pp. 289–320. London: Academic Press.
48. Mead JG. 1989 Bottlenose whales *Hyperoodon ampullatus* (Forster, 1770) and *Hyperoodon planifrons* Flower 1882. In *Handbook of marine mammals* (eds SH Ridgway, R Harrison), pp. 321–348. London: Academic Press.
49. Mead JG. 1989 Beaked whales of the genus *Mesoplodon*. In *Handbook of marine mammals* (eds SH Ridgway, R Harrison), pp. 349–430. London: Academic Press.
50. MacLeod CD. 2000 Distribution of beaked whales of the genus *Mesoplodon* in the North Atlantic. *Mammal Review* **30**, 1–8.
51. Clarke MR. 1986 Cephalopods in the diets of odontocetes. In *Research on Dolphins* (eds MM Bryden, R Harrison), pp. 281–321. Oxford: Clarendon Press.
52. Bianucci G, Lambert O, Post K. 2008 Beaked whale mysteries revealed by seafloor fossils trawled off South Africa. *South African Journal of Science* **104**.
53. de Muizon C. 1984 *Les vertébrés fossiles de la Formation Pisco (Pérou). deuxième partie: les Odontocètes (Cetacea, Mammalia) du Pliocène inférieur de Sud-Saeco*. Travaux de l'Institut Français d'Etudes Andines.
54. Dalebout ML, Steel D, Baker CS. 2008 Phylogeny of the Beaked Whale Genus *Mesoplodon* (Ziphiidae: Cetacea) Revealed by Nuclear Introns: Implications for the Evolution of Male Tusks. *Systematic Biology* **57**, 857–875. (doi:10.1080/10635150802559257)

Tables

TABLE 1. Measurements of the mandible, cranial remains, and forelimb bones of the specimen *genus and sp. indet.* NHMD 189993.

Feature	NHMD 189993
Mandibles	
Anteroposterior length as preserved	1032
Maximum posterior width as preserved	412
Symphyseal portion anteroposterior length	289
Symphyseal portion maximal transverse width	61

Humerus

Anteroposterior humeral head diameter	73
Maximal humeral length	204
Maximal distal width	80
Width at the level of the deltoid ridge	92

Radius

Maximal radius length	186
Width at mid-length	63
Proximal width	67
Distal width	73

Measurements in mm

TABLE 2. Measurements of the periotic and the tympanic bone of ~~genus and sp. indet.~~ NHMD 189993.

Feature	NHMD 189993
Right Periotic	
Maximal anteroposterior length	47
Maximal transverse width	33
Pars cochlearis maximum anteroposterior length	25
Pars cochlearis maximum transverse width	27
Anterior process maximum anteroposterior length	21
Anterior process maximum transverse width	20
Posterior process maximum anteroposterior length	18
Posterior process maximum transverse width	21
Lateral tuberosity in lateral view transverse width	10
Right Tympanic bulla	
Tympanic anteroposterior length	43
Tympanic maximum transverse width	26
Inner posterior prominence maximum transverse width	11
Outer posterior prominence maximum width	15
Dorsoventral height in lateral view as preserved	21
Involucrum indentation on the tympanic	
Dorsoventral height in medial view	12

All measurements are in mm and taken in ventral view unless noted otherwise.

Figures

Figure 1. Current extension of the outcrops of the Gram Formation in Denmark (shaded area). The type locality finding site is situated in the Gram claypit, 1.5 km north of Gram. Modified from Rasmussen [28].

Figure 2. Mandibles, cranial, and postcranial remains of *genus and sp. indet.*-NHMD 189993. **A₅**, ventral view; **B₅**, corresponding drawing; **C**, detail of the preserved nasal and other postcranial remains.

Figure 3. Digital reconstruction of the mandible and postcranial remains of *genus and sp. Indet.* NHMD 189993. **A₅**, dorsal view; **B₅**, lateral view; **C**, detail of the anterodorsal part of the mandible.

Figure 4. Right periotic of *genus and sp. indet.*-NHMD 189993. **A₅**, dorsal view; **B₅**, lateral view; **C₅**, medial view; **D₅₋₂**, ventral view; **E₅**-**H₅**, corresponding drawings.

Figure 5. Right tympanic of *genus and sp. indet.* NHMD 189993. **A₅**, dorsal view; **B₅₋₂**, medial view; **C₅₋₂**, ventral view; **D₅₋₂**, lateral view.

Figure 6. Postcranial elements of *genus and sp. indet.* NHMD 189993. Right humerus in **A₅**, lateral view; **B₅₋₂**, dorsal view; **C₅**, posterior view. Associated right radius **D₅** in lateral view; **E₅**, medial view; **F₅**, dorsal view; isolated tusk in **G₅₋₂**, lateral view; **H₅₋₂**, ventral view; three isolated teeth in **I₅₋₂**, **K₅**, **M₅**, medial view; **J₅₋₂**, **L₅₋₂**, **N₅**, lateral view; right stylohyal in **QA₅**, lateral view; **P₅**, **B₅**, medial view; **Q₅**, **C₅**, dorsal view.

Figure 7. Log-transformed bizygomatic width plotted against condylobasal length in *extinct and extant* Ziphiidae. **Abbreviations:** **Bear:** *Berardius arnuxii*; **Beba:** *Berardius bairdii*; **Hyam:** *Hyperoodon ampullatus*; **Hypl:** *Hyperoodon planifrons*; **Megr:** *Mesoplodon grayi*; **Mela:** *Mes. layardii*; **Mepe:** *M. peruvianus*; **Messgr:** *Messapicetus gregarius*; **Naur:** *Nazcacetus urbinai*; **Nipl:** *Ninoziphius platyrostris*; **Tash:** *Tasmacetus shepherdi*; **Zica:** *Ziphius cavirostris*. The cluster of medium ziphiids corresponds to the species *M. bidens*, *M. europaeus*, *M. ginkgodens*, and *M. mirus*.

Supplementary Data

Supplementary data S1. Condylobasal lengths and postorbital widths from specimens of Ziphiidae used in the analysis.

Supplementary data S2. Pdf file containing a 3D reconstruction of the *genus and sp. indet.* NHMD 189993.